# Multi-Task Vehicle Routing Solver via Mixture of Specialized Experts under State-Decomposable MDP

Yuxin Pan[1][*]  Zhiguang Cao[2]  Chengyang Gu[3]  Liu Liu[4][†]
Peilin Zhao[4,6]  Yize Chen[5][†]  Fangzhen Lin[1][†]
[1]The Hong Kong University of Science and Technology
[2]Singapore Management University
[3]The Hong Kong University of Science and Technology (Guangzhou)
[4]Tencent AI Lab
[5]University of Alberta
[6]Shanghai Jiao Tong University
yuxin.pan@connect.ust.hk

## Abstract

Existing neural methods for multi-task vehicle routing problems (VRPs) typically learn unified solvers to handle multiple constraints simultaneously. However, they often underutilize the compositional structure of VRP variants, each derivable from a common set of basis VRP variants. This critical oversight causes unified solvers to miss out the potential benefits of basis solvers, each specialized for a basis VRP variant. To overcome this limitation, we propose a framework that enables unified solvers to perceive the shared-component nature across VRP variants by proactively reusing basis solvers, while mitigating the exponential growth of trained neural solvers. Specifically, we introduce a State-Decomposable MDP (SDMDP) that reformulates VRPs by expressing the state space as the Cartesian product of basis state spaces associated with basis VRP variants. More crucially, this formulation inherently yields the optimal basis policy for each basis VRP variant. Furthermore, a Latent Space-based SDMDP extension is developed by incorporating both the optimal basis policies and a learnable mixture function to enable the policy reuse in the latent space. Under mild assumptions, this extension provably recovers the optimal unified policy of SDMDP through the mixture function that computes the state embedding as a mapping from the basis state embeddings generated by optimal basis policies. For practical implementation, we introduce the Mixture-of-Specialized-Experts Solver (MoSES), which realizes basis policies through specialized Low-Rank Adaptation (LoRA) experts, and implements the mixture function via an adaptive gating mechanism. Extensive experiments conducted across VRP variants showcase the superiority of MoSES over prior methods. The source code is available at https://github.com/panyxy/moses_vrp.

## 1 Introduction

Vehicle routing problems (VRPs) are a canonical class of combinatorial optimization problems (COPs) with broad applications spanning transportation [19], logistics [6], and manufacturing [80]. While exact methods are computationally prohibitive [36], and heuristic methods rely on substantial domain-specific knowledge [23, 66], recent studies craft learning-based neural solvers for empirically sound performance with lower makespan and marginal domain expertise [67, 54, 34, 35]. Prevailing

---

[*]This work is done when Yuxin Pan works as an intern in Tencent AI Lab.
[†]Corresponding Authors.

approaches primarily target one specific VRP, with tremendous efforts dedicated to out-of-distribution (OOD) generalization [4, 84, 25, 15, 46, 77, 82, 56, 47]. However, practical applications involve multiple VRP variants with diverse node attributes and solution constraints, making the development of specialized neural solver costly due to the need for retraining from scratch.

Recent advances in multi-task neural solvers are conscious of inherent partial similarities present among these variants, enabling efficient knowledge transfer via shared embeddings. These methods commonly resort to specialized adaptations of a pretrained backbone model [41, 11]. Although these methods perform favorably without requiring training from scratch, they struggle to scale due to the combinatorial explosion of VRP variants. This is because arbitrary combinations of orthogonal attributes can introduce new variants, inevitably leading to costly repetitive fine-tuning and adapter proliferation. As an alternative, unified neural solvers instead are capable of handling numerous VRP variants simultaneously. These methods typically unify VRP variants via attribute composition [42], and propose various architectural innovations [83, 3, 38]. Although these methods treat VRP variants as combinations of attributes, they fail to fully exploit this compositional structure inherent in VRP variants. This structure implies that attributes of each VRP variant actually derive from a shared set of basis VRP variants, each characterized by a unique attribute. We argue that there exist basis solvers, each tailored to a specific basis VRP variant, may enjoy experience valuable for unified solvers.

To elucidate the possible valuable insights that basis neural solvers could offer for unified solvers, we exemplify the widely-used encoder-decoder based neural solver as a case study. The encoder is tasked with yielding static node embeddings for the subsequent step-wise solution decoding. Thus, each basis solver's encoder produces embeddings capturing unique attribute of its corresponding basis VRP variant. Since attributes of each VRP variant stem from shared basis variants, the unified solver's encoder possibly perceive both individual attribute information and inter-attribute connections in its embeddings. The decoder gradually builds solutions using static node embeddings and the dynamic context while adhering to constraints. Likewise, two critical properties emerge within each VRP variant: its dynamic context can be broken down into conditionally independent components, each corresponding to a distinct basis variant; and its constraints form a superset of those associated with corresponding basis variants. The internal embeddings of the unified solver's decoder thus may partially coincide with those of the decoders of basis solvers. Therefore, we argue that *the unified solver could benefit from valuable insights of basis solvers*. Moreover, the key insights of a neural solver predominantly reside in its continuous embeddings. This motivates us to reformulate VRPs with the aim of reusing basis solvers in the latent space.

In this paper, to seamlessly bridge between VRP variants and basis VRP variants, we propose a VRPs reformulation through the novel State-Decomposable MDP (SDMDP) framework. To be specific, this framework expresses the state space as the Cartesian product of basis state spaces, each associated with a basis VRP variant. As a result, any state admits a decomposition into conditionally independent basis states. Indeed, as disclosed above, SDMDP is fundamentally grounded in the observation that both static attributes and dynamic contexts of VRP variants originate from their corresponding basis variants. More importantly, SDMDP not only targets towards an optimal unified policy, but also yields optimal basis policies inherently, each tailored to a specific basis variant in cases where a VRP variant comprises only a single attribute. To fully reuse optimal basis policies, we operate under the primary assumptions that any unified policy can generate basis state embeddings and a mixture function exists to map these resulting basis state embeddings to the corresponding state embedding. Built upon this, we further develop a Latent Space-based SDMDP (LS-SDMDP) extension by incorporating the optimal basis policies and the mixture function. In this extension, each basis state inherent in a state is fed to its corresponding optimal basis policy for the basis state embedding. These embeddings are then transformed into the state embedding through the mixture function, which subsequently informs the action selection. Under mild assumptions, LS-SDMDP provably recovers the optimal unified policy for SDMDP. For practical implementation, we introduce the Mixture-of-Specialized-Experts Solver (MoSES), which realizes basis policies through specially designed Low-Rank Adaption (LoRA) [26] experts, each fine-tuned for a specific basis variant from a frozen pretrained backbone model, and implements the mixture function via an adaptive gating mechanism. In addition, we design multiple adaptive gating mechanisms for comparative analysis, and implement MoSES with different backbone networks to show its plug-and-play versatility. Extensive experiments across VRP variants validate the superiority of MoSES against prior methods.

## 2 Related Works

This Section reviews recent advances in task-specific neural VRP solvers, multi-task learning approaches for VRPs, and mixture-of-specialized-experts (MoSE) methods. Please refer to Appendix D for more detailed reviews.

**Task-Specific Neural VRP solvers.** Neural solvers, individually developed for each specific VRP, fall into three main categories: constructive methods end-to-end infer solutions via autoregressive mechanisms [67, 34, 35, 33, 51, 60, 84, 4, 17, 22, 30, 15, 46, 47], iterative methods refine solutions via local search operators until convergence [45, 7, 24, 50, 74, 49], and divide-and-conquer methods decompose problem instances into smaller solvable sub-instances [16, 32, 8, 39, 85, 25, 77, 82, 56]. However, these methods typically require specialized network architectures and retraining from scratch to handle numerous VRP variants, bringing about excessively high costs.

**Multi-Task Learning for VRPs.** To cope with practical scenarios involving multiple VRP variants, the efficient transfer learning is leveraged to obtain specialized neural solvers by considering inherent similarities among VRP variants. Current methods predominantly adopt either full-parameter fine-tuning [11] or problem-specific adapter fine-tuning [41, 14], both built on a pretrained backbone model. However, these methods struggle with the combinatorial explosion of VRP variants. *Notably, although the LoRA adapter is used for problem-specific adaptation in [41], its potential as part of a unified solver to handle the exponential growth of variants remains unexplored.* In contrast, unified solvers are designed to handle multiple VRP variants simultaneously. These methods commonly unify VRP variants via attribute compositions [42] and employ various architectural innovations such as mixture of experts (MoE) [83], modified Transformer [3], large language model (LLM) based encoder [29], dual attention model [38], mixture of depths (MoD) [20], specialized decoders [68, 40], diffusion model [37], or mixed-curvature based encoder [43]. However, the potential benefits of incorporating explicitly specialized basis solvers remain unexplored. *Notably, while MoE is employed in [83], it learns an implicit and less interpretable specialization rather than incorporating off-the-shelf basis solvers as experts.*

**Mixture of Specialized Experts.** Prevailing MoSE methods can be broadly categorized into two paradigms: merging entire models and module composition. Approaches based on merging entire models seek to combine independently trained models to efficiently achieve the performance comparable to model ensembling or multi-task learning [70, 52, 75, 64, 12, 1, 63, 61, 31, 76]. However, these approaches exhibit inferior OOD generalization, compared to layer-wise aggregation of expert models. Our implementation aligns more closely with the module composition paradigm which supports the finer-grained aggregation. These methods primarily include: selective adapter averaging [9, 59], module fusion via arithmetic operations [10, 81, 28], adapter routing based on task similarity [48, 21, 72], and adaptive mixture of LoRA experts (MoLE) [27, 78, 73, 44, 5, 13, 58, 53, 55, 79, 18]. However, the potential of MoLE methods for unified VRP solvers remains underexplored, and existing composition methods possibly prove inadequate for multi-task VRP scenarios.

## 3 Preliminaries

**VRP Variants.** We adopt the vehicle routing environment from [3]. A capacitated VRP (CVRP) instance of size $N$ is defined on a graph $\mathcal{G} = \{\mathcal{V}, \mathcal{E}\}$ with nodes $\mathcal{V} = \{v_0\} \bigcup \{v_i\}_{i=1}^N$ (depot and customers) and edges $\mathcal{E} = \{e(v_i, v_j) | 0 \leq i \neq j \leq N\}$. Each node $v_i$ $(i \geq 0)$ has coordinates $(x_i, y_i)$, with each customer $v_j$ $(j \geq 1)$ having demand $q_j^{\mathrm{LH}} > 0$. Each vehicle has a capacity $Q$. A feasible solution (i.e., tour) $\tau$ consists of subtours, each beginning and ending at the depot while visiting a customer subset, with each customer visited exactly once and total demand of each subtour not exceeding $Q$. The cost function $c(\cdot)$ is the total Euclidean length of the tour. The objective is to find the optimal tour $\tau^*$ with the minimal cost. CVRP, as a *Basis VRP Variant*, features the capacity constraint *(C)*, serving as the foundation for deriving the remaining *Basis VRP Variants* by adding one of the following constraints. 1) *Open Route (O):* A binary variable $o$ indicates whether the vehicle needs to return to depot $(o = 0)$ or not $(o = 1)$; 2) *Backhaul (B):* Each customer $v_j$ is either a linehaul with $q_j^{\mathrm{LH}} > 0$ or a backhaul with $q_j^{\mathrm{BH}} < 0$, where linehauls require deliveries and backhauls require pickups. VRPs with backhaul allow traversing both types in a mixed manner, but linehauls must precede backhauls in each subtour [62]. In VRPs without backhaul, only linehaul customers are present; 3) *Duration Limit (L):* Each subtour's cost cannot exceed a threshold $l^{\mathrm{dur}}$; 4) *Time Window*

*(TW):* Each node $v_i$ has a time window $[w_i^{\text{beg}}, w_i^{\text{end}}]$ and a service duration $w_i^{\text{dur}}$, requiring service to begin within this window. Vehicles arriving before $w_i^{\text{beg}}$ must wait until the window opens, and all vehicles must return to the depot by $w_0^{\text{end}}$. Thus, 16 VRP variants emerge from adding arbitrary combinations of the remaining four constraints to CVRP. Please refer to Appendix A.1 A.2 for details of the VRP variants.

**Learning to Solve VRPs.** We adopt the widely-used attention-based neural network [34, 3, 38] to parameterize the VRP policy $\pi_\theta$, which generates feasible solutions autoregressively through masked decoding. The encoder generates static node embeddings, which, with the dynamic context of the constructed partial tour $\tau^{(<t)}$, are fed to the decoder to output the probabilities of valid nodes for the next node $\tau^{(t)}$. The policy factorizes as $\pi_\theta(\tau|\mathcal{G}) = \prod_{t=1}^{T} \pi_\theta(\tau^{(t)}|\tau^{(<t)}, \mathcal{G})$, where $T$ is the solution horizon. REINFORCE [69] algorithm with reward $-c(\tau)$ is used to optimize the policy.

**Mixture of LoRA Experts.** LoRA [26] is a parameter-efficient fine-tuning method that adapts pretrained frozen LLMs through low-rank matrix factorization. For a linear layer with weights $W_0 \in \mathbb{R}^{d_1 \times d_2}$, it introduces trainable matrices $A \in \mathbb{R}^{r \times d_2}$ and $B \in \mathbb{R}^{d_1 \times r}$ (where $r < \min(d_1, d_2)$ denotes the LoRA rank), modifying the forward pass as $h^{\text{out}} = W_0 h^{\text{in}} + \beta B A h^{\text{in}}$, where $h^{\text{in}} \in \mathbb{R}^{d_2}$ and $h^{\text{out}} \in \mathbb{R}^{d_1}$ are the input and the output, and $\beta \in (0, 1]$. To enhance the cross-task generalization, $K$ task-specific LoRA experts $\{B_k A_k\}_{k=1}^{K}$ are integrated into the LLM [44]. A trainable gating function $G(h^{\text{in}}) = \text{softmax}(W^G h^{\text{in}})$ with weights $W^G \in \mathbb{R}^{K \times d_2}$ computes coefficients $\{\alpha_k\}_{k=1}^{K}$ for LoRA experts, yielding the forward pass as $h^{\text{out}} = W_0 h^{\text{in}} + \sum_{k=1}^{K} \alpha_k B_k A_k h^{\text{in}}$.

# 4 Methodology

In this Section, we first present the State-Decomposable MDP framework to reformulate VRP variants. To efficiently reuse readily available basis neural solvers, we extend it to the Latent Space-based SDMDP to enable the unified neural solver. Finally, we propose the Mixture-of-Specialized-Experts Solver which implements basis neural solvers using specialized LoRA experts.

## 4.1 State-Decomposable MDP

The State-Decomposable Markov Decision Process (SDMDP) framework is described by a 7-tuple $(\mathcal{S}, \mathcal{A}, \mathcal{P}, \mathcal{R}, \mu, \gamma, \bar{\mathcal{S}})$, which extends the standard MDP by introducing a *Full State Space* $\bar{\mathcal{S}}$. This full state space $\bar{\mathcal{S}}$ can be partitioned into $T + 1$ disjoint sets over a finite horizon $T$: $\bar{\mathcal{S}} = \bigcup_{t=0}^{T} \bar{\mathcal{S}}_t$. For each time step $0 \le t \le T$, the full state space $\bar{\mathcal{S}}_t$ can be further represented as the Cartesian product of $n + 1$ basis state spaces: $\bar{\mathcal{S}}_t = \prod_{i=0}^{n} \mathcal{S}_t^{(i)}$. This allows any full state $\bar{s}_t \in \bar{\mathcal{S}}_t$ to be broken down into $n + 1$ conditionally independent basis states: $\bar{s}_t = \{s_t^{(i)}\}_{i=0}^{n}$, where $s_t^{(i)} \in \mathcal{S}_t^{(i)}$, for $i = 0, \ldots, n$. Please note that the full state $\bar{s}_t$ is neither observed nor influenced by the agent.

Likewise, the *State Space* $\mathcal{S}$ can be partitioned as: $\mathcal{S} = \bigcup_{t=0}^{T} \mathcal{S}_t$. During each episode, prior to the policy rollout, the initial state space $\mathcal{S}_0$ is built by randomly sampling $m + 1$ basis state spaces, where $0 \le m \le n$. This results in $\mathcal{S}_0 = \prod_{i=0}^{m} \mathcal{S}_0^{(b_i)}$, where $\forall 0 \le i \ne j \le m$, $0 \le b_i \ne b_j \le n$. *Especially, we stipulate that $\mathcal{S}_0^{(b_0)} = \mathcal{S}_0^{(0)}$.* Please note that the value of $m$ may vary across different episodes. The initial state distribution $\mu$ is thus defined on $\mathcal{S}_0$. The state space $\mathcal{S}_t$ ($0 \le t \le T$) is defined as the joint space of $m + 1$ basis state spaces evolving from the initial sampled basis state spaces, such that $\mathcal{S}_t = \prod_{i=0}^{m} \mathcal{S}_t^{(b_i)}$. The state $s_t \in \mathcal{S}_t$ can be decomposed into $m + 1$ conditionally independent basis states: $s_t = \{s_t^{(b_i)}\}_{i=0}^{m}$, where $s_t^{(b_i)} \in \mathcal{S}_t^{(b_i)}$, for $i = 0, \ldots, m$.

The *Action Space* $\mathcal{A}$ is conditioned on the state $s_t$, denoted as $\mathcal{A}_t = \mathcal{A}(s_t)$, indicating that the basis states in the state $s_t$ jointly define the feasible action space. The *Transition Probability Function* $\mathcal{P}$ returns the probability distribution of the next state $s_{t+1} \in \mathcal{S}_{t+1}$ given the current state-action pair $(s_t, a_t) \in \mathcal{S}_t \times \mathcal{A}_t$. Due to the conditional independence of the basis states, the transition probability function $\mathcal{P}$ factorizes as: $\mathcal{P}(s_{t+1}|s_t, a_t) = \prod_{i=0}^{m} \mathcal{P}(s_{t+1}^{(b_i)}|s_t^{(b_i)}, a_t)$. The *Reward Function* $\mathcal{R}$ maps the state-action pair $(s_t, a_t)$ to a scalar reward $R_t$. $\gamma \in (0, 1]$ is the discount factor. Given the state $s_t$, the policy $\pi$ yields the probability distribution over $\mathcal{A}_t$.

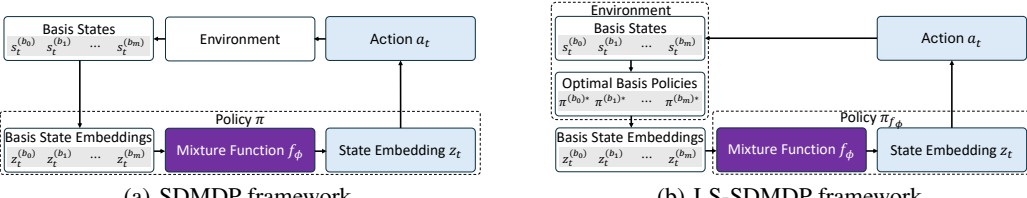

Figure 1: **Left:** SDMDP employs the policy that incorporates a mixture function. **Right:** LS-SDMDP integrates both the optimal basis policies and the mixture function.

We abuse $\tau$ to denote the trajectory $(s_0, a_0, \ldots, s_T)$. The objective of the SDMDP framework is to identify an *Optimal Unified Policy* $\pi^* = \arg\max_\pi \mathbb{E}_{s \sim \mu} \mathbb{E}_{\tau \sim (\pi, \mathcal{P})} [\sum_{t=0}^{T-1} \gamma^t R_t | s_0 = s] = \arg\max_\pi \mathbb{E}_{s \sim \mu} [V^{\pi, \mathcal{P}}(s)]$, where $V^{\pi, \mathcal{P}}$ is the value function. This framework defines the *0-th Basis Task* using the initial state distribution over $\mathcal{S}_0^{(0)}$, while the *i-th Basis Task* $(1 \leq i \leq n)$ extends this with initial state distribution over $\mathcal{S}_0^{(0)} \times \mathcal{S}_0^{(i)}$. The *i-th Optimal Basis Policy* $\pi^{(i)*}$ $(0 \leq i \leq n)$ is the policy that maximizes the expected cumulative rewards for the *i*-th basis Task.

***Remark.*** To reformulate VRP variants within the SDMDP framework, we begin by aligning basis states with basis VRP variants. At each step $t$, the state $s_t$ decomposes into 5 basis states: 0) *CVRP*: $s_t^{(0)}$ consists of node coordinates $\{x_i, y_i\}_{i=0}^N$, linehaul demands $\{q_i^{\mathrm{LH}}\}_{i=1}^N$, and the remaining linehaul capacity $Q_t^{\mathrm{LH}}$; 1) *Open Route*: $s_t^{(1)}$ introduces the binary variable $o$; 2) *Backhaul*: $s_t^{(2)}$ integrates backhaul demands $\{q_i^{\mathrm{BH}}\}_{i=1}^N$ and the remaining backhaul capacity $Q_t^{\mathrm{BH}}$; 3) *Duration Limit*: $s_t^{(3)}$ encompasses the duration limit $l^{\mathrm{dur}}$ and the current traveled length $l_t^{\mathrm{cur}}$ along the present subtour; 4) *Time Window*: $s_t^{(4)}$ includes time windows $\{w_i^{\mathrm{beg}}, w_i^{\mathrm{end}}\}_{i=0}^N$, service durations $\{w_i^{\mathrm{dur}}\}_{i=0}^N$, and the current time $w_t^{\mathrm{cur}}$. Each VRP variant can be formed by composing $(m + 1)$ $(0 \leq m \leq 4)$ basis state spaces, using SDMDP's initial sampling mechanism prior to the policy rollout. During rollout, each current basis state evolves conditionally independently from its corresponding previous basis state given the action. The action space is defined by a masking mechanism that filters out nodes according to visitation status and VRP constraints. Please refer to Appendix A.3 A.4 for details on VRP constraints and formulation. There exist 5 basis tasks, each associated with a basis VRP variant.

**Theorem 1.** *The optimal unified policy $\pi^*$ and the i-th optimal basis policy $\pi^{(i)*}$ coincide in their value functions for each state $s_t$ associated with the i-th basis task: $V^{\pi^*, \mathcal{P}}(s_t) = V^{\pi^{(i)*}, \mathcal{P}}(s_t)$. Furthermore, if both the optimal unified policy and the optimal basis policy are unique, then for each state-action pair $(s_t, a_t)$ corresponding to the i-th basis task, it holds that $\pi^*(a_t | s_t) = \pi^{(i)*}(a_t | s_t)$.*

*Proof.* Please refer to Appendix C for a detailed proof of Theorem 1. $\qquad \square$

**Assumption 1.** *In the SDMDP framework, any state $s$ is composed of $m+1$ conditionally independent basis states, denoted as $s = \{s^{(b_i)}\}_{i=0}^m$. Accordingly, we assume that any policy $\pi$ is capable of extracting the basis state embedding $z^{(b_i)} \in \mathbb{R}^d$ for each $s^{(b_i)}$, where $i = 0, \ldots, m$. Under this assumption, we further posit that there exists a deterministic bijective mixture function $f_\phi : \mathcal{S} \times \prod_{i=0}^m \mathbb{R}^d \to \mathbb{R}^d$, parameterized by $\phi$, which maps the basis state embeddings $\{z^{(b_i)}\}_{i=0}^m$ to the state embedding $z \in \mathbb{R}^d$ for the given state $s$, represented as $z = f_\phi(z^{(b_0)}, \ldots, z^{(b_m)}; s)$. Thus, the policy defined over the action space can be rewritten as*

$$\pi(a|s) = \sum_z \pi(a|z)\pi(z|s) = \sum_{z^{(b_0)}, \ldots, z^{(b_m)}} \pi(a | f_\phi(z^{(b_0)}, \ldots, z^{(b_m)}; s)) \prod_{i=0}^m \pi(z^{(b_i)} | s^{(b_i)}) \tag{1}$$

*where $\prod_{i=0}^m \pi(z^{(b_i)} | s^{(b_i)}) = \pi(z^{(b_0)}, \ldots, z^{(b_m)} | s)$. The second equivalence in Equation 1 holds because $f_\phi$ is assumed to be deterministic.*

**Assumption 2.** *For any state $s$, and for any two policies $\pi$ and $\pi'$, we assume that if $\forall a \in \mathcal{A}(s), \pi(a|s) = \pi'(a|s)$, then $\forall z \in \mathbb{R}^d, \pi(z|s) = \pi'(z|s)$, and conversely.*

***Remark.*** Theorem 1 discloses the partial consensus between the optimal unified policy and each optimal basis policy, laying the foundation of the policy reuse. Motivated by the observation that neural

networks primarily encode knowledge in their embeddings, with distinct features captured at different layers, we develop a latent space approach for reusing optimal basis policies. Assumptions 1 2 serve as the prerequisites for Theorem 2. Figure 1(a) illustrates the SDMDP framework.

## 4.2 Latent Space-based SDMDP

The Latent Space-based SDMDP (LS-SDMDP) incorporates all elements of SDMDP, and extends SDMDP by introducing $(\pi^{(0)*}, \ldots, \pi^{(n)*}, f_\phi)$. During each episode, at each time step $0 \leq t \leq T$, a state $s_t \in \mathcal{S}_t$ is sampled from either the initial distribution $\mu(s_0)$ or the transition probability function $\mathcal{P}(s_{t+1}|s_t, a_t)$, both of which are well defined within the SDMDP framework. Following that, each component $s_t^{(b_i)}$ of $s_t$ ($0 \leq i \leq m$) is fed to the corresponding optimal basis policy $\pi^{(b_i)*}$ for the embedding $z_t^{(b_i)} \in \mathbb{R}^d$. Collectively, these embeddings form a tuple, denoted as $(z_t^{(b_0)}, \ldots, z_t^{(b_m)})$. To infer the state embedding $z_t$ for the state $s_t$, $f_\phi$ takes as input this tuple along with the state, formally expressed as $z_t = f_\phi(z_t^{(b_0)}, \ldots, z_t^{(b_m)}; s_t)$. The policy, exclusively designed for LS-SDMDP, directly observes the embedding vector $z_t$ and outputs an action distribution, denoted as $\pi_z$. For brevity, the integration of $f_\phi$ into the policy $\pi_z$ is abbreviated as $\pi_{f_\phi}$, such that $\pi_z(\cdot|z_t) = \pi_{f_\phi}(\cdot|z_t^{(b_0)}, \ldots, z_t^{(b_m)}; s)$. Accordingly, the underlying initial distribution over $z_0$ and the transition probability function for $z_{t+1}$ given $s_t$ and $a_t$ are written as follows

$$\mu_z(z_0) = \mu(s_0) \prod_{i=0}^{m} \pi^{(b_i)*}(z_0^{(b_i)}|s_0^{(b_i)}); \quad \mathcal{P}_z(z_{t+1}|s_t, a_t) = \mathcal{P}(s_{t+1}|s_t, a_t) \prod_{i=0}^{m} \pi^{(b_i)*}(z_{t+1}^{(b_i)}|s_{t+1}^{(b_i)}). \quad (2)$$

Equation 2 holds due to the deterministic and bijective nature of $f_\phi$. The reward function remains unchanged. The objective is to discover an optimal unified policy $\pi_{f_\phi}^*$ which can maximize the expected discounted cumulative rewards.

**Theorem 2.** *Let $J(\pi, \mathcal{P}, \mu)$ and $J(\pi_{f_\phi}, \mathcal{P}_z, \mu_z)$ denote the objective functions (expected returns) in SDMDP and LS-SDMDP, respectively. By Theorem 1 and Assumptions 1 2, it follows that the values of the objective functions are equal at their respective optimal policies, formally written as $J(\pi^*, \mathcal{P}, \mu) = J(\pi_{f_\phi}^*, \mathcal{P}_z, \mu_z)$. Moreover, the value functions of SDMDP and LS-SDMDP at their respective optimal policies satisfy the following relationship $V^{\pi^*, \mathcal{P}}(s) = \mathbb{E}_{z^{(b_0)} \sim \pi^{(b_0)*}} \cdots \mathbb{E}_{z^{(b_m)} \sim \pi^{(b_m)*}} V^{\pi_{f_\phi}^*, \mathcal{P}_z}(z^{(b_0)}, \ldots, z^{(b_m)}; s)$.*

*Proof.* Please refer to Appendix C for a detailed proof of Theorem 2. □

***Remark.*** Theorem 2 indicates that the optimal unified policy $\pi^*$ of SDMDP can be recovered given both the optimal unified policy $\pi_{f_\phi}^*$ of LS-SDMDP and optimal basis policies $(\pi^{(0)*}, \ldots, \pi^{(n)*})$. Figure 1(b) depicts the LS-SDMDP framework.

## 4.3 Mixture-of-Specialized-Experts Solver

For the practical implementation, we need to realize both the optimal basis policies and the mixture function to enable the policy reuse in the latent space. If we consider the embedding $z^{(b_i)}$ ($0 \leq i \leq m$) generated by the optimal basis policy $\pi^{(b_i)*}$ to encapsulate all layer-wise embeddings, the mixture function would effectively perform entire-model merging. However, as disclosed in [55, 53], such entire-model merging leads to inferior OOD generalization. We therefore introduce the Mixture-of-Specialized-Experts Solver (MoSES), which implements a layer-wise and token-wise aggregation approach through three stages: 1) pretraining a shared backbone model; 2) fine-tuning specialized experts, and 3) dynamically aggregating experts.

**Pretraining a shared backbone model.** Since all the other basis VRP variants are derived from CVRP by adding different constraints, we pretrain the shared backbone model exclusively on CVRP instances and freeze its parameters throughout subsequent stages. Please note the backbone model indeed acts as the 0-th optimal basis policy $\pi^{(0)*}$.

**Fine-tuning specialized experts.** To obtain the optimal basis policy $\pi^{(i)*}$ ($i > 0$), we employ the specialized LoRA expert that performs parameter-efficient fine-tuning on the frozen backbone model. Problem instances of any basis VRP variant (excluding CVRP) are OOD inputs to the frozen

backbone model, leading to task-misaligned features in embeddings generated from the backbone model. We thus propose the Gated-LoRA method that uses a dynamic gating mechanism with the trainable weights $W^g \in \mathbb{R}^{d_2}$ to suppress these irrelevant features from the backbone. Specifically, for the $i$-th optimal basis policy $\pi^{(i)*}$ ($i > 0$), the forward pass through a frozen backbone linear layer with weights $W_0$ augmented with trainable LoRA matrices $B_i$ and $A_i$ is computed as:

$$h^{\text{out}} = \text{sigmoid}(\langle W^g, h^{\text{in}} \rangle) W_0 h^{\text{in}} + B_i A_i h^{\text{in}}. \tag{3}$$

**Dynamically aggregating experts.** In this stage, we freeze both the backbone weights $W_0$ and the LoRA weights $\{B_i A_i\}_{i=1}^4$ in the unified policy. The adaptive gating function $G(h^{\text{in}}) = \text{act}(W^G h^{\text{in}})$ with weights $W^G \in \mathbb{R}^{5 \times d_2}$ computes coefficients $\{\alpha_i\}_0^4$, where $\text{act}(\cdot)$ is the activation function. Let $\tilde{h}^{\text{out}}$ denote the output of the unknown optimal unified solver. The aggregation goal is to determine coefficients $\{\alpha_i\}_0^4$ satisfying the linear system $\sum_{i=0}^4 \alpha_i W_i h^{\text{in}} = \tilde{h}^{\text{out}}$ (where $W_i = B_i A_i$, for $i \geq 1$). However, this linear system may not be solvable. We thus introduce the trainable LoRA weights $\hat{B}\hat{A}$ to model the residual. The forward pass of a linear layer in the unified policy is written as:

$$h^{\text{out}} = \alpha_0 W_0 h^{\text{in}} + \sum_{i=1}^4 \alpha_i B_i A_i h^{\text{in}} + \hat{B}\hat{A} h^{\text{in}}; \quad \{\alpha_i\}_{i=0}^4 = G(h^{\text{in}}) = \text{act}(W^G h^{\text{in}}). \tag{4}$$

We propose three activation functions for the gating mechanism: 1) $\text{softmax}(\cdot)$ enforces the convex combination; 2) $\text{norm\_softplus}(\cdot)$ $L_1$-normalizes $\text{softplus}(W^G h^{\text{in}})$ to prevent gradient vanishing; 3) $\text{sigmoid}(\cdot)$ expands the coefficient space by relaxing the unit-sum constraint. Additionally, we introduce three routing strategies: 1) *Dense Routing* activates all optimal basis policies ; 2) *Variant-Aware Routing-I* selects Top-$K$ optimal basis policies, where $K$ is the number of basis variants in the current VRP variant. 3) *Variant-Aware Routing-II* selects optimal basis policies corresponding to basis variants present in the current VRP variant;

## 5 Experiments

In this Section, we empirically validate the superiority of MoSES through evaluations on 16 VRP variants with five constraints, supplemented by hyperparameter and ablation studies. All experiments are conducted on NVIDIA Tesla V100-32GB GPUs on and Intel(R) Xeon(R) Platinum 8255C CPU @ 2.50GHz. Please refer to Appendix B for more empirical results, hyperparameter and ablation studies, along with the analysis.

**Baselines.** *Traditional Solvers:* We benchmark against the open-source PyVRP [71] solver built on HGS-CVRP [65], and the widely-used Google OR-Tools [57]. Both solvers process each instance on a single CPU core with time limits of 10s for instances with 50 nodes; and 20s for instances with 100 nodes, while we parallelize their execution across 16 CPU cores for efficiency. *Neural Solvers:* We compare our method with state-of-the-art unified neural solvers for multi-task VRPs. MTPOMO [42] extends POMO [35] and unifies VRP variants. MVMoE [83] introduces mixture-of-experts. RouteFinder [3] utilizes the mixed batch training to produce three models: RF-POMO (MTPOMO-based), RF-MoE (MVMoE-based) and RF-TE (modiled Transformer-based). CaDA [38] enhances model capacity through dual attention mechanism. Since CaDA lacks publicly available code, we implement it according to the original paper specifications. Our reproduction of CaDA achieves comparable performance to the reported results.

**MoSES Architecture.** To demonstrate the plug-and-play versatility of MoSES, we implement it upon both RF-TE and CaDA, denoted as MoSES(RF) and MoSES(CaDA) respectively. MoSES(RF) uses $\text{norm\_softplus}(\cdot)$ activation for its adaptive gating function $G(\cdot)$, while MoSES(CaDA) utilizes $\text{sigmoid}(\cdot)$ activation. Both adopt the dense routing strategy, and set LoRA rank as 32 for both frozen modules $\{B_i A_i\}_{i=1}^4$ and the trainable module $\hat{B}\hat{A}$.

**Training and Evaluation.** We consider two problem scales $N = \{50, 100\}$, and adopt the same training settings with prior works [3, 38] for baseline models. We optimize MoSES using REIN-FORCE [69]. The backbone model is first pretrained on randomly generated CVRP instances. Then, the LoRA adapter fine-tuning produces 4 specialized experts, each optimized for a distinct VRP variant: OVRP, VRPB, VRPL, and VRPTW. Finally, the unified solver is trained on all 16 VRP variants. All phases share the training hyperparameters. Each model undergoes 300 training epochs,

Table 1: Performance comparison of multi-task VRP solvers. The lower, the better (↓).

| Task | Solver | N=50 Cost | N=50 Gap | N=50 Time | N=100 Cost | N=100 Gap | N=100 Time |
|---|---|---|---|---|---|---|---|
| CVRP | HGS-PyVRP | 10.372 | * | 10.4m | 15.628 | * | 20.8m |
| | OR-Tools | 10.572 | 1.907% | 10.4m | 16.280 | 4.178% | 20.8m |
| | MTPOMO | 10.518 | 1.408% | 2s | 15.933 | 1.986% | 8s |
| | MVMoE | 10.501 | 1.241% | 3s | 15.888 | 1.692% | 11s |
| | RF-POMO | 10.508 | 1.315% | 2s | 15.908 | 1.830% | 8s |
| | RF-MoE | 10.499 | 1.228% | 2s | 15.877 | 1.624% | 11s |
| | RF-TE | 10.504 | 1.276% | 2s | 15.857 | 1.507% | 8s |
| | MoSES(RF) | **10.465** | **0.900%** | 6s | **15.808** | **1.190%** | 21s |
| | CaDA | 10.483 | 1.072% | 3s | **15.831** | **1.336%** | 10s |
| | MoSES(CaDA) | **10.462** | **0.873%** | 7s | 15.833 | 1.354% | 24s |
| OVRP | HGS-PyVRP | 6.507 | * | 10.4m | 9.725 | * | 20.8m |
| | OR-Tools | 6.553 | 0.686% | 10.4m | 9.995 | 2.732% | 20.8m |
| | MTPOMO | 6.718 | 3.211% | 2s | 10.210 | 4.959% | 8s |
| | MVMoE | 6.702 | 2.969% | 3s | 10.176 | 4.615% | 11s |
| | RF-POMO | 6.698 | 2.906% | 2s | 10.181 | 4.671% | 8s |
| | RF-MoE | 6.697 | 2.879% | 3s | 10.139 | 4.238% | 11s |
| | RF-TE | 6.684 | 2.693% | 2s | 10.121 | 4.060% | 8s |
| | MoSES(RF) | **6.632** | **1.892%** | 5s | **10.064** | **3.469%** | 20s |
| | CaDA | 6.662 | 2.350% | 2s | 10.093 | 3.763% | 10s |
| | MoSES(CaDA) | **6.629** | **1.857%** | 7s | **10.084** | **3.679%** | 24s |
| VRPB | HGS-PyVRP | 9.687 | * | 10.4m | 14.377 | * | 20.8m |
| | OR-Tools | 9.802 | 1.159% | 10.4m | 14.933 | 3.853% | 20.8m |
| | MTPOMO | 10.033 | 3.564% | 2s | 15.082 | 4.917% | 8s |
| | MVMoE | 10.005 | 3.268% | 3s | 15.022 | 4.506% | 10s |
| | RF-POMO | 9.996 | 3.173% | 2s | 15.016 | 4.465% | 8s |
| | RF-MoE | 9.980 | 3.014% | 3s | 14.973 | 4.165% | 10s |
| | RF-TE | 9.978 | 2.996% | 2s | 14.942 | 3.950% | 8s |
| | MoSES(RF) | **9.915** | **2.342%** | 5s | **14.884** | **3.546%** | 19s |
| | CaDA | 9.945 | 2.654% | 2s | 14.905 | 3.684% | 10s |
| | MoSES(CaDA) | **9.904** | **2.225%** | 7s | **14.901** | **3.668%** | 23s |
| VRPBL | HGS-PyVRP | 10.186 | * | 10.4m | 14.779 | * | 20.8m |
| | OR-Tools | 10.331 | 1.390% | 10.4m | 15.426 | 4.338% | 20.8m |
| | MTPOMO | 10.672 | 4.699% | 2s | 15.712 | 6.253% | 8s |
| | MVMoE | 10.637 | 4.349% | 3s | 15.640 | 5.763% | 11s |
| | RF-POMO | 10.592 | 3.937% | 2s | 15.628 | 5.696% | 8s |
| | RF-MoE | 10.575 | 3.767% | 3s | 15.541 | 5.121% | 10s |
| | RF-TE | 10.578 | 3.798% | 2s | 15.528 | 5.038% | 8s |
| | MoSES(RF) | **10.518** | **3.185%** | 6s | **15.469** | **4.638%** | 20s |
| | CaDA | 10.535 | 3.379% | 2s | 15.481 | 4.713% | 10s |
| | MoSES(CaDA) | **10.517** | **3.193%** | 7s | **15.478** | **4.705%** | 24s |
| VRPBTW | HGS-PyVRP | 18.292 | * | 10.4m | 29.467 | * | 20.8m |
| | OR-Tools | 18.366 | 0.383% | 10.4m | 29.945 | 1.597% | 20.8m |
| | MTPOMO | 18.639 | 1.876% | 2s | 30.435 | 3.278% | 9s |
| | MVMoE | 18.640 | 1.884% | 3s | 30.438 | 3.287% | 12s |
| | RF-POMO | 18.601 | 1.669% | 2s | 30.343 | 2.967% | 9s |
| | RF-MoE | 18.617 | 1.760% | 3s | 30.339 | 2.947% | 12s |
| | RF-TE | 18.600 | 1.675% | 2s | 30.240 | 2.618% | 9s |
| | MoSES(RF) | **18.499** | **1.121%** | 6s | **30.148** | **2.303%** | 21s |
| | CaDA | 18.534 | 1.302% | 3s | 30.131 | 2.242% | 11s |
| | MoSES(CaDA) | **18.495** | **1.095%** | 8s | **30.050** | **1.969%** | 25s |
| OVRPB | HGS-PyVRP | 6.898 | * | 10.4m | 10.335 | * | 20.8m |
| | OR-Tools | 6.928 | 0.412% | 10.4m | 10.577 | 2.315% | 20.8m |
| | MTPOMO | 7.108 | 3.004% | 2s | 10.878 | 5.224% | 8s |
| | MVMoE | 7.090 | 2.743% | 3s | 10.840 | 4.859% | 11s |
| | RF-POMO | 7.086 | 2.689% | 2s | 10.836 | 4.823% | 8s |
| | RF-MoE | 7.080 | 2.613% | 3s | 10.806 | 4.526% | 11s |
| | RF-TE | 7.071 | 2.477% | 2s | 10.772 | 4.212% | 8s |
| | MoSES(RF) | **7.037** | **1.979%** | 6s | **10.733** | **3.829%** | 20s |
| | CaDA | 7.040 | 2.034% | 2s | 10.724 | 3.738% | 10s |
| | MoSES(CaDA) | **7.034** | **1.942%** | 7s | 10.726 | 3.765% | 24s |
| OVRPBLTW | HGS-PyVRP | 11.668 | * | 10.4m | 19.156 | * | 20.8m |
| | OR-Tools | 11.681 | 0.106% | 10.4m | 19.305 | 0.767% | 20.8m |
| | MTPOMO | 11.817 | 1.259% | 3s | 19.637 | 2.494% | 9s |
| | MVMoE | 11.823 | 1.303% | 4s | 19.641 | 2.516% | 12s |
| | RF-POMO | 11.805 | 1.155% | 3s | 19.608 | 2.344% | 10s |
| | RF-MoE | 11.823 | 1.307% | 4s | 19.607 | 2.334% | 12s |
| | RF-TE | 11.804 | 1.147% | 2s | 19.551 | 2.045% | 9s |
| | MoSES(RF) | **11.762** | **0.791%** | 6s | 19.508 | 1.821% | 22s |
| | CaDA | 11.771 | 0.865% | 2s | 19.471 | 1.626% | 11s |
| | MoSES(CaDA) | **11.761** | **0.781%** | 8s | **19.440** | **1.470%** | 26s |
| OVRPL | HGS-PyVRP | 6.507 | * | 10.4m | 9.724 | * | 20.8m |
| | OR-Tools | 6.552 | 0.668% | 10.4m | 10.001 | 2.791% | 20.8m |
| | MTPOMO | 6.719 | 3.229% | 2s | 10.214 | 5.000% | 8s |
| | MVMoE | 6.707 | 3.029% | 3s | 10.184 | 4.697% | 11s |
| | RF-POMO | 6.701 | 2.951% | 2s | 10.180 | 4.662% | 8s |
| | RF-MoE | 6.696 | 2.870% | 3s | 10.141 | 4.253% | 11s |
| | RF-TE | 6.685 | 2.713% | 2s | 10.121 | 4.054% | 8s |
| | MoSES(RF) | **6.634** | **1.917%** | 6s | **10.063** | **3.463%** | 20s |
| | CaDA | 6.661 | 2.335% | 2s | 10.093 | 3.766% | 11s |
| | MoSES(CaDA) | **6.629** | **1.846%** | 7s | **10.081** | **3.652%** | 24s |

| Task | Solver | N=50 Cost | N=50 Gap | N=50 Time | N=100 Cost | N=100 Gap | N=100 Time |
|---|---|---|---|---|---|---|---|
| VRPTW | HGS-PyVRP | 16.031 | * | 10.4m | 25.423 | * | 20.8m |
| | OR-Tools | 16.089 | 0.347% | 10.4m | 25.814 | 1.506% | 20.8m |
| | MTPOMO | 16.409 | 2.358% | 2s | 26.410 | 3.863% | 9s |
| | MVMoE | 16.405 | 2.333% | 3s | 26.391 | 3.793% | 11s |
| | RF-POMO | 16.366 | 2.089% | 2s | 26.335 | 3.570% | 9s |
| | RF-MoE | 16.390 | 2.239% | 3s | 26.319 | 3.506% | 11s |
| | RF-TE | 16.363 | 2.069% | 2s | 26.234 | 3.177% | 8s |
| | MoSES(RF) | **16.264** | **1.445%** | 6s | **26.143** | **2.822%** | 21s |
| | CaDA | 16.297 | 1.652% | 2s | 26.128 | 2.753% | 10s |
| | MoSES(CaDA) | **16.262** | **1.435%** | 7s | **26.032** | **2.383%** | 25s |
| VRPL | HGS-PyVRP | 10.587 | * | 10.4m | 15.766 | * | 20.8m |
| | OR-Tools | 10.570 | 2.343% | 10.4m | 16.466 | 5.302% | 20.8m |
| | MTPOMO | 10.775 | 1.732% | 2s | 16.151 | 2.445% | 8s |
| | MVMoE | 10.751 | 1.508% | 3s | 16.099 | 2.117% | 11s |
| | RF-POMO | 10.751 | 1.525% | 2s | 16.106 | 2.166% | 8s |
| | RF-MoE | 10.737 | 1.388% | 3s | 16.070 | 1.937% | 11s |
| | RF-TE | 10.748 | 1.499% | 2s | 16.051 | 1.829% | 8s |
| | MoSES(RF) | **10.704** | **1.089%** | 5s | **16.005** | **1.532%** | 20s |
| | CaDA | 10.722 | 1.252% | 2s | 16.062 | 1.662% | 10s |
| | MoSES(CaDA) | **10.704** | **1.083%** | 7s | **16.024** | **1.659%** | 24s |
| OVRPTW | HGS-PyVRP | 10.510 | * | 10.4m | 16.926 | * | 20.8m |
| | OR-Tools | 10.519 | 0.078% | 10.4m | 17.027 | 0.583% | 20.8m |
| | MTPOMO | 10.667 | 1.472% | 2s | 17.421 | 2.896% | 9s |
| | MVMoE | 10.669 | 1.495% | 3s | 17.416 | 2.874% | 12s |
| | RF-POMO | 10.657 | 1.376% | 2s | 17.392 | 2.725% | 9s |
| | RF-MoE | 10.673 | 1.533% | 3s | 17.387 | 2.698% | 12s |
| | RF-TE | 10.652 | 1.328% | 2s | 17.326 | 2.341% | 9s |
| | MoSES(RF) | **10.613** | **0.959%** | 6s | **17.284** | **2.101%** | 21s |
| | CaDA | 10.621 | 1.037% | 3s | 17.253 | 1.906% | 11s |
| | MoSES(CaDA) | **10.611** | **0.946%** | 8s | **17.217** | **1.702%** | 26s |
| VRPBLTW | HGS-PyVRP | 18.361 | * | 10.4m | 29.026 | * | 20.8m |
| | OR-Tools | 18.422 | 0.332% | 10.4m | 29.830 | 2.770% | 20.8m |
| | MTPOMO | 18.990 | 2.130% | 2s | 30.896 | 3.616% | 9s |
| | MVMoE | 18.986 | 2.106% | 3s | 30.893 | 3.612% | 12s |
| | RF-POMO | 18.937 | 1.853% | 2s | 30.794 | 3.278% | 9s |
| | RF-MoE | 18.956 | 1.956% | 3s | 30.807 | 3.321% | 12s |
| | RF-TE | 18.941 | 1.877% | 2s | 30.688 | 2.923% | 9s |
| | MoSES(RF) | **18.846** | **1.370%** | 6s | **30.627** | **2.712%** | 22s |
| | CaDA | 18.877 | 1.531% | 2s | 30.586 | 2.579% | 11s |
| | MoSES(CaDA) | **18.858** | **1.425%** | 8s | **30.510** | **2.329%** | 26s |
| VRPLTW | HGS-PyVRP | 16.356 | * | 10.4m | 25.757 | * | 20.8m |
| | OR-Tools | 16.441 | 0.499% | 10.4m | 26.259 | 1.899% | 20.8m |
| | MTPOMO | 16.823 | 2.818% | 2s | 26.891 | 4.364% | 9s |
| | MVMoE | 16.811 | 2.751% | 3s | 26.866 | 4.271% | 12s |
| | RF-POMO | 16.750 | 2.383% | 2s | 26.784 | 3.951% | 9s |
| | RF-MoE | 16.776 | 2.547% | 3s | 26.775 | 3.918% | 12s |
| | RF-TE | 16.763 | 2.460% | 2s | 26.691 | 3.587% | 9s |
| | MoSES(RF) | **16.657** | **1.811%** | 6s | **26.620** | **3.320%** | 21s |
| | CaDA | 16.694 | 2.038% | 2s | 26.592 | 3.204% | 11s |
| | MoSES(CaDA) | **16.667** | **1.864%** | 8s | **26.493** | **2.824%** | 25s |
| OVRPBL | HGS-PyVRP | 6.899 | * | 10.4m | 10.335 | * | 20.8m |
| | OR-Tools | 6.927 | 0.386% | 10.4m | 10.582 | 2.363% | 20.8m |
| | MTPOMO | 7.112 | 3.056% | 2s | 10.883 | 5.272% | 8s |
| | MVMoE | 7.098 | 2.850% | 3s | 10.847 | 4.928% | 11s |
| | RF-POMO | 7.087 | 2.695% | 2s | 10.837 | 4.835% | 8s |
| | RF-MoE | 7.083 | 2.635% | 3s | 10.807 | 4.540% | 11s |
| | RF-TE | 7.075 | 2.515% | 2s | 10.779 | 4.268% | 8s |
| | MoSES(RF) | **7.040** | **2.014%** | 6s | **10.736** | **3.862%** | 20s |
| | CaDA | 7.042 | 2.045% | 2s | 10.723 | 3.732% | 10s |
| | MoSES(CaDA) | **7.036** | **1.964%** | 7s | 10.724 | 3.743% | 24s |
| OVRPBTW | HGS-PyVRP | 11.669 | * | 10.4m | 19.156 | * | 20.8m |
| | OR-Tools | 11.682 | 0.109% | 10.4m | 19.303 | 0.757% | 20.8m |
| | MTPOMO | 11.814 | 1.231% | 3s | 19.635 | 2.484% | 9s |
| | MVMoE | 11.819 | 1.272% | 4s | 19.639 | 2.505% | 13s |
| | RF-POMO | 11.804 | 1.148% | 3s | 19.608 | 2.343% | 10s |
| | RF-MoE | 11.823 | 1.300% | 4s | 19.606 | 2.327% | 12s |
| | RF-TE | 11.805 | 1.151% | 2s | 19.551 | 2.046% | 9s |
| | MoSES(RF) | **11.761** | **0.783%** | 6s | 19.509 | 1.829% | 22s |
| | CaDA | 11.770 | 0.854% | 2s | 19.472 | 1.630% | 11s |
| | MoSES(CaDA) | **11.760** | **0.773%** | 8s | **19.441** | **1.475%** | 26s |
| OVRPLTW | HGS-PyVRP | 10.510 | * | 10.4m | 16.926 | * | 20.8m |
| | OR-Tools | 10.497 | 0.114% | 10.4m | 17.023 | 0.728% | 20.8m |
| | MTPOMO | 10.670 | 1.503% | 2s | 17.420 | 2.892% | 9s |
| | MVMoE | 10.671 | 1.511% | 3s | 17.418 | 2.881% | 12s |
| | RF-POMO | 10.657 | 1.372% | 3s | 17.392 | 2.727% | 9s |
| | RF-MoE | 10.656 | 1.532% | 3s | 17.385 | 2.690% | 12s |
| | RF-TE | 10.652 | 1.330% | 2s | 17.327 | 2.348% | 9s |
| | MoSES(RF) | **10.613** | **0.962%** | 6s | **17.281** | **2.081%** | 22s |
| | CaDA | 10.622 | 1.045% | 2s | 17.255 | 1.914% | 11s |
| | MoSES(CaDA) | **10.611** | **0.940%** | 8s | **17.219** | **1.714%** | 26s |

each containing 100,000 VRP instances generated on the fly. Adam Optimizer is used with a learning rate of $3 \times 10^{-4}$, weight decay of $1 \times 10^{-6}$, and batch size of 256. We decay the learning rate by a factor of 10 at epochs 270 and 295. During evaluation, each neural solver employs greedy multi-start

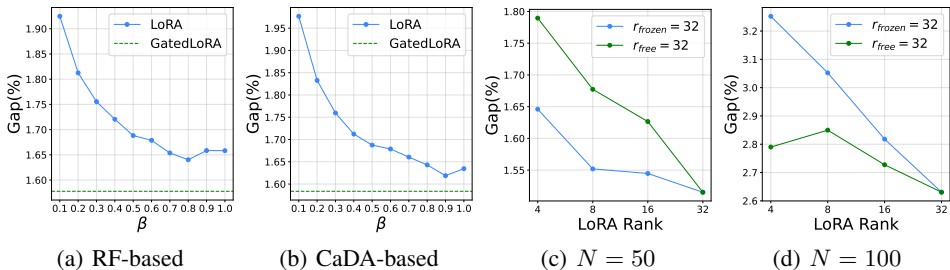

| (a) RF-based | (b) CaDA-based | (c) $N = 50$ | (d) $N = 100$ |

Figure 2: 2(a) 2(b) compare Gated-LoRA against standard LoRA with varying $\beta \in (0, 1]$. 2(c) 2(d) investigate the effect of LoRA ranks on the model performance.

rollouts with $8\times$ augmentations, selecting the best one from the generated solutions per instance. We report average costs and optimality gaps over 1K test instances, and the total time taken for testing. Gaps are calculated w.r.t. the results of the best heuristic solver (i.e., $*$ in Table 1).

## 5.1 Empirical Results

We conduct comprehensive benchmarking across all 16 VRP variants, with complete results presented in Table 1. MoSES(RF) consistently outperforms over its baseline RF-TE across all 16 VRP variants, achieving both lower solution costs and reduced optimality gaps for problem scales of $N = 50$ and $N = 100$. In terms of average optimality gap over 16 VRP variants, RF-TE achieves 2.063% and 3.125% for $N = 50$ and $N = 100$, respectively, while MoSES(RF) demonstrates superior performance with gaps of 1.535% and 2.782%, indicating relative improvements of 25.6% and 11.0%. MoSES(CaDA) demonstrates consistent improvements over its baseline CaDA across all VRP variants at $N = 50$. For $N = 100$, it shows superiority on 13 out of 16 tasks while maintaining comparable performance on CVRP, OVRPB, and OVRPBL tasks, with marginal decreases of $\leq 0.002$ in costs and $\leq 0.027\%$ in optimality gaps. MoSES(CaDA) reduces average optimality gaps over 16 VRP tasks from 1.715% to 1.515% (11.7% relative improvement) at $N = 50$ and from 2.766% to 2.631% (4.9% relative improvement) at $N = 100$, compared to its baseline CaDA. From the perspective of average optimality gap, MoSES(CaDA) is more preferred than MoSES(RF). In terms of the average total time overhead over 16 tasks, MoSES(RF) requires 5.8s and 20.8s for $N = 50$ and $N = 100$, respectively, compared to 2.0s and 8.4s consumed by RF-TE. MoSES(CaDA) requires 7.4s compared to CaDA's 2.3s for $N = 50$, and 24.8s compared to CaDA's 10.5s for $N = 100$. Since this time overhead represents the total time for 1K instances averaged over 16 tasks, the amortized time per instance remains at the microsecond level for both MoSES(RF) and MoSES(CaDA). Thus, the slightly more computational time of our method, resulting from the layer-wise token-wise routing mechanisms and dynamic aggregation operations, is acceptable given the favorable empirical improvements over prior methods, and may not be a major concern in practical applications.

## 5.2 Hyperparameter Studies

We also evaluate the impact of LoRA ranks on model performance, using MoSES(CaDA) as a case study across both $N = 50$ and $N = 100$ problem scales. We allow the ranks for frozen modules $\{B_i A_i\}_{i=1}^{4}$ and trainable module $\hat{B}\hat{A}$ to differ, denoted as $r_{\text{frozen}}$ and $r_{\text{free}}$ respectively. Figures 2(c) 2(d) present the impact of LoRA ranks on the average optimality gap over 16 VRPs, where blue curves fix $r_{\text{frozen}} = 32$ while varying $r_{\text{free}}$ from 4 to 32, and green curves fix $r_{\text{free}} = 32$ while varying $r_{\text{frozen}}$ from 4 to 32. Figure 2(c) reveals that reducing the LoRA rank of trainable module $\hat{B}\hat{A}$ with fixed $r_{\text{frozen}} = 32$ incurs smaller performance degradation than reducing the LoRA rank of frozen modules $\{B_i A_i\}_{i=1}^{4}$ with fixed $r_{\text{free}} = 32$. It suggests that the frozen LoRA experts contribute more significantly to MoSES(CaDA) than the trainable LoRA expert at $N = 50$.

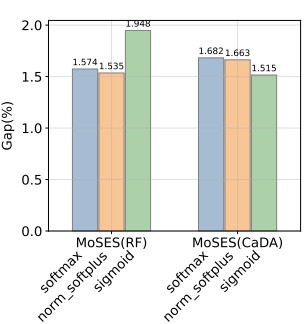

Figure 3: Activations.

$N = 50$. Figure 2(d) demonstrates an inverse relationship at the scale of $N = 100$ that the trainable LoRA expert contributes more significantly to MoSES(CaDA) than the frozen LoRA experts.

Before delving into the in-depth analysis of Figure 2(c) 2(d), we first clarify the respective roles of the gating function and the trainable LoRA expert in MoSES. The gating function is primarily designed to extract individual insights from basis VRP solvers by identifying which solvers offer the most relevant experience for a given problem instance. In contrast, the trainable LoRA expert is intended to capture a holistic understanding of the given VRP variant, which naturally includes correlations among the basis VRP variants. From Figure 2(c), we observe that for smaller problem instances ($N = 50$), reducing the LoRA rank of the frozen LoRA experts leads to a more significant performance drop than reducing the LoRA rank of the trainable LoRA expert. This suggests that for simpler problems, the basis solvers already contain sufficient useful insights, and the unified solver relies less on the holistic understanding provided by the trainable LoRA expert. Conversely, Figure 2(d) shows that for larger problem instances ($N = 100$), reducing the LoRA rank of the trainable LoRA expert results in a greater performance drop than reducing that of the frozen experts. This indicates that for more complex problems, the unified solver requires a deeper and more holistic understanding of the VRP variant, which the trainable LoRA expert is better suited to provide.

### 5.3 Ablation Studies

We evaluate our proposed Gated-LoRA against standard LoRA that varies $\beta$ from 0.1 to 1.0, both used in the phase of fine-tuning specialized experts. Figures 2(a) 2(b) demonstrate Gated-LoRA's consistent superiority by showing lower average optimality gaps over four basis variants (OVRP, VRPB, VRPL, VRPTW) at $N = 50$ for both RF-based and CaDA-based backbones. Figure 3 uncovers that MoSES(RF) prefers norm_softplus($\cdot$) as the activation function in the gating mechanism, while MoSES(CaDA) prefers sigmoid($\cdot$). CaDA-based basis solvers may exhibit stronger generalization capability than RF-based ones, owing to the two parallel Transformer blocks in each layer. Consequently, MoSES(CaDA), which uses sigmoid($\cdot$) as its activation function, tends to prioritize relevant solvers while assigning moderately higher scores to less relevant ones. This behavior suggests that MoSES(CaDA) recognizes that even task-irrelevant solvers can

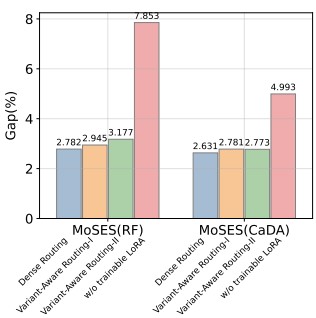

Figure 4: Routing Methods.

contribute useful insights for solving the current VRP variant. Figure 4 shows that dense routing method achieves best in both MoSES(RF) and MoSES(CaDA), and significant performance degradation occurs when the trainable LoRA module $\hat{B}\hat{A}$ is ablated. The dense routing strategy in the gating mechanism does not hinder interpretability, as evidenced by Figure 13 of the Appendix B. Figure 13 presents a behavioral analysis of the gating functions used in our models across 16 VRP variants. In this figure, we plot the scores assigned by the gating function to each basis solver. It is clear that for a given VRP variant, the gating mechanism tends to assign higher scores to basis solvers associated with the corresponding underlying basis VRPs. The reason the gating function does not completely zero out the remaining tasks is that many VRP variants are inherently correlated and solving instances of one task may benefit from insights learned from others. Since our method reuses only the basis VRP solvers, the number of solvers increases linearly, which is more manageable compared to the exponential growth in neural solver.

## 6 Conclusions and Limitations

In this paper, our objective is to design a unified neural solver which is capable of handling multiple VRP variants simultaneously. To achieve this, we propose the State-Decomposable MDP (SDMDP) to reformulate multi-task VRPs, grounded in observation that each VRP variant derives from a shared set of basis VRP variants. Then, the LS-SDMDP extension is developed to reuse basis neural solvers, each specialized for a basis VRP, in the latent space. We finally implement mixture-of-LoRA-experts as the unified solver. While our method demonstrates empirical superiority over prior approaches, it incurs mild computational overhead compared to other neural solvers due to its finer-grained layer-wise and token-wise aggregation and adaptive routing mechanisms. This limitation suggests designing more efficient aggregation techniques which preserve decent performance as future research directions. In addition, extending this framework to broader and general decision-making settings is also appealing.

## Acknowledgments and Disclosure of Funding

This research is supported by a generous research grant from Xiaoi Robot Technology Limited, and the Singapore Ministry of Education (MOE) Academic Research Fund (AcRF) Tier 1 grant. We appreciate the anonymous reviewers, (S)ACs, and PCs of NeurIPS 2025 for their insightful comments to further improve our paper and their service to the community.

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

# A Environment Details

## A.1 VRP Variants

We adopt the same environment configurations for Vehicle Routing Problems (VRPs) as those described in RouteFinder [3]. In this setting, all VRP variants extend the Capacitated VRP (CVRP) by incorporating arbitrary combinations of the following four constraints: 1) *Open Route (O):* A binary variable $o$ indicates whether the vehicle needs to return to depot ($o = 0$) or not ($o = 1$); 2) *Backhaul (B):* Customers $\{v_j\}_{j=1}^{N}$ are categorized into linehauls with $q_j^{\mathrm{LH}} > 0$ or backhauls with $q_j^{\mathrm{BH}} < 0$, where linehauls require deliveries and backhauls require pickups. These are mutually exclusive: $q_j^{\mathrm{LH}} q_j^{\mathrm{BH}} = 0$ for all $j \in \{1, \ldots, N\}$. VRPs with backhaul allow traversing both types in a mixed manner, but linehauls must precede backhauls in each subtour [62]. In VRPs without backhaul, only linehaul customers are present; 3) *Duration Limit (L):* The cost of each subtour within a solution cannot exceed a threshold $l^{\mathrm{dur}}$; 4) *Time Window (TW):* Each node $v_i$ ($i \in \{0, \ldots, N\}$) is associated with a time window $[w_i^{\mathrm{beg}}, w_i^{\mathrm{end}}]$ and a service duration $w_i^{\mathrm{dur}}$, requiring that service must begin within the specified window. The vehicle arriving prior to $w_i^{\mathrm{beg}}$ must wait until the window opens, after which the node is occupied for exactly $w_i^{\mathrm{dur}}$ time units. Furthermore, all vehicles are constrained to return to the depot no later than $w_0^{\mathrm{end}}$. CVRP serves as a basis VRP variant, from which the remaining four basis VRP variants emerge by individually incorporating one additional constraint: CVRP with open routes (OVRP), CVRP with Backhauls (VRPB), CVRP with Duration Limits (VRPL), and CVRP with Time Windows (VRPTW). The complete combinatorial space of these four constraints results in 16 VRP variants ($2^4$ possible combinations): CVRP, OVRP, VRPB, VRPL, VRPTW, OVRPTW, VRPBL, VRPBLTW, VRPBTW, VRPLTW, OVRPB, OVRPBL, OVRPBLTW, OVRPBTW, OVRPL, OVRPLTW.

## A.2 Problem Instance Generation

In this section, we present the procedures for creating the problem instances of various VRP variants for both the training and testing phases. As each VRP variant stems from a common set of attributes, we proceed to outline the generation process for each attribute directly.

**Locations.** In each problem instance, $N + 1$ coordinates are uniformly sampled from a unit square, denoted as $\{(x_0, y_0), (x_1, y_1), \ldots, (x_N, y_N)\}$, where $x_i, y_i \sim U(0, 1)$ for $i = 0, \ldots, N$. $(x_0, y_0)$ represent the depot coordinates, while $\{(x_i, y_i)\}_{i=1}^{N}$ denote the coordinates of the customers.

**Vehicle Capacity.** The vehicle capacity $Q$ remains constant for all vehicles within each problem instance and is calculated as follows:

$$Q = \begin{cases} 30 + \lfloor \frac{1000}{5} + \frac{N - 1000}{33.3} \rfloor & \text{if } N > 1000 \\ 30 + \lfloor \frac{N}{5} \rfloor & \text{if } 20 < N \leq 1000 \\ 30 & \text{otherwise} \end{cases} \tag{5}$$

**Linehaul and Backhaul Demands.** For the depot node $v_0$, both linehaul and backhaul demands are set to zero. For each customer $v_j$ ($j \geq 1$), the linehaul demand $q_j^{\mathrm{LH}}$ is drawn uniformly at random from the integer set $\{1, 2, \ldots, 9\}$, while the backhaul demand $q_j^{\mathrm{BH}}$ is sampled uniformly from $\{-1, -2, \ldots, -9\}$. A binary decision variable $\hat{q}_j \in \{0, 1\}$ is then introduced, with probabilities $\mathbb{P}(\hat{q}_j = 0) = 0.8$ and $\mathbb{P}(\hat{q}_j = 1) = 0.2$. If $\hat{q}_j = 0$, the customer is designated as a linehaul, retaining the sampled value of $q_j^{\mathrm{LH}}$ while setting $q_j^{\mathrm{BH}} = 0$. Conversely, if $\hat{q}_j = 1$, the customer is treated as a backhaul, preserving $q_j^{\mathrm{BH}}$ and setting $q_j^{\mathrm{LH}} = 0$. It is important to note that both linehaul and backhaul demands are present only in VRP variants that incorporate backhauls. In all other VRP variants, only linehaul demands are considered.

**Open Routes.** A binary variable $o \in \{0, 1\}$ is introduced to indicate whether the vehicle is required to return to the depot. The proportion of VRP instances with open routes is governed by the probability $\mathbb{P}(o = 1)$, where $o = 1$ signifies that the open route constraint is active.

**Time Windows.** For the depot node $v_0$, the time window is defined as $[w_0^{\mathrm{beg}}, w_0^{\mathrm{end}}]$, where $w_0^{\mathrm{beg}} = 0$ and $w_0^{\mathrm{end}}$ represents the system's end time. The service duration at the depot, denoted by $w_0^{\mathrm{dur}}$, is set

to zero. For each customer $v_i$ ($i \in \{1, \ldots, N\}$), a time window $[w_i^{\text{beg}}, w_i^{\text{end}}]$ and a service duration $w_i^{\text{dur}}$ require to be generated. The service duration $w_i^{\text{dur}}$ is sampled uniformly from the interval $[I_1, I_2]$. The length of the time window, defined as $w_i^{\text{len}} = w_i^{\text{end}} - w_i^{\text{beg}}$, is drawn uniformly from the interval $[I_2, I_3]$. Let $\text{Dist}(v_0, v_i)$ denote the distance between the depot $v_0$ and customer $v_i$. The upper bound $w_i^{\text{UB}}$ for the start time of the time window is computed as:

$$w_i^{\text{UB}} = \frac{w_0^{\text{end}} - w_i^{\text{dur}} - w_i^{\text{len}}}{\text{Dist}(v_0, v_i)} - 1. \tag{6}$$

The start time of the time window is then determined by:

$$w_i^{\text{beg}} = (1 + (w_i^{\text{UB}} - 1) \cdot u_i) \cdot \text{Dist}(v_0, v_i); \tag{7}$$

where $u_i \sim U(0, 1)$ is a uniformly distributed random variable. Finally, the end time of the time window is given by:

$$w_i^{\text{end}} = w_i^{\text{beg}} + w_i^{\text{len}}. \tag{8}$$

Additionally, if the time window constraint is not active in a given problem instance, for all nodes $v_i$ ($i \geq 0$), the start time $w_i^{\text{beg}}$ is set to zero, the end time $w_i^{\text{end}}$ is set to $\infty$, and the service duration $w_i^{\text{dur}}$ is set to zero.

**Distance Limit.** The distance limit $l^{\text{dur}}$ is sampled from the uniform distribution $U(2 \cdot \max_i(\text{Dist}(v_0, v_i)), l_{\text{max}}^{\text{dur}})$, where $l_{\text{max}}^{\text{dur}}$ is a predefined upper bound. This sampling strategy ensures that every customer node is reachable from the depot. If the distance limit constraint is not enforced, the distance limit $l^{\text{dur}}$ is set to $\infty$.

### A.3 Feasible Action Space

To derive the feasible action space in the environment, we need a action mask based on the combined constraints of basis VRP variants. In each time step, we use the following feasibility testing procedures to mask out infeasible actions from the action space. Please note that the vehicle speed is fixed at 1.0, and the time windows are normalized accordingly. As a result, the travel time between any two nodes is numerically equivalent to the Euclidean distance between them.

1) Each customer needs to be visited exactly once. If the vehicle is currently located at the depot and there are still unserved customers, selecting the depot as the next action is not permitted.

2) The vehicle must arrive at node $v_j$ from node $v_i$ before the end of the service time window: $w_t^{\text{cur}} + \text{Dist}(v_i, v_j) \leq w_j^{\text{end}}$, where $w_t^{\text{cur}}$ is the current time.

3) The cost of a subtour, when traveling from node $v_i$ to node $v_j$, must not exceed the predefined distance limit: $l_t^{\text{cur}} + \text{Dist}(v_i, v_j) \leq l^{\text{dur}}$, where $l_t^{\text{cur}}$ denotes the cumulative distance traveled along the current subtour.

4) If the vehicle is required to return to the depot and intends to visit node $v_j$ from its current location $v_i$, it must satisfy both time and distance feasibility conditions: the vehicle must be able to complete the visit and return to the depot before the system's end time: $\max(w_t^{\text{cur}} + \text{Dist}(v_i, v_j), w_j^{\text{beg}}) + w_j^{\text{dur}} + \text{Dist}(v_0, v_j) < w_0^{\text{end}}$, and the total length of the resulting subtour, including travel from $v_i$ to $v_j$ and from $v_j$ back to the depot, must not exceed the distance limit: $l_t^{\text{cur}} + \text{Dist}(v_i, v_j) + \text{Dist}(v_0, v_j) < l^{\text{dur}}$.

5) If the vehicle intends to visit a linehaul customer $v_j$, the corresponding linehaul demand $q_j^{\text{LH}}$ must not exceed the remaining linehaul capacity $Q_t^{\text{LH}}$. Similarly, if the vehicle intends to visit a backhaul customer $v_j$, the backhaul demand $q_j^{\text{BH}}$ must not exceed the remaining backhaul capacity $Q_t^{\text{BH}}$. Additionally, all linehaul customers must be visited before any backhaul customers within each subtour.

### A.4 VRP Formulation

To illustrate the VRP formulation within the SDMDP framework, we provide an example for clarification. Under the SDMDP framework, there are five types of basis state spaces: $\mathcal{S}^{(0)}, \ldots, \mathcal{S}^{(4)}$. Specifically, $\mathcal{S}^{(0)}$ encodes node coordinates, linehaul demands, and remaining linehaul capacity, $\mathcal{S}^{(1)}$

Table 2: Performance comparison of multi-task VRP solvers on OOD CVRP instances.

| Vehicle Capacity | 30 | | 50(**ID**) | | 70 | | 90 | | 110 | | 130 | | 150 | | 200 | |
|---|---|---|---|---|---|---|---|---|---|---|---|---|---|---|---|---|
| | Cost | Time | Cost | Time | Cost | Time | Cost | Time | Cost | Time | Cost | Time | Cost | Time | Cost | Time |
| MTPOMO | 23.415 | 9s | 15.933 | 8s | 13.144 | 8s | 11.684 | 8s | 10.835 | 8s | 10.359 | 8s | 10.027 | 8s | 9.566 | 8s |
| MVMoE | 23.162 | 12s | 15.888 | 11s | 13.068 | 11s | 11.534 | 11s | 10.616 | 11s | 10.089 | 11s | 9.727 | 11s | 9.203 | 11s |
| RF-MoE | 23.251 | 12s | 15.877 | 11s | 13.080 | 11s | 11.580 | 11s | 10.685 | 11s | 10.170 | 11s | 9.802 | 11s | 9.294 | 11s |
| RF-POMO | 23.229 | 9s | 15.908 | 8s | 13.104 | 8s | 11.607 | 8s | 10.715 | 8s | 10.200 | 8s | 9.846 | 8s | 9.362 | 8s |
| RF-TE | 23.085 | 9s | 15.857 | 8s | 13.023 | 8s | 11.466 | 8s | 10.521 | 8s | 9.956 | 8s | 9.582 | 8s | 9.003 | 8s |
| MoSES(RF)$_{\text{Sftm}}$ | 23.058 | 22s | 15.826 | 21s | 12.999 | 21s | 11.443 | 20s | **10.485** | 20s | **9.922** | 20s | **9.550** | 20s | **8.968** | 20s |
| MoSES(RF)$_{\text{Sftp}}$ | **23.035** | 22s | **15.808** | 21s | **12.978** | 20s | **11.431** | 20s | 10.504 | 20s | 10.107 | 20s | 10.188 | 20s | 10.271 | 20s |
| MoSES(RF)$_{\text{Sigm}}$ | 23.049 | 21s | 15.839 | 20s | 13.009 | 19s | 11.447 | 19s | 10.494 | 19s | 9.922 | 19s | 9.550 | 19s | 8.968 | 19s |
| CaDA | 23.083 | 10s | 15.831 | 10s | 12.999 | 10s | 11.438 | 9s | 10.479 | 9s | 9.906 | 9s | 9.532 | 9s | 8.942 | 9s |
| MoSES(CaDA)$_{\text{Sftm}}$ | 23.066 | 28s | 15.840 | 26s | 13.015 | 26s | 11.457 | 25s | 10.513 | 25s | 9.950 | 25s | 9.589 | 25s | 9.025 | 25s |
| MoSES(CaDA)$_{\text{Sftp}}$ | **23.031** | 27s | 15.836 | 26s | 13.005 | 26s | 11.441 | 25s | 10.482 | 25s | 9.906 | 25s | 9.537 | 25s | 8.960 | 25s |
| MoSES(CaDA)$_{\text{Sigm}}$ | 23.047 | 25s | 15.833 | 24s | **12.999** | 24s | **11.437** | 24s | **10.473** | 24s | **9.897** | 23s | **9.526** | 23s | **8.937** | 23s |

represents a binary variable indicating whether the vehicle needs to return to the depot; $\mathcal{S}^{(2)}$ captures backhaul demands and remaining backhaul capacity; $\mathcal{S}^{(3)}$ includes the duration limit and the current traveled length of the subtour; and $\mathcal{S}^{(4)}$ contains time windows, service times, and the current time. Before each episode begins, the initial state space is constructed by selecting $\mathcal{S}^{(0)}$ and any subset of the remaining basis states. In this example, we choose $\mathcal{S}^{(0)}$, $\mathcal{S}^{(3)}$ and $\mathcal{S}^{(4)}$, which correspond to the VRP with limited duration and time windows (VRPLTW). At each time step, the policy observes the basis states $s^{(0)}$, $s^{(3)}$, and $s^{(4)}$, and a masking mechanism is applied to filter out infeasible nodes from the set of unvisited nodes, specifically, those whose demands exceed the remaining linehaul capacity, those that would violate the duration limit, and those whose arrival time would exceed the end of their time window, from unvisited nodes. This results in a list of feasible nodes from which the next node is selected. Upon selection, the linehaul capacity, traveled length, and current time are updated accordingly, and the selected node is masked out for subsequent steps. Once a subtour is completed, the linehaul capacity is reset to its default value, the traveled length is reset to zero, and the current time is also reset to zero. This reset reflects the assumption in VRP that multiple vehicles can begin their routes concurrently.

# B  Experiments

## B.1  Out-of-distribution Attribute Generalization

In this section, we evaluate the out-of-distribution (OOD) generalization capabilities of our proposed methods, MoSES(RF) and MoSES(CaDA), in comparison with existing unified multi-task neural solvers. The evaluation specifically targets generalization to unseen attribute values pertaining to vehicle capacities, time windows, and distance limits. For each evaluation setting, we construct a testing dataset consisting of 1,000 problem instances, each with $N = 100$ nodes. Performance is measured using two metrics: the average cost across the 1,000 problem instances, and the total computational time required to solve all instances, with lower values indicating better performance. Notably, all neural solvers evaluated were trained on problem instances with the same number of nodes (i.e., $N = 100$).

In CVRP, each neural solver is trained on problem instances with a fixed vehicle capacity of $Q = 50$. To investigate OOD generalization to unseen vehicle capacities, we generate a separate testing dataset for each capacity value in the set $\{30, 50, 70, 90, 110, 130, 150, 200\}$. The corresponding results are presented in Table 2. Please note that $Q = 50$ corresponds to the in-distribution (ID) evaluation setting. Both MoSES(RF) and MoSES(CaDA) adopt the dense routing strategy. We use Sftm, Sftp and Sigm to denote the activation functions $\text{softmax}(\cdot)$, $\text{norm\_softplus}(\cdot)$ and $\text{sigmoid}(\cdot)$, respectively. For MoSES(RF), we observe MoSES(RF)$_{\text{Sftm}}$ consistently outperforms its baseline, RF-TE, in terms of generalization performance on OOD vehicle capacities. However, MoSES(RF)$_{\text{Sftp}}$ demonstrates superior performance on small-scale problem instances ($N \leq 90$), compared to MoSES(RF)$_{\text{Sftm}}$. MoSES(CaDA)$_{\text{Sigm}}$ demonstrates stronger OOD generalization than its baseline, CaDA, across all evaluation settings, with the exception of a marginal performance drop observed in the ID case at $Q = 50$. Furthermore, MoSES(CaDA)$_{\text{Sigm}}$ outperforms MoSES(RF)$_{\text{Sftm}}$ across the majority of evaluation scenarios.

In VRPL, the predefined upper bound on the distance limit, denoted as $l_{\max}^{\text{dur}}$, is set to 2.8 (approximately $2\sqrt{2}$) during training, which serves as the ID evaluation setting. To examine OOD

Table 3: Performance comparison of multi-task VRP solvers on OOD VRPL instances.

| Distance Limit | 2.7 | | 2.8(**ID**) | | 2.9 | | 3.0 | | 3.1 | | 3.2 | | 3.3 | | 3.4 | |
|---|---|---|---|---|---|---|---|---|---|---|---|---|---|---|---|---|
| | Cost | Time | Cost | Time | Cost | Time | Cost | Time | Cost | Time | Cost | Time | Cost | Time | Cost | Time |
| MTPOMO | 16.193 | 9s | 16.151 | 9s | 16.121 | 9s | 16.105 | 9s | 16.090 | 9s | 16.078 | 9s | 16.069 | 9s | 16.060 | 9s |
| MVMoE | 16.142 | 12s | 16.099 | 12s | 16.073 | 12s | 16.054 | 12s | 16.040 | 12s | 16.030 | 12s | 16.017 | 12s | 16.010 | 12s |
| RF-MoE | 16.109 | 12s | 16.070 | 12s | 16.046 | 12s | 16.032 | 12s | 16.017 | 12s | 16.005 | 12s | 15.997 | 12s | 15.991 | 12s |
| RF-POMO | 16.152 | 9s | 16.106 | 9s | 16.079 | 9s | 16.063 | 9s | 16.049 | 9s | 16.041 | 9s | 16.029 | 9s | 16.019 | 9s |
| RF-TE | 16.091 | 9s | 16.051 | 9s | 16.029 | 9s | 16.012 | 9s | 15.997 | 9s | 15.987 | 9s | 15.975 | 9s | 15.969 | 9s |
| CaDA | 16.067 | 10s | 16.026 | 10s | 16.002 | 10s | 15.983 | 10s | 15.971 | 10s | 15.960 | 10s | 15.952 | 10s | 15.946 | 10s |
| MoSES(RF) | **16.045** | 21s | **16.005** | 21s | **15.982** | 21s | **15.966** | 21s | **15.952** | 21s | **15.940** | 21s | **15.928** | 21s | **15.923** | 21s |
| MoSES(CaDA) | 16.070 | 25s | 16.024 | 25s | 16.002 | 25s | 15.984 | 25s | 15.971 | 25s | 15.961 | 25s | 15.947 | 25s | 15.942 | 25s |

Table 4: Performance comparison of multi-task VRP solvers on OOD VRPTW instances.

| Time Window | [0.05, 0.08, 0.10] | | [0.15, 0.18, 0.20](**ID**) | | [0.25, 0.28, 0.30] | | [0.35, 0.38, 0.40] | | [0.45, 0.48, 0.50] | |
|---|---|---|---|---|---|---|---|---|---|---|
| | Cost | Time | Cost | Time | Cost | Time | Cost | Time | Cost | Time |
| MTPOMO | 25.549 | 9s | 26.410 | 9s | 28.294 | 9s | 31.339 | 10s | 35.217 | 10s |
| MVMoE | 25.490 | 12s | 26.391 | 12s | 28.258 | 12s | 31.237 | 13s | 35.041 | 13s |
| RF-MoE | 25.459 | 12s | 26.319 | 12s | 28.227 | 12s | 31.338 | 13s | 35.251 | 13s |
| RF-POMO | 25.492 | 9s | 26.335 | 9s | 28.242 | 9s | 31.408 | 10s | 35.315 | 10s |
| RF-TE | 25.377 | 9s | 26.234 | 9s | 28.156 | 9s | 31.268 | 9s | 35.135 | 10s |
| CaDA | 25.309 | 11s | 26.128 | 10s | 28.095 | 11s | 31.334 | 11s | 35.329 | 11s |
| MoSES(RF) | 25.271 | 22s | 26.143 | 22s | **28.027** | 23s | **30.943** | 24s | **34.500** | 24s |
| MoSES(CaDA) | **25.255** | 26s | **26.032** | 26s | 28.054 | 26s | 31.354 | 27s | 35.223 | 28s |

| Time Window | [0.55, 0.58, 0.60] | | [0.65, 0.68, 0.70] | | [0.75, 0.78, 0.80] | | [0.85, 0.88, 0.90] | | [0.95, 0.98, 1.00] | |
|---|---|---|---|---|---|---|---|---|---|---|
| | Cost | Time | Cost | Time | Cost | Time | Cost | Time | Cost | Time |
| MTPOMO | 39.549 | 10s | 44.035 | 11s | 48.215 | 11s | 52.295 | 11s | 55.933 | 11s |
| MVMoE | 39.349 | 14s | 43.781 | 14s | 47.995 | 15s | 52.160 | 16s | 55.903 | 16s |
| RF-MoE | 39.606 | 14s | 44.026 | 14s | 48.082 | 15s | 52.029 | 15s | 55.465 | 15s |
| RF-POMO | 39.668 | 10s | 44.102 | 11s | 48.144 | 11s | 52.090 | 11s | 55.539 | 11s |
| RF-TE | 39.402 | 10s | 43.783 | 10s | 47.812 | 11s | 51.869 | 11s | 55.424 | 11s |
| CaDA | 39.697 | 12s | 44.152 | 12s | 48.231 | 12s | 52.122 | 13s | 55.650 | 13s |
| MoSES(RF) | **38.490** | 25s | **42.600** | 26s | **46.770** | 27s | **51.475** | 28s | 55.973 | 29s |
| MoSES(CaDA) | 39.488 | 29s | 43.864 | 29s | 47.854 | 30s | 51.784 | 30s | **55.172** | 31s |

generalization to unseen distance limits, we evaluate performance across a range of values for $l_{\max}^{\mathrm{dur}}$ from the set $\{2.7, 2.8, 2.9, 3.0, 3.1, 3.2, 3.3, 3.4\}$, where $l_{\max}^{\mathrm{dur}} = 2.8$ corresponds to the ID case. The results of this evaluation are summarized in Table 3. MoSES(RF) employs the dense routing strategy with the $\mathrm{norm\_softplus}(\cdot)$ activation function, while MoSES(CaDA) utilizes the same routing strategy with $\mathrm{sigmid}(\cdot)$ as the activation function. Experimental results indicate that MoSES(RF) consistently outperforms all other methods, including MoSES(CaDA), across all evaluation settings. This demonstrates its superior OOD generalization capability when faced with unseen distance limits.

In VRPTW, each problem instance is primarily defined by both the time window and the service duration. These are generated based on two intervals $[I_1, I_2]$ and $I_2, I_3$ as described in Section A.2. Let the triplet $[I_1, I_2, I_3]$ represent the full configurations. The setting $[0.15, 0.18, 0.20]$ serves as the in-distribution ID evaluation. To assess OOD generalization to unseen time window configurations, we consider a range of settings from the set $\{[0.05, 0.08, 0.10], [0.15, 0.18, 0.20], [0.25, 0.28, 0.30], \ldots, [0.95, 0.98, 1.00]\}$, as reported in Table 4. Both MoSES(RF) and MoSES(CaDA) adopt the dense routing strategy. MoSES(RF) uses $\mathrm{norm\_softplus}(\cdot)$ as the activation function, while MoSES(CaDA) employs $\mathrm{sigmoid}(\cdot)$. Experimental results show that MoSES(RF) generally outperforms its baseline, RF-TE, across most evaluation settings, with the exception of a performance drop observed at $[0.95, 0.98, 1.00]$. Similarly, MoSES(CaDA) surpasses its baseline, CaDA, except for a marginal decline in performance at $[0.35, 0.38, 0.40]$. Overall, MoSES(RF) demonstrates superior performance compared to MoSES(CaDA) in the majority of task settings.

## B.2 CVRPLIB Evaluation

We report the performance comparison of our proposed methods, MoSES(RF) and MoSES(CaDA), against baseline approaches on CVRPLIB instances from the X set, which includes problem sizes ranging from 101 to at most 1,001 nodes, as done in [83, 3]. Both MoSES(RF) and MoSES(CaDA) use the Variant-Aware Routing-I strategy. MoSES(RF) uses $\mathrm{norm\_softplus}(\cdot)$ as the activation function, while MoSES(CaDA) employs $\mathrm{sigmoid}(\cdot)$. The detailed results are presented in Table 5.

Table 5: Performance comparison of multi-task VRP solvers on CVRPLIB instances from the X set.

| Set-X | | MTPOMO | | | MVMoE | | | RF-MoE | | | RF-POMO | | | RF-TE | | | CaDA | | | MoSES(RF) | | | MoSES(CaDA) | | |
|---|---|---|---|---|---|---|---|---|---|---|---|---|---|---|---|---|---|---|---|---|---|---|---|---|---|
| Instance | Opt. | Cost | Gap | Time | Cost | Gap | Time | Cost | Gap | Time | Cost | Gap | Time | Cost | Gap | Time | Cost | Gap | Time | Cost | Gap | Time | Cost | Gap | Time |
| X-n101-k25 | 27591 | 29470 | 6.810% | 0.4s | 29076 | 5.382% | 0.5s | 28934 | 4.868% | 0.5s | 29090 | 5.433% | 0.3s | 29048 | 5.281% | 0.4s | 28944 | 4.904% | 0.5s | 28895 | 4.726% | 0.8s | 29110 | 5.505% | 0.8s |
| X-n106-k14 | 26362 | 28029 | 6.323% | 0.3s | 27443 | 4.101% | 0.5s | 27292 | 3.528% | 0.6s | 27378 | 3.854% | 0.3s | 27159 | 3.023% | 0.4s | 27042 | 2.579% | 0.3s | 27205 | 3.198% | 0.7s | 27051 | 2.614% | 0.8s |
| X-n110-k13 | 14971 | 15100 | 0.862% | 0.3s | 15327 | 2.378% | 0.5s | 15260 | 1.930% | 0.5s | 15519 | 3.660% | 0.3s | 15314 | 2.291% | 0.4s | 15229 | 1.723% | 0.3s | 15242 | 1.810% | 0.8s | 15332 | 2.411% | 0.8s |
| X-n115-k10 | 12747 | 13433 | 5.382% | 0.4s | 13475 | 5.711% | 0.6s | 13638 | 6.990% | 0.5s | 13263 | 4.048% | 0.3s | 13338 | 4.636% | 0.4s | 13060 | 2.455% | 0.4s | 13313 | 4.440% | 0.8s | 13085 | 2.652% | 0.8s |
| X-n120-k6 | 13332 | 14051 | 5.393% | 0.3s | 13782 | 3.375% | 0.6s | 13908 | 4.320% | 0.6s | 14061 | 5.468% | 0.4s | 13765 | 3.248% | 0.4s | 13678 | 2.595% | 0.3s | 13781 | 3.368% | 0.8s | 13619 | 2.153% | 0.9s |
| X-n125-k30 | 55539 | 59015 | 6.259% | 0.4s | 58200 | 4.791% | 0.7s | 58587 | 5.488% | 0.6s | 58770 | 5.818% | 0.4s | 58570 | 5.457% | 0.4s | 57748 | 3.977% | 0.4s | 58220 | 4.827% | 0.9s | 57620 | 3.747% | 1.0s |
| X-n129-k18 | 28940 | 30176 | 4.271% | 0.4s | 29334 | 1.361% | 0.6s | 30039 | 3.798% | 0.6s | 29645 | 2.436% | 0.4s | 29457 | 1.786% | 0.5s | 29500 | 1.935% | 0.4s | 29558 | 2.135% | 0.9s | 29620 | 2.350% | 0.9s |
| X-n134-k13 | 10916 | 11707 | 7.246% | 0.4s | 11462 | 5.002% | 0.6s | 11439 | 4.791% | 0.6s | 11463 | 5.011% | 0.4s | 11624 | 6.486% | 0.4s | 11652 | 6.742% | 0.4s | 11584 | 6.119% | 0.8s | 11573 | 6.019% | 1.0s |
| X-n139-k10 | 13590 | 14058 | 3.444% | 0.4s | 14099 | 3.745% | 0.6s | 13917 | 2.406% | 0.6s | 13945 | 2.612% | 0.4s | 13812 | 1.634% | 0.3s | 13940 | 2.575% | 0.4s | 13908 | 2.340% | 0.8s | 13877 | 2.112% | 0.9s |
| X-n143-k7 | 15700 | 16626 | 5.898% | 0.4s | 16349 | 4.134% | 0.6s | 16655 | 6.083% | 0.6s | 16603 | 5.752% | 0.5s | 16257 | 3.548% | 0.4s | 16189 | 3.115% | 0.4s | 16024 | 2.064% | 0.9s | 15980 | 1.783% | 0.9s |
| X-n148-k46 | 43448 | 46648 | 7.365% | 0.5s | 45893 | 5.627% | 0.6s | 46542 | 7.121% | 0.8s | 46082 | 6.062% | 0.5s | 45026 | 3.632% | 0.6s | 45606 | 4.967% | 0.6s | 45408 | 4.511% | 1.0s | 45600 | 4.953% | 1.0s |
| X-n153-k22 | 21220 | 23514 | 10.811% | 0.5s | 23661 | 11.503% | 0.7s | 23906 | 12.658% | 0.7s | 22991 | 8.346% | 0.5s | 23478 | 10.641% | 0.6s | 23142 | 9.057% | 0.5s | 23347 | 10.024% | 0.9s | 23310 | 9.849% | 1.0s |
| X-n157-k13 | 16876 | 17922 | 6.198% | 0.5s | 17439 | 3.336% | 0.7s | 17801 | 5.481% | 0.8s | 17536 | 3.911% | 0.5s | 17315 | 2.601% | 0.5s | 17295 | 2.483% | 0.5s | 17227 | 2.080% | 1.0s | 17317 | 2.613% | 1.0s |
| X-n162-k11 | 14138 | 14616 | 3.381% | 0.5s | 14705 | 4.010% | 0.7s | 14524 | 2.730% | 0.7s | 14663 | 3.713% | 0.5s | 14664 | 3.720% | 0.5s | 14704 | 4.003% | 0.5s | 14683 | 3.855% | 1.0s | 14677 | 3.812% | 1.0s |
| X-n167-k10 | 20557 | 21662 | 5.375% | 0.5s | 21504 | 4.607% | 0.7s | 21481 | 4.495% | 0.7s | 21410 | 4.149% | 0.5s | 21425 | 4.222% | 0.5s | 21078 | 2.534% | 0.5s | 21368 | 3.945% | 1.0s | 21384 | 4.023% | 1.0s |
| X-n172-k51 | 45607 | 48960 | 7.352% | 0.6s | 47883 | 4.990% | 0.9s | 49726 | 9.032% | 1.0s | 48412 | 6.150% | 0.6s | 48162 | 5.602% | 0.7s | 48198 | 5.681% | 0.6s | 48136 | 5.545% | 1.0s | 48145 | 5.565% | 1.0s |
| X-n176-k26 | 47812 | 51989 | 8.736% | 0.5s | 52117 | 9.004% | 0.8s | 53626 | 12.160% | 0.9s | 52347 | 9.485% | 0.6s | 51501 | 7.716% | 0.6s | 51120 | 6.919% | 0.6s | 52001 | 8.761% | 1.0s | 51612 | 7.948% | 1.0s |
| X-n181-k23 | 25569 | 26572 | 3.923% | 0.6s | 26456 | 3.469% | 0.8s | 29154 | 14.021% | 0.9s | 26544 | 3.813% | 0.6s | 26097 | 2.065% | 0.6s | 26262 | 2.710% | 0.6s | 26181 | 2.394% | 1.0s | 26143 | 2.245% | 1.0s |
| X-n186-k15 | 24145 | 25236 | 4.519% | 0.5s | 25151 | 4.166% | 0.8s | 25140 | 4.121% | 0.9s | 25238 | 4.527% | 0.6s | 25153 | 4.175% | 0.6s | 25343 | 4.970% | 0.6s | 25115 | 4.017% | 1.0s | 25246 | 4.560% | 1.0s |
| X-n190-k8 | 16980 | 18369 | 8.180% | 0.5s | 19078 | 12.356% | 0.9s | 18217 | 7.285% | 0.9s | 18696 | 10.106% | 0.6s | 17871 | 5.247% | 0.6s | 17882 | 5.312% | 0.6s | 17929 | 5.589% | 1.0s | 17569 | 3.469% | 1.0s |
| X-n195-k51 | 44225 | 48310 | 9.237% | 0.7s | 46974 | 6.216% | 1.0s | 48965 | 10.718% | 1.0s | 47479 | 7.358% | 0.7s | 47396 | 7.170% | 0.7s | 46723 | 5.648% | 0.7s | 46541 | 5.237% | 1.0s | 47479 | 7.358% | 1.0s |
| X-n200-k36 | 58578 | 62041 | 5.912% | 0.6s | 61627 | 5.205% | 0.9s | 61696 | 5.323% | 0.9s | 61662 | 5.265% | 0.6s | 61139 | 4.372% | 0.7s | 61010 | 4.152% | 0.8s | 61088 | 4.285% | 1.0s | 61089 | 4.287% | 2.0s |
| X-n204-k19 | 19565 | 20652 | 5.556% | 0.6s | 20584 | 5.208% | 0.9s | 20466 | 4.605% | 0.9s | 20730 | 5.955% | 0.6s | 20531 | 4.937% | 0.6s | 20735 | 5.980% | 0.6s | 20620 | 5.392% | 1.0s | 20420 | 4.370% | 1.0s |
| X-n209-k16 | 30656 | 32333 | 5.470% | 0.6s | 32358 | 5.552% | 0.9s | 32145 | 4.857% | 0.9s | 32585 | 6.292% | 0.6s | 31876 | 3.980% | 0.6s | 32184 | 4.984% | 0.6s | 31775 | 3.650% | 1.0s | 32053 | 4.557% | 1.0s |
| X-n214-k11 | 10856 | 11699 | 7.765% | 0.6s | 11597 | 6.826% | 0.9s | 11534 | 6.245% | 0.9s | 11638 | 7.203% | 0.6s | 11668 | 7.480% | 0.6s | 11748 | 8.217% | 0.6s | 11635 | 7.176% | 1.0s | 11716 | 7.922% | 1.0s |
| X-n219-k73 | 117595 | 121980 | 3.729% | 0.8s | 124434 | 5.816% | 1.0s | 121627 | 3.429% | 1.0s | 123500 | 5.021% | 0.9s | 120344 | 2.338% | 0.8s | 120011 | 2.055% | 0.8s | 119497 | 1.617% | 2.0s | 119710 | 1.799% | 2.0s |
| X-n223-k34 | 40437 | 43381 | 7.280% | 0.7s | 42694 | 5.582% | 1.0s | 43097 | 6.578% | 1.0s | 42601 | 5.352% | 0.7s | 42251 | 4.486% | 0.7s | 42273 | 4.540% | 0.7s | 42312 | 4.637% | 1.0s | 42128 | 4.182% | 2.0s |
| X-n228-k23 | 25742 | 28523 | 10.803% | 0.7s | 28033 | 8.900% | 1.0s | 29590 | 14.948% | 1.0s | 28212 | 9.595% | 0.8s | 28699 | 11.487% | 0.8s | 27821 | 8.076% | 0.7s | 27701 | 7.610% | 1.0s | 27724 | 7.699% | 1.0s |
| X-n233-k16 | 19230 | 20644 | 7.353% | 0.7s | 20656 | 7.415% | 1.0s | 20507 | 6.641% | 1.0s | 20427 | 6.225% | 0.7s | 20761 | 7.962% | 0.7s | 20285 | 5.486% | 0.9s | 20552 | 6.875% | 1.0s | 20623 | 7.244% | 1.0s |
| X-n237-k14 | 27042 | 30047 | 11.112% | 0.7s | 29772 | 10.095% | 1.0s | 29514 | 9.141% | 1.0s | 30084 | 11.249% | 0.7s | 29595 | 9.441% | 0.7s | 30282 | 11.981% | 0.7s | 29720 | 9.903% | 1.0s | 29518 | 9.156% | 1.0s |
| X-n242-k48 | 82751 | 88179 | 6.559% | 0.8s | 87497 | 5.735% | 1.0s | 87832 | 6.140% | 1.0s | 87029 | 5.170% | 0.8s | 85704 | 3.569% | 0.9s | 85813 | 3.700% | 0.8s | 85420 | 3.225% | 2.0s | 85643 | 3.495% | 2.0s |
| X-n247-k50 | 37274 | 41610 | 11.633% | 0.8s | 40973 | 9.924% | 1.0s | 43153 | 15.772% | 1.0s | 41120 | 10.318% | 0.8s | 40642 | 9.036% | 0.9s | 39918 | 7.093% | 0.8s | 40131 | 7.665% | 2.0s | 40736 | 9.288% | 2.0s |
| Avg. Gap (N < 251) | | | 6.566% | | | 5.829% | | | 6.754% | | | 5.905% | | | 5.061% | | | 4.772% | | | 4.789% | | | 4.724% | |
| X-n251-k28 | 38684 | 41211 | 6.532% | 0.7s | 41330 | 6.840% | 1.0s | 40691 | 5.188% | 1.0s | 40811 | 5.498% | 0.8s | 40127 | 3.730% | 0.8s | 40359 | 4.330% | 0.8s | 40630 | 5.031% | 2.0s | 40290 | 4.152% | 2.0s |
| X-n256-k16 | 18839 | 20400 | 8.286% | 0.7s | 20559 | 9.130% | 1.0s | 20015 | 6.242% | 1.0s | 20238 | 7.426% | 0.7s | 19994 | 6.131% | 0.8s | 20372 | 8.137% | 0.7s | 20034 | 6.343% | 1.0s | 20068 | 6.524% | 2.0s |
| X-n261-k13 | 26558 | 28741 | 8.220% | 0.7s | 28524 | 7.403% | 1.0s | 28203 | 6.194% | 1.0s | 28527 | 7.350% | 0.8s | 28510 | 7.350% | 0.8s | 28833 | 8.566% | 1.0s | 28447 | 7.113% | 2.0s | 28577 | 7.602% | 2.0s |
| X-n266-k58 | 75478 | 84617 | 12.108% | 0.9s | 82048 | 8.705% | 1.0s | 81135 | 7.495% | 1.0s | 81053 | 7.386% | 0.9s | 79832 | 5.769% | 0.9s | 80115 | 6.144% | 0.9s | 79820 | 5.753% | 2.0s | 80036 | 6.039% | 2.0s |
| X-n270-k35 | 35291 | 38146 | 8.090% | 0.9s | 38333 | 8.620% | 1.0s | 37401 | 5.979% | 1.0s | 38051 | 7.821% | 0.9s | 37382 | 5.925% | 0.9s | 37674 | 6.752% | 0.8s | 37420 | 6.033% | 2.0s | 36923 | 4.624% | 2.0s |
| X-n275-k28 | 21245 | 24688 | 16.206% | 0.8s | 25021 | 17.774% | 1.0s | 25241 | 18.809% | 1.0s | 24321 | 14.479% | 0.8s | 24187 | 13.848% | 0.9s | 24482 | 15.237% | 0.8s | 24292 | 14.342% | 2.0s | 24312 | 14.436% | 2.0s |
| X-n280-k17 | 33503 | 36677 | 9.474% | 0.8s | 36636 | 9.351% | 1.0s | 36538 | 9.059% | 1.0s | 35558 | 6.134% | 0.9s | 36653 | 9.402% | 0.9s | 36081 | 7.695% | 0.8s | 35988 | 7.417% | 2.0s | 35494 | 5.943% | 2.0s |
| X-n284-k15 | 20226 | 22474 | 11.114% | 0.8s | 22583 | 11.653% | 1.0s | 21857 | 8.064% | 1.0s | 21976 | 8.652% | 0.8s | 22154 | 9.532% | 0.8s | 22295 | 10.229% | 0.8s | 22035 | 8.944% | 2.0s | 22071 | 9.122% | 2.0s |
| X-n289-k60 | 95151 | 104159 | 9.467% | 0.9s | 102202 | 7.410% | 1.0s | 102267 | 7.479% | 1.0s | 101494 | 6.666% | 1.0s | 100418 | 5.535% | 1.0s | 99739 | 4.822% | 1.0s | 100733 | 5.866% | 2.0s | 100080 | 5.180% | 2.0s |
| X-n294-k50 | 47161 | 52769 | 11.891% | 0.9s | 50886 | 7.898% | 1.0s | 51924 | 10.099% | 1.0s | 51033 | 8.210% | 0.9s | 50637 | 7.370% | 1.0s | 49929 | 5.869% | 1.0s | 50538 | 7.161% | 2.0s | 49877 | 5.759% | 2.0s |
| X-n298-k31 | 34231 | 37652 | 9.994% | 0.9s | 37344 | 9.094% | 1.0s | 36808 | 7.528% | 1.0s | 36785 | 7.461% | 0.9s | 37163 | 8.565% | 0.9s | 36993 | 8.069% | 1.0s | 36876 | 7.727% | 2.0s | 37068 | 8.288% | 2.0s |
| X-n303-k21 | 21736 | 23556 | 8.373% | 0.9s | 23263 | 7.025% | 1.0s | 23027 | 5.939% | 1.0s | 23097 | 6.262% | 0.9s | 23442 | 7.849% | 0.9s | 23748 | 9.257% | 0.9s | 23453 | 7.899% | 2.0s | 23548 | 8.336% | 2.0s |
| X-n308-k13 | 25859 | 28736 | 11.126% | 0.9s | 28518 | 10.283% | 1.0s | 29079 | 12.452% | 1.0s | 28030 | 8.396% | 0.9s | 28326 | 9.540% | 0.9s | 28913 | 11.810% | 0.9s | 28138 | 8.813% | 2.0s | 28440 | 9.981% | 2.0s |
| X-n313-k71 | 94043 | 102253 | 8.730% | 1.0s | 100620 | 6.994% | 2.0s | 100714 | 7.094% | 2.0s | 100083 | 6.423% | 1.0s | 99564 | 5.871% | 1.0s | 98899 | 5.164% | 1.0s | 98738 | 4.992% | 2.0s | 98931 | 5.198% | 2.0s |
| X-n317-k53 | 78355 | 82587 | 5.401% | 1.0s | 83632 | 6.735% | 2.0s | 87360 | 11.493% | 2.0s | 81981 | 4.628% | 1.0s | 80690 | 2.980% | 1.0s | 80542 | 2.791% | 1.0s | 80709 | 3.004% | 2.0s | 80472 | 2.702% | 2.0s |
| X-n322-k28 | 29834 | 32593 | 9.248% | 1.0s | 33497 | 12.278% | 2.0s | 32143 | 7.739% | 2.0s | 32403 | 8.611% | 0.9s | 32658 | 9.466% | 1.0s | 33206 | 11.303% | 1.0s | 32648 | 9.432% | 2.0s | 32541 | 9.074% | 2.0s |
| X-n327-k20 | 27532 | 30646 | 11.310% | 1.0s | 30603 | 11.154% | 2.0s | 29649 | 7.689% | 1.0s | 29638 | 7.649% | 0.9s | 29784 | 8.180% | 1.0s | 30053 | 12.426% | 1.0s | 29793 | 8.212% | 2.0s | 30089 | 9.287% | 2.0s |
| X-n331-k15 | 31102 | 34734 | 11.678% | 0.9s | 33636 | 8.147% | 1.0s | 34431 | 10.703% | 2.0s | 33597 | 8.022% | 1.0s | 34048 | 9.472% | 1.0s | 34578 | 11.176% | 1.0s | 33526 | 7.794% | 2.0s | 34014 | 9.363% | 2.0s |
| X-n336-k84 | 139111 | 152846 | 9.873% | 1.0s | 149229 | 7.273% | 2.0s | 150468 | 8.164% | 2.0s | 147371 | 5.938% | 1.0s | 146620 | 5.398% | 1.0s | 146707 | 5.460% | 1.0s | 147177 | 5.798% | 3.0s | 146465 | 5.286% | 2.0s |
| X-n344-k43 | 42050 | 46619 | 10.866% | 1.0s | 46947 | 11.646% | 2.0s | 45143 | 7.356% | 2.0s | 46098 | 9.627% | 1.0s | 44914 | 6.811% | 1.0s | 45571 | 8.373% | 1.0s | 45232 | 7.567% | 2.0s | 44746 | 6.411% | 2.0s |
| X-n351-k40 | 25896 | 29243 | 12.925% | 1.0s | 28373 | 9.565% | 2.0s | 28728 | 10.936% | 2.0s | 28628 | 10.550% | 1.0s | 28236 | 9.036% | 1.0s | 28059 | 8.353% | 1.0s | 28124 | 8.604% | 2.0s | 28130 | 8.627% | 2.0s |
| X-n359-k29 | 51505 | 55778 | 8.296% | 1.0s | 56165 | 9.048% | 2.0s | 54690 | 6.184% | 2.0s | 55013 | 6.811% | 1.0s | 55122 | 7.023% | 1.0s | 55183 | 7.141% | 1.0s | 55231 | 7.234% | 2.0s | 55158 | 7.093% | 2.0s |
| X-n367-k17 | 22814 | 26132 | 14.544% | 1.0s | 25588 | 12.159% | 2.0s | 26470 | 16.025% | 2.0s | 25150 | 10.239% | 1.0s | 25522 | 11.870% | 1.0s | 25534 | 11.923% | 1.0s | 25489 | 11.725% | 2.0s | 25489 | 11.725% | 2.0s |
| X-n376-k94 | 147713 | 156857 | 6.190% | 1.0s | 156546 | 5.980% | 2.0s | 156077 | 5.662% | 2.0s | 158456 | 7.273% | 1.0s | 151975 | 2.885% | 1.0s | 151390 | 2.489% | 1.0s | 151521 | 2.573% | 3.0s | 151614 | 2.641% | 3.0s |
| X-n384-k52 | 65940 | 73705 | 11.776% | 1.0s | 73570 | 11.571% | 2.0s | 70853 | 7.451% | 2.0s | 71089 | 7.809% | 1.0s | 70471 | 6.871% | 1.0s | 70611 | 7.084% | 1.0s | 70775 | 7.332% | 2.0s | 70479 | 6.884% | 2.0s |
| X-n393-k38 | 38260 | 43533 | 13.782% | 1.0s | 44638 | 16.670% | 2.0s | 41843 | 9.365% | 2.0s | 42161 | 10.196% | 1.0s | 41552 | 8.604% | 1.0s | 42934 | 12.216% | 1.0s | 41924 | 9.577% | 2.0s | 42192 | 10.277% | 3.0s |
| X-n401-k29 | 66154 | 71565 | 8.179% | 1.0s | 71787 | 8.515% | 2.0s | 69492 | 5.046% | 2.0s | 70480 | 6.539% | 1.0s | 69430 | 4.952% | 1.0s | 69875 | 5.625% | 1.0s | 69241 | 4.666% | 3.0s | 69991 | 5.800% | 2.0s |
| X-n411-k19 | 19712 | 23869 | 21.089% | 1.0s | 23139 | 17.385% | 2.0s | 24162 | 22.575% | 2.0s | 22203 | 12.637% | 1.0s | 22849 | 15.914% | 1.0s | 23521 | 19.323% | 1.0s | 22489 | 14.088% | 2.0s | 22768 | 15.503% | 2.0s |
| X-n420-k130 | 107798 | 122761 | 13.881% | 2.0s | 116362 | 7.944% | 2.0s | 120841 | 12.099% | 2.0s | 118046 | 9.507% | 2.0s | 117418 | 8.924% | 2.0s | 115012 | 6.692% | 2.0s | 115838 | 7.458% | 3.0s | 116853 | 8.400% | 3.0s |
| X-n429-k61 | 65449 | 74261 | 13.464% | 1.0s | 74158 | 13.307% | 2.0s | 71017 | 8.507% | 2.0s | 71070 | 8.588% | 1.0s | 70164 | 7.204% | 2.0s | 70969 | 8.434% | 1.0s | 70639 | 7.930% | 3.0s | 70617 | 7.896% | 2.0s |
| X-n439-k37 | 36391 | 41165 | 13.119% | 1.0s | 42161 | 15.856% | 2.0s | 38998 | 7.164% | 2.0s | 39947 | 9.772% | 1.0s | 39752 | 9.236% | 1.0s | 41149 | 13.075% | 1.0s | 39799 | 9.365% | 3.0s | 39697 | 9.085% | 3.0s |
| X-n449-k29 | 55233 | 60162 | 8.924% | 1.0s | 60015 | 8.658% | 2.0s | 59919 | 8.484% | 2.0s | 59925 | 8.495% | 1.0s | 60634 | 9.779% | 1.0s | 61144 | 10.702% | 1.0s | 60340 | 9.246% | 3.0s | 60723 | 9.940% | 3.0s |
| X-n459-k26 | 24139 | 29543 | 22.387% | 1.0s | 29100 | 20.552% | 2.0s | 26995 | 11.831% | 2.0s | 27247 | 12.780% | 1.0s | 27347 | 13.290% | 2.0s | 28267 | 17.101% | 1.0s | 27107 | 12.295% | 3.0s | 27510 | 13.965% | 3.0s |
| X-n469-k138 | 221824 | 252031 | 13.618% | 2.0s | 245581 | 10.710% | 3.0s | 242533 | 9.336% | 3.0s | 242197 | 9.184% | 2.0s | 238904 | 7.700% | 2.0s | 237548 | 7.089% | 2.0s | 236850 | 6.778% | 4.0s | 237001 | 6.842% | 3.0s |
| X-n480-k70 | 89449 | 101314 | 13.265% | 2.0s | 100611 | 11.931% | 2.0s | 96042 | 7.371% | 3.0s | 96484 | 7.865% | 2.0s | 95032 | 6.242% | 2.0s | 95466 | 6.727% | 2.0s | 95101 | 6.319% | 4.0s | 95211 | 6.442% | 3.0s |
| X-n491-k59 | 66483 | 77536 | 16.625% | 2.0s | 75226 | 13.151% | 2.0s | 72443 | 8.965% | 3.0s | 72142 | 8.512% | 2.0s | 72618 | 9.228% | 2.0s | 71702 | 7.850% | 2.0s | 72383 | 8.874% | 3.0s | 71730 | 7.892% | 3.0s |
| Avg. Gap (251 < N < 501) | | | 11.529% | | | 10.616% | | | 9.217% | | | 8.399% | | | 8.107% | | | 8.889% | | | 7.741% | | | 7.948% | |
| X-n502-k39 | 69226 | 75711 | 9.368% | 2.0s | 77033 | 11.278% | 3.0s | 73557 | 6.256% | 3.0s | 74317 | 7.354% | 2.0s | 71908 | 3.874% | 2.0s | 72655 | 4.953% | 2.0s | 72023 | 4.040% | 3.0s | 71682 | 3.548% | 3.0s |
| X-n513-k21 | 24201 | 34910 | 44.250% | 2.0s | 32858 | 35.771% | 3.0s | 27867 | 15.148% | 3.0s | 27871 | 15.165% | 2.0s | 28542 | 17.937% | 2.0s | 29422 | 21.573% | 2.0s | 27907 | 15.313% | 3.0s | 29139 | 20.404% | 3.0s |
| X-n524-k153 | 154593 | 176491 | 14.165% | 2.0s | 171734 | 11.088% | 3.0s | 178794 | 15.655% | 3.0s | 172181 | 11.377% | 2.0s | 174150 | 12.651% | 2.0s | 168181 | 8.790% | 2.0s | 171777 | 11.116% | 4.0s | 172580 | 11.635% | 4.0s |
| X-n536-k96 | 94846 | 109897 | 15.869% | 2.0s | 106031 | 11.793% | 3.0s | 103862 | 9.506% | 3.0s | 103854 | 9.498% | 2.0s | 103242 | 8.852% | 2.0s | 102355 | 7.917% | 2.0s | 102432 | 7.998% | 4.0s | 101712 | 7.239% | 4.0s |
| X-n548-k50 | 86700 | 110984 | 28.009% | 2.0s | 104240 | 20.231% | 3.0s | 101294 | 16.833% | 3.0s | 101549 | 17.127% | 2.0s | 100850 | 16.321% | 2.0s | 102318 | 18.014% | 2.0s | 100550 | 15.975% | 4.0s | 101918 | 17.552% | 4.0s |
| X-n561-k42 | 42717 | 55936 | 30.946% | 2.0s | 53110 | 24.330% | 3.0s | 47544 | 11.300% | 3.0s | 47835 | 11.981% | 2.0s | 49133 | 15.020% | 2.0s | 50287 | 17.721% | 2.0s | 48805 | 14.252% | 4.0s | 49363 | 15.558% | 4.0s |
| X-n573-k30 | 50673 | 60884 | 20.151% | 2.0s | 62033 | 22.418% | 3.0s | 59670 | 17.755% | 3.0s | 57388 | 13.252% | 2.0s | 56048 | 10.607% | 2.0s | 55353 | 9.236% | 2.0s | 54322 | 7.201% | 4.0s | 55058 | 8.654% | 4.0s |
| X-n586-k159 | 190316 | 226245 | 18.879% | 3.0s | 212545 | 11.680% | 4.0s | 209373 | 10.013% | 4.0s | 210049 | 10.369% | 3.0s | 206536 | 8.059% | 3.0s | 204649 | 7.531% | 3.0s | 204194 | 7.292% | 5.0s | 204848 | 7.636% | 5.0s |
| X-n599-k92 | 108451 | 131035 | 20.824% | 3.0s | 126654 | 16.785% | 4.0s | 118761 | 9.507% | 3.0s | 120022 | 10.669% | 3.0s | 116840 | 7.735% | 3.0s | 117784 | 8.606% | 3.0s | 117881 | 8.511% | 4.0s | 116938 | 7.826% | 4.0s |
| X-n613-k62 | 59535 | 77555 | 30.268% | 3.0s | 73633 | 23.680% | 3.0s | 67477 | 13.340% | 3.0s | 66818 | 12.233% | 2.0s | 67545 | 13.454% | 3.0s | 69069 | 16.014% | 3.0s | 67832 | 13.936% | 4.0s | 67730 | 13.765% | 4.0s |
| X-n627-k43 | 62164 | 76776 | 23.506% | 3.0s | 70744 | 13.802% | 3.0s | 68747 | 10.590% | 3.0s | 69716 | 12.149% | 3.0s | 67523 | 8.621% | 3.0s | 69361 | 11.577% | 3.0s | 68036 | 9.446% | 4.0s | 67896 | 9.221% | 4.0s |
| X-n641-k35 | 63684 | 83138 | 30.548% | 3.0s | 71986 | 13.036% | 4.0s | 70691 | 11.007% | 4.0s | 71120 | 11.676% | 3.0s | 70631 | 10.909% | 3.0s | 73624 | 15.608% | 3.0s | 70687 | 10.996% | 4.0s | 71974 | 13.017% | 4.0s |
| X-n655-k131 | 106780 | 120771 | 13.103% | 3.0s | 118758 | 11.217% | 4.0s | 119665 | 12.067% | 4.0s | 117339 | 9.889% | 3.0s | 112289 | 5.159% | 3.0s | 110657 | 3.631% | 4.0s | 111563 | 4.479% | 5.0s | 110267 | 3.266% | 5.0s |
| X-n670-k130 | 146332 | 183183 | 25.183% | 3.0s | 168210 | 14.951% | 4.0s | 180539 | 23.376% | 4.0s | 166596 | 13.848% | 3.0s | 168829 | 15.374% | 3.0s | 161571 | 10.414% | 3.0s | 167248 | 14.294% | 5.0s | 163051 | 11.425% | 5.0s |
| X-n685-k75 | 68205 | 92701 | 35.915% | 3.0s | 82607 | 21.116% | 4.0s | 78039 | 14.418% | 4.0s | 77265 | 13.283% | 3.0s | 77890 | 14.200% | 3.0s | 78473 | 15.055% | 4.0s | 77618 | 13.801% | 5.0s | 77132 | 13.088% | 5.0s |
| X-n701-k44 | 81923 | 92723 | 13.183% | 3.0s | 89704 | 9.498% | 4.0s | 89743 | 9.546% | 4.0s | 89006 | 8.887% | 3.0s | 90080 | 10.567% | 3.0s | 92198 | 12.542% | 3.0s | 90359 | 10.297% | 5.0s | 90703 | 10.717% | 5.0s |
| X-n716-k35 | 43373 | 59383 | 36.912% | 3.0s | 53170 | 20.282% | 4.0s | 49166 | 13.356% | 4.0s | 49524 | 14.182% | 3.0s | 49480 | 14.080% | 3.0s | 50605 | 16.674% | 3.0s | 49005 | 12.995% | 5.0s | 49403 | 13.907% | 5.0s |
| X-n733-k159 | 136187 | 175848 | 29.122% | 4.0s | 156268 | 14.745% | 5.0s | 158156 | 16.131% | 5.0s | 154339 | 13.329% | 4.0s | 148581 | 9.101% | 4.0s | 146080 | 7.264% | 4.0s | 149852 | 10.034% | 6.0s | 147334 | 8.185% | 6.0s |
| X-n749-k98 | 77269 | 102208 | 32.276% | 4.0s | 92403 | 19.586% | 5.0s | 88483 | 14.513% | 5.0s | 87621 | 13.397% | 4.0s | 85046 | 10.065% | 4.0s | 85325 | 10.426% | 4.0s | 85594 | 10.774% | 6.0s | 84712 | 9.633% | 6.0s |
| X-n766-k71 | 114417 | 132968 | 16.213% | 4.0s | 130101 | 13.708% | 5.0s | 133549 | 16.721% | 6.0s | 126445 | 10.512% | 4.0s | 125884 | 10.021% | 4.0s | 127552 | 11.479% | 4.0s | 126865 | 10.880% | 6.0s | 126387 | 10.462% | 6.0s |
| X-n783-k48 | 72386 | 108577 | 49.997% | 4.0s | 96432 | 33.219% | 5.0s | 92299 | 27.508% | 5.0s | 82041 | 13.337% | 4.0s | 82839 | 14.441% | 4.0s | 87562 | 20.965% | 5.0s | 83324 | 15.109% | 6.0s | 83864 | 15.855% | 6.0s |
| X-n801-k40 | 73311 | 92125 | 25.663% | 4.0s | 87187 | 18.928% | 6.0s | 89100 | 21.537% | 6.0s | 88259 | 20.390% | 5.0s | 86121 | 17.474% | 4.0s | 94076 | 28.325% | 4.0s | 85696 | 16.894% | 6.0s | 89478 | 22.053% | 6.0s |
| X-n819-k171 | 158121 | 192102 | 21.491% | 5.0s | 178856 | 13.113% | 7.0s | 175286 | 10.856% | 6.0s | 177119 | 12.015% | 5.0s | 174446 | 10.324% | 5.0s | 172387 | 9.022% | 5.0s | 171520 | 8.474% | 7.0s | 171676 | 8.573% | 7.0s |
| X-n837-k142 | 193737 | 231602 | 19.235% | 5.0s | 216838 | 11.922% | 7.0s | 213765 | 10.338% | 7.0s | 215009 | 10.980% | 5.0s | 206869 | 6.777% | 5.0s | 209540 | 8.157% | 5.0s | 208667 | 7.706% | 8.0s | 209031 | 7.894% | 7.0s |
| X-n856-k95 | 88965 | 117243 | 31.786% | 5.0s | 105763 | 18.882% | 6.0s | 109164 | 22.704% | 7.0s | 99273 | 11.587% | 5.0s | 98164 | 10.340% | 5.0s | 102312 | 15.003% | 5.0s | 99233 | 11.542% | 7.0s | 99914 | 11.807% | 7.0s |
| X-n876-k59 | 99299 | 114212 | 15.018% | 5.0s | 114175 | 14.981% | 7.0s | 110476 | 11.256% | 7.0s | 112919 | 13.716% | 5.0s | 107477 | 8.236% | 5.0s | 109693 | 10.467% | 5.0s | 107589 | 8.349% | 7.0s | 110843 | 11.625% | 7.0s |
| X-n895-k37 | 53860 | 106062 | 96.922% | 6.0s | 70363 | 30.641% | 6.0s | 64648 | 20.030% | 6.0s | 64208 | 19.248% | 5.0s | 64225 | 19.244% | 5.0s | 73280 | 36.056% | 6.0s | 62460 | 15.967% | 7.0s | 67830 | 25.938% | 8.0s |
| X-n916-k207 | 329179 | 387367 | 17.677% | 7.0s | 374899 | 13.889% | 8.0s | 361709 | 9.882% | 9.0s | 360505 | 9.516% | 7.0s | 353039 | 7.248% | 7.0s | 353185 | 7.681% | 7.0s | 353232 | 7.304% | 8.0s | 352488 | 7.081% | 10.0s |
| X-n936-k151 | 132715 | 200816 | 51.314% | 7.0s | 161700 | 21.840% | 8.0s | 182393 | 37.432% | 8.0s | 158680 | 19.564% | 7.0s | 162903 | 22.746% | 7.0s | 154847 | 16.678% | 7.0s | 157310 | 18.532% | 9.0s | 155618 | 17.258% | 9.0s |
| X-n957-k87 | 85465 | 126220 | 47.686% | 7.0s | 124190 | 45.311% | 8.0s | 106292 | 24.369% | 8.0s | 104024 | 21.715% | 7.0s | 103089 | 20.621% | 7.0s | 108664 | 27.144% | 7.0s | 103134 | 20.674% | 9.0s | 108603 | 25.084% | 9.0s |
| X-n979-k58 | 118976 | 138987 | 16.819% | 7.0s | 132651 | 11.494% | 9.0s | 133186 | 11.944% | 8.0s | 133188 | 11.945% | 8.0s | 129633 | 8.957% | 7.0s | 133201 | 11.956% | 7.0s | 129535 | 8.875% | 9.0s | 132728 | 11.559% | 9.0s |
| X-n1001-k43 | 72355 | 132976 | 83.783% | 7.0s | 89175 | 23.249% | 9.0s | 85919 | 18.746% | 8.0s | 84377 | 16.615% | 7.0s | 85852 | 18.654% | 7.0s | 92974 | 28.497% | 7.0s | 84390 | 16.633% | 9.0s | 93476 | 29.191% | 10.0s |
| Avg. Gap (501 < N < 1001) | | | 30.190% | | | 18.918% | | | 14.994% | | | 13.188% | | | 12.253% | | | 14.199% | | | 11.509% | | | 12.814% | |
| Avg. Gap | | | 15.863% | | | 11.693% | | | 10.253% | | | 9.108% | | | 8.428% | | | 9.230% | | | 7.973% | | | 8.441% | |

To facilitate analysis, we partition the evaluation set based on problem instance size into three subsets: instances with $N < 251$, $251 \leq N < 501$ and $501 < N \leq 1001$. Across all three subsets, both MoSES(RF) and MoSES(CaDA) consistently outperform their respective baselines, RF-TE and CaDA. On the larger instance sets ($251 \leq N < 501$ and $501 < N \leq 1001$), MoSES(RF) demonstrates superior performance compared to MoSES(CaDA). Conversely, on the smaller instance set ($N < 251$), MoSES(CaDA) achieves better results than MoSES(RF). Overall, MoSES(RF) exhibits the best performance among all evaluated methods.

## B.3  Hyperparameter Studies

Figure 5 presents an analysis of how varying LoRA ranks influence the performance of MoSES(RF) and MoSES(CaDA) under the settings of $N = 50$ and $N = 100$. We allow the ranks for the frozen modules $\{B_i A_i\}_{i=1}^4$ and the trainable module $\hat{B}\hat{A}$ to differ, denoted as $r_{\text{frozen}}$ and $r_{\text{free}}$ respectively. In Figure 5, the blue curves represent experiments where $r_{\text{frozen}} = 32$ is fixed while $r_{\text{free}}$ varies from 4 to 32. Conversely, the green curves fix $r_{\text{free}} = 32$ while varying $r_{\text{frozen}}$ over the same range. The y-axis indicates the average optimality gap across 16 VRP variants. For MoSES(RF)

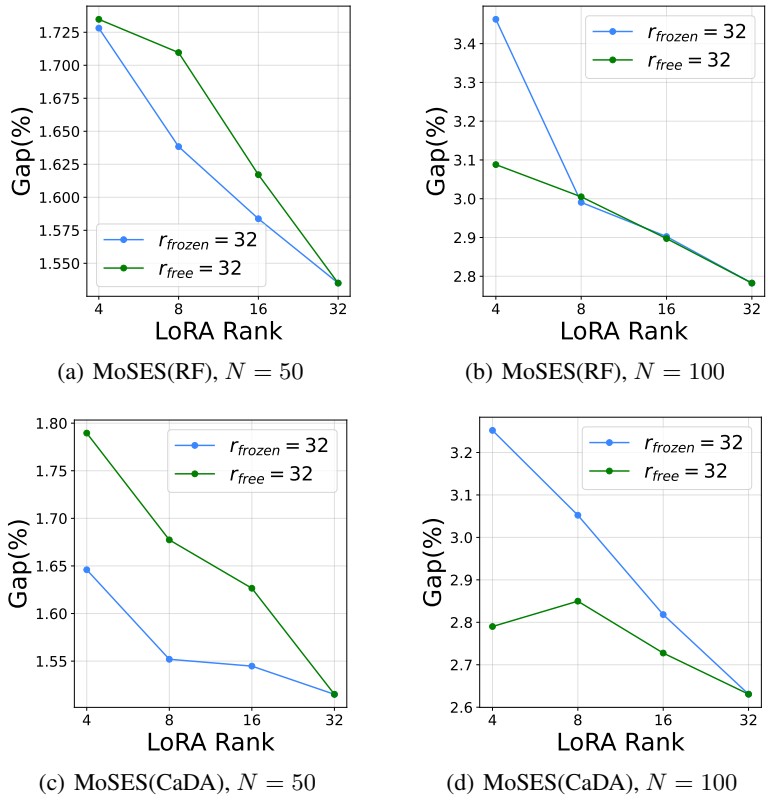

(a) MoSES(RF), $N = 50$  (b) MoSES(RF), $N = 100$

(c) MoSES(CaDA), $N = 50$  (d) MoSES(CaDA), $N = 100$

Figure 5: This figure investigates the effect of LoRA ranks on MoSES(RF) and MoSES(CaDA) under both $N = 50$ and $N = 100$ settings.

under the setting of $N = 50$, it is observed that reducing the LoRA rank of the trainable module $\hat{B}\hat{A}$ while keeping $r_{\text{frozen}} = 32$ incurs less performance degradation compared to reducing the LoRA rank of the frozen modules $\{B_i A_i\}_{i=1}^4$ with $r_{\text{free}} = 32$, as shown in Figure 5(a). However, under the setting of $N = 100$, the performance decline caused by reducing $r_{\text{frozen}}$ with $r_{\text{free}} = 32$ is generally smaller than that caused by reducing $r_{\text{free}}$ with $r_{\text{frozen}} = 32$ (see Figure 5(b)). This suggests that for smaller-scale problem instances ($N = 50$), MoSES(RF) relies more heavily on the frozen modules $\{B_i A_i\}_{i=1}^4$ inherited from the basis solvers. In contrast, for larger-scale problem instances ($N = 100$), the trainable module $\hat{B}\hat{A}$ becomes at least as critical as the frozen modules in contributing to overall performance. Likewise, similar trends are observed for MoSES(CaDA) under the setting of $N = 50$,, as illustrated in Figure 5(c). Under the setting of $N = 100$, the performance gap between reducing the LoRA rank $r_{\text{frozen}}$ of the frozen modules (with fixed trainable module rank $r_{\text{free}}$) and reducing the LoRA rank $r_{\text{free}}$ of the trainable module (with fixed frozen module rank $r_{\text{frozen}}$) becomes more pronounced (see Figure 5(d)). This suggests that at the smaller scale ($N = 50$), the frozen LoRA experts contribute more significantly to the performance of MoSES(CaDA) than the trainable LoRA expert. In contrast, at the larger scale ($N = 100$), the trainable LoRA expert plays a more critical role in driving performance than the frozen LoRA experts. We observe similar performance trends across each individual VRP variant, as shown in Figures 9 and 10.

## B.4  Ablation studies

We evaluate our proposed Gated-LoRA against standard LoRA that varies $\beta$ from 0.1 to 1.0, as shown in Figure 6. Since both methods are applied during the fine-tuning phase to produce specialized basis solvers for the basis VRP variants (OVRP, VRPB, VRPL, VRPTW), Figure 6 reports the average optimality gap and average cost across these four variants. When built on the RF-based backbone, the standard LoRA expert achieves its best performance at $\beta = 0.8$ in terms of both optimality gap and cost (blue curves in Figures 6(a) and 6(b)). However, the noticeable gap between the lowest

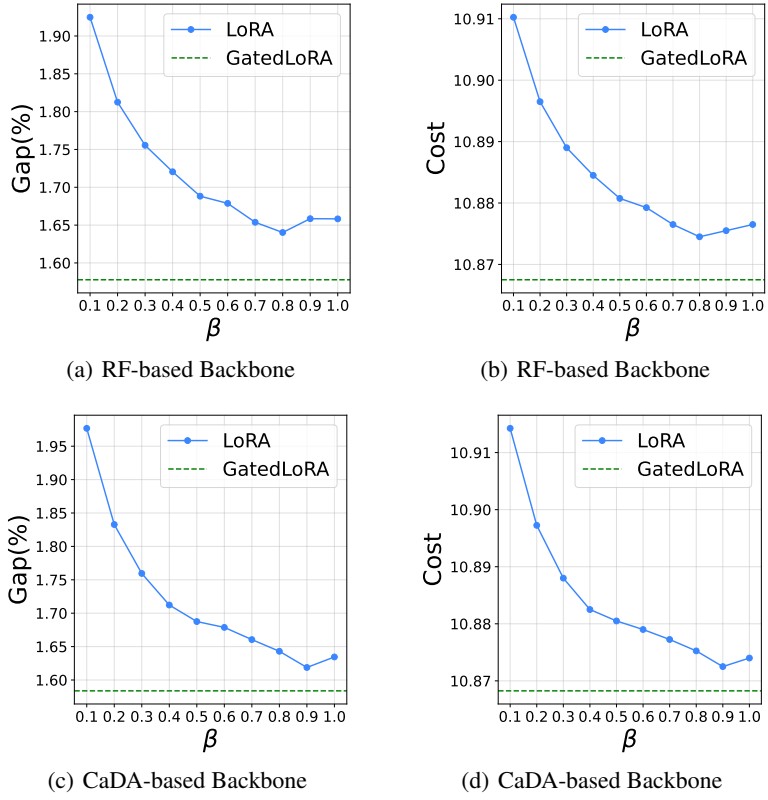

(a) RF-based Backbone       (b) RF-based Backbone

(c) CaDA-based Backbone       (d) CaDA-based Backbone

Figure 6: This figure compares the performance of Gated-LoRA with standard LoRA across varying values of $\beta \in (0, 1]$, under both RF-based and CaDA-based backbones.

point of the blue curves and the green dashed line indicates that our proposed Gated-LoRA module outperforms standard LoRA. Similarly, when built on the CaDA-based backbone, the standard LoRA expert performs best at $\beta = 0.9$, yet the Gated-LoRA module still achieves superior performance. These experimental results support our hypothesis that problem instances from any basis VRP variant (excluding CVRP) are OOD inputs to the frozen backbone model, resulting in task-misaligned features in the embeddings generated by the backbone.

Figure 7 presents a performance comparison of different activation functions used in the adaptive gating mechanism of the mixture of specialized experts, including $\mathrm{softmax}(\cdot)$, $\mathrm{norm\_softplus}(\cdot)$, and $\mathrm{sigmoid}(\cdot)$. In both the $N = 50$ and $N = 100$ settings, MoSES(RF) demonstrates superior performance when employing the $\mathrm{norm_softplus}(\cdot)$ function, while performance degrades when using $\mathrm{sigmoid}(\cdot)$. This suggests that MoSES(RF) benefits from a convex combination of the pretrained, frozen LoRA experts. In contrast, MoSES(CaDA) achieves its best performance with the $\mathrm{sigmoid}(\cdot)$ function, indicating a preference for selectively reusing or suppressing specific LoRA experts without being constrained by a unit-sum requirement. Figure 11 presents a performance comparison of different activation functions across each VRP variant. As shown in Figure 8, the dense routing method yields the best performance for both MoSES(RF) and MoSES(CaDA), while a significant performance degradation is observed when the trainable LoRA module $\hat{B}\hat{A}$ is ablated. This suggests that, to achieve better in-distribution performance, the unified neural solver should leverage the knowledge encoded in the pretrained specialized LoRA experts, which capture the ability to solve basis VRP variants. At the same time, the new knowledge learned through the trainable LoRA expert is also crucial for enhancing performance. Figure 12 presents a comparative result of different routing strategies across each VRP variant.

To investigate the impact of incorporating explicit task descriptors into the gating mechanism on performance, we modified the input of the gating mechanism to explicitly include constraint flags. This modification is implemented on MoSES(RF) under the $N = 50$ setting. As shown in Table 6,

Table 6: Impact of incorporating task descriptors into the gating mechanism on performance.

| Task | w/o Task Descriptors | w/ Task Descriptors |
|---|---|---|
| CVRP | 0.900% | 0.880% |
| VRPTW | 1.445% | 1.452% |
| OVRP | 1.892% | 1.940% |
| VRPL | 1.089% | 1.058% |
| VRPB | 2.342% | 2.339% |
| OVRPTW | 0.959% | 0.950% |
| VRPBL | 3.185% | 3.187% |
| VRPBLTW | 1.370% | 1.382% |
| VRPBTW | 1.121% | 1.109% |
| VRPLTW | 1.811% | 1.820% |
| OVRPB | 1.979% | 1.995% |
| OVRPBL | 2.014% | 2.013% |
| OVRPBLTW | 0.791% | 0.786% |
| OVRPBTW | 0.783% | 0.789% |
| OVRPL | 1.917% | 1.941% |
| OVRPLTW | 0.962% | 0.952% |
| **Average** | **1.535%** | 1.537% |

Table 7: Performance Comparison of LoRA Expert vs. Linear Expert.

| Task | LoRA Expert | Linear Expert |
|---|---|---|
| CVRP | 0.900% | 0.878% |
| VRPTW | 1.445% | 1.427% |
| OVRP | 1.892% | 1.808% |
| VRPL | 1.089% | 1.061% |
| VRPB | 2.342% | 2.217% |
| OVRPTW | 0.959% | 0.938% |
| VRPBL | 3.185% | 2.960% |
| VRPBLTW | 1.370% | 1.431% |
| VRPBTW | 1.121% | 1.145% |
| VRPLTW | 1.811% | 1.845% |
| OVRPB | 1.979% | 1.764% |
| OVRPBL | 2.014% | 1.775% |
| OVRPBLTW | 0.791% | 0.783% |
| OVRPBTW | 0.783% | 0.778% |
| OVRPL | 1.917% | 1.819% |
| OVRPLTW | 0.962% | 0.943% |
| **Average** | 1.535% | **1.473%** |

we observe that including task descriptors in the gating mechanism reduces the optimality gap for some tasks, while increasing it for others. Overall, the performance impact is minimal. We speculate that the gating mechanism is capable of implicitly identifying different VRP variants based solely on the problem instance. As a result, the additional task descriptors do not provide a performance gain.

To investigate linear task-specific experts as an alternative, we designed an experiment where the basis VRP solvers for OVRP, VRPB, VRPL, and VRPTW are trained using linear adapters. These linear adapters are then integrated into the unified solver to enable reuse. We implemented this experiment on the MoSES(RF) model under the setting of $N = 50$. The experimental results are presented in Table 7. We observe that the proposed linear expert contributes to performance improvement.

Since our method introduces additional time overhead due to the dynamic gating mechanism compared to its direct baseline, we apply the $32\times$ data augmentation technique proposed in [2] to the baseline method to examine whether increasing the inference time of the baseline method to match the runtime of our method can also yield a performance improvement. We conduct this experiment based on

Table 8: Impact of data augmentation on performance.

| Method | Avg. Gap | Avg. Time |
|---|---|---|
| RF w/ $8\times$ dihedral | 2.063% | 2.0s |
| RF w/ $32\times$ symmetric | 3.261% | 6.1s |
| MoSES(RF) w/ 8x dihedral | 1.535% | 5.8s |

Table 9: Impact of poorly trained basis VRP solver on performance.

| | Tasks w/ OVRP | Tasks w/o OVRP |
|---|---|---|
| Fully Trained OVRP Solver | 1.412% | 1.658% |
| Poorly Trained OVRP Solver | 1.524% | 1.654% |
| | Tasks w/ VRPL | Tasks w/o VRPL |
| Fully Trained VRPL Solver | 1.642% | 1.428% |
| Poorly Trained VRPL Solver | 1.660% | 1.453% |
| | Tasks w/ VRPB | Tasks w/o VRPB |
| Fully Trained VRPB Solver | 1.698% | 1.372% |
| Poorly Trained VRPB Solver | 1.722% | 1.393% |
| | Tasks w/ VRPTW | Tasks w/o VRPTW |
| Fully Trained VRPTW Solver | 1.155% | 1.915% |
| Poorly Trained VRPTW Solver | 1.378% | 1.969% |

the RF model in the $N = 50$ setting. As shown in Table 8, we observe that RF with $32\times$ data augmentation does not result in a performance improvement, despite the increased inference time. Therefore, we conclude that the additional inference time introduced by our method is justified by the performance gains it delivers.

To evaluate the robustness of our method against poorly trained basis solvers, we conducted experiments using MoSES(RF) under the $N = 50$ setting. In our method, each basis solver typically employs a LoRA rank of 32. We observed that reducing the LoRA rank to 4 significantly degrades performance, so we used basis solvers with a LoRA rank of 4 to simulate poorly trained solvers. Specifically, we trained four such poorly trained solvers, each corresponding to one of the basis VRP variants. We then replaced one of the original high-performing solvers in MoSES(RF) with a corresponding poor solver and retrained the gating mechanism accordingly. In Table 9, 'Fully Trained OVRP Solver' and 'Poorly Trained OVRP Solver' refer to cases where MoSES(RF) uses a fully trained basis solver and a poorly trained basis solver for OVRP, respectively, while the remaining basis VRP solvers remain unchanged. We compare their performance separately on VRP variants with and without the open route constraint. Thus, 'Tasks w/ OVRP' and 'Tasks w/o OVRP' represent the average optimality gap across 8 VRP variants with and without the open route constraint, respectively. The same notation is also applied to the remaining constraints in Table 9. From the results, we observe that the optimality gap is more significantly affected for tasks that include the corresponding constraints when the related basis solver is poorly trained.

## B.5 Visualizing Adaptive Gating Mechanisms

In Figure 13, we present a visualization of the weights assigned to five basis solvers, each corresponding to a specific constraint: Capacity (C), Open Route (O), Distance Limit (L), Backhaul (B), and Time Window (TW), for both MoSES(RF) and MoSES(CaDA) under the setting $N = 100$. Each weight is calculated by averaging across layers, time steps, and problem instances, based on evaluations on 1,000 problem instances per model.

We observe that, since each VRP variant is derived from CVRP, the CVRP solver consistently receives a dominant weight allocation in both MoSES(RF) and MoSES(CaDA). Furthermore, when a VRP variant incorporates the Time Window constraint, the corresponding VRPTW solver receives

significantly higher weights compared to other solvers, with the exception of the CVRP solver. Overall, basis solvers associated with the constraints present in the current VRP variant tend to be preferred. Additionally, solvers corresponding to irrelevant basis variants may still receive non-negligible weights, likely due to partial similarities among the basis VRP variants. Due to the use of the $\text{sigmoid}(\cdot)$ activation function in MoSES(CaDA), the model tends to assign weights with greater magnitudes, thereby capturing more informative signals from the basis solvers.

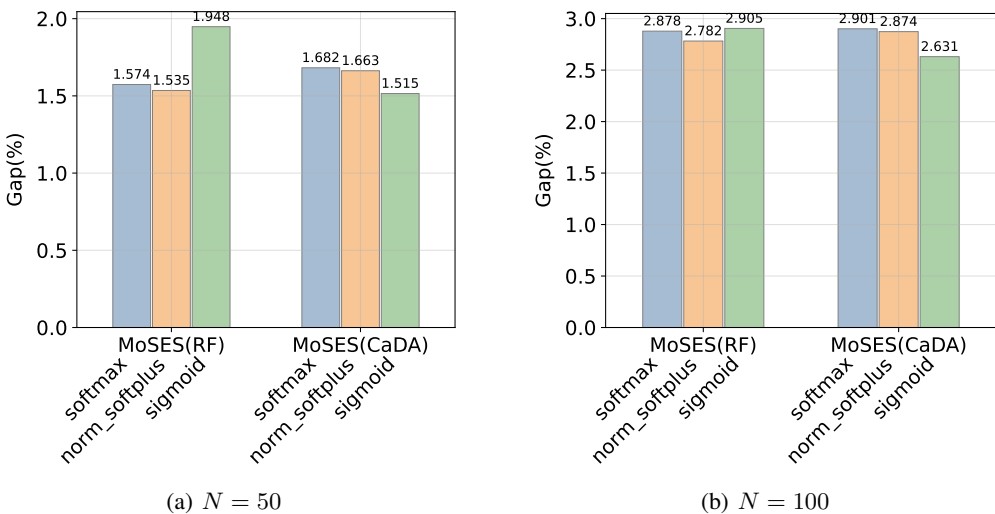

(a) $N = 50$          (b) $N = 100$

Figure 7: This figure compares the performance of activation functions used in the adaptive gating mechanism of the mixture of specialized experts for both MoSES(RF) and MoSES(CaDA) under the settings of $N = 50$ and $N = 100$.

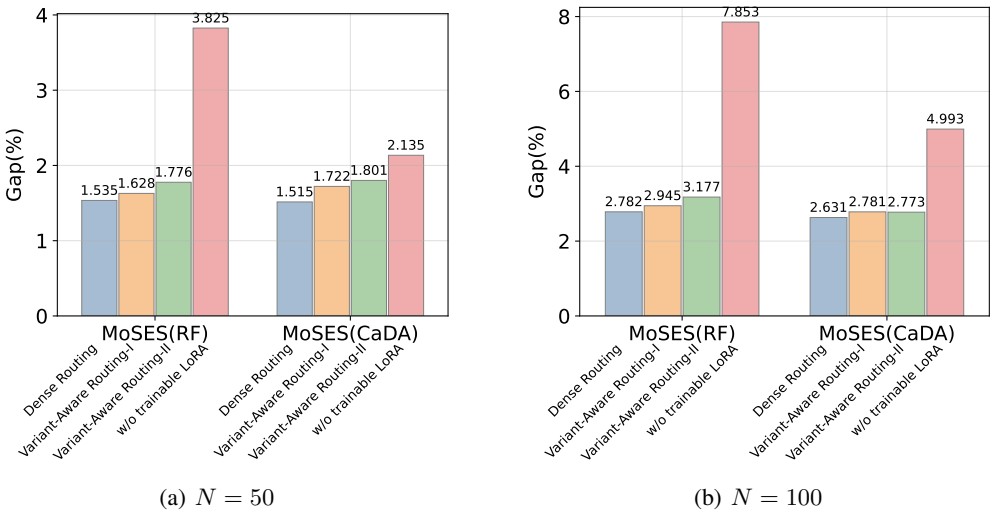

(a) $N = 50$          (b) $N = 100$

Figure 8: This figure compares the performance of routing strategies used in the adaptive gating mechanism of the mixture of specialized experts for both MoSES(RF) and MoSES(CaDA) under the settings of $N = 50$ and $N = 100$.

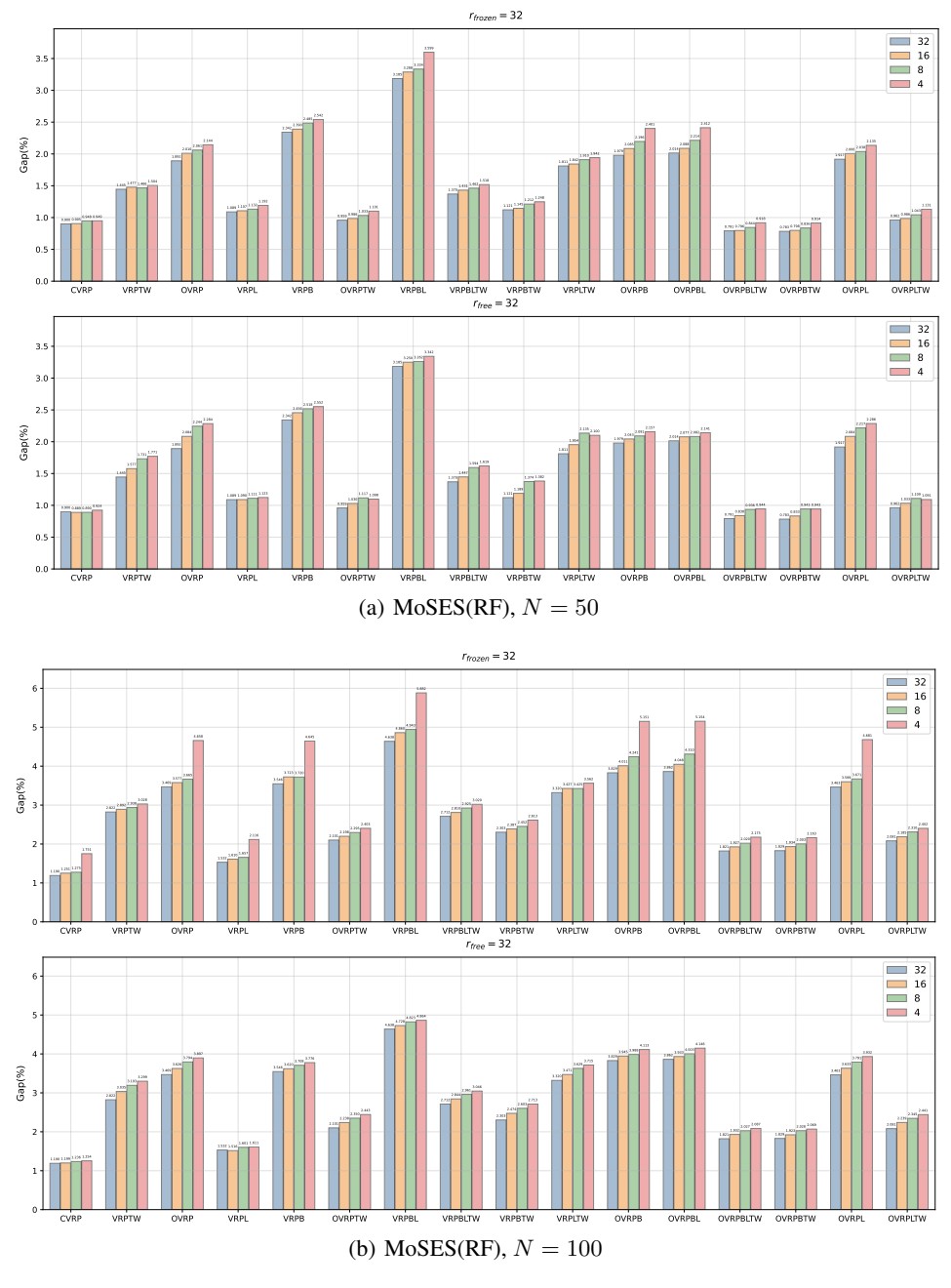

(a) MoSES(RF), $N = 50$

(b) MoSES(RF), $N = 100$

Figure 9: This figure illustrates the performance trends of MoSES(RF) across each VRP variant under both $N = 50$ and $N = 100$ settings, as either $r_{\text{frozen}}$ or $r_{\text{free}}$ is varied while keeping the other fixed.

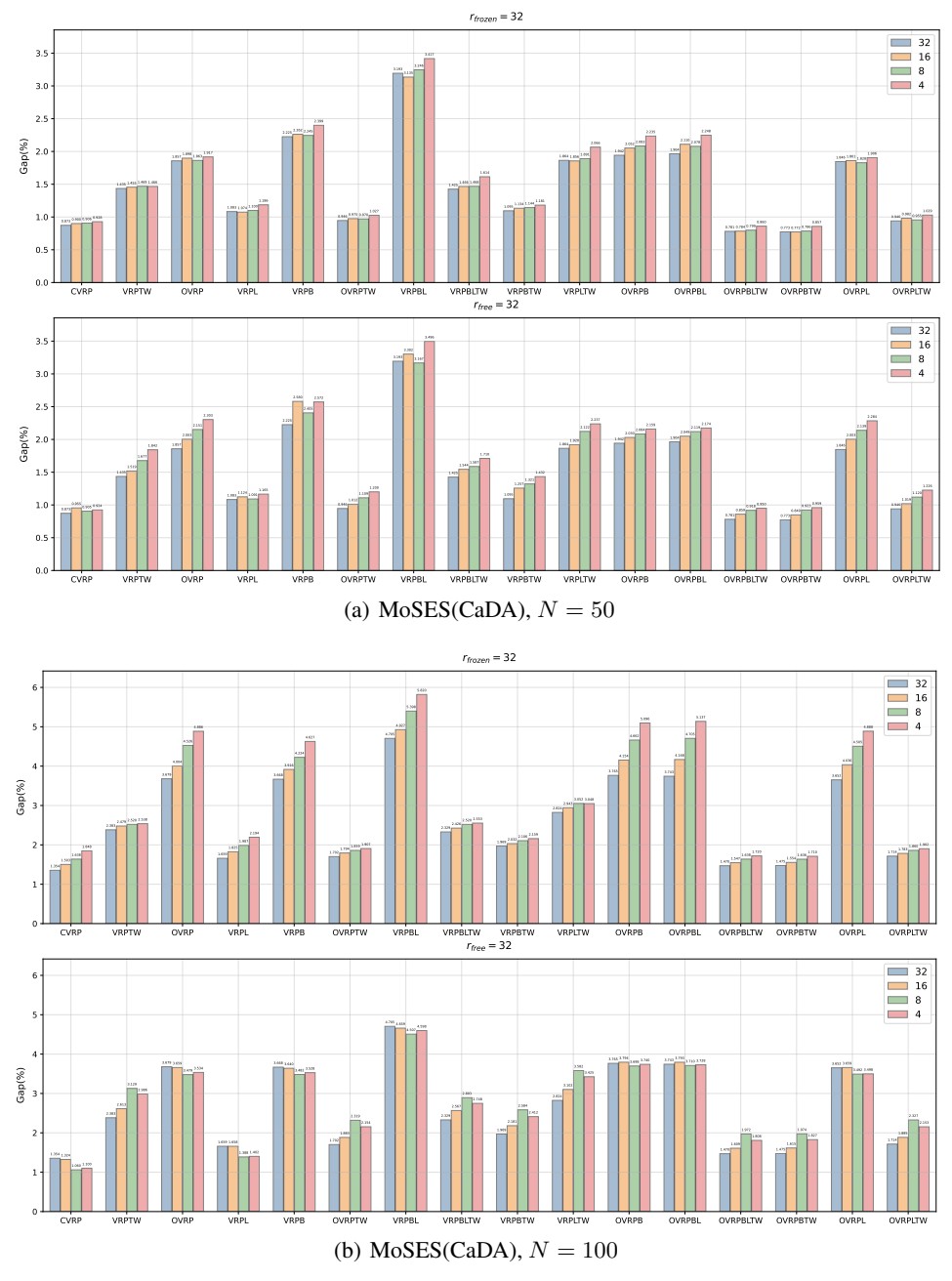

(a) MoSES(CaDA), $N = 50$

(b) MoSES(CaDA), $N = 100$

Figure 10: This figure illustrates the performance trends of MoSES(CaDA) across each VRP variant under both $N = 50$ and $N = 100$ settings, as either $r_{\text{frozen}}$ or $r_{\text{free}}$ is varied while the other is held constant.

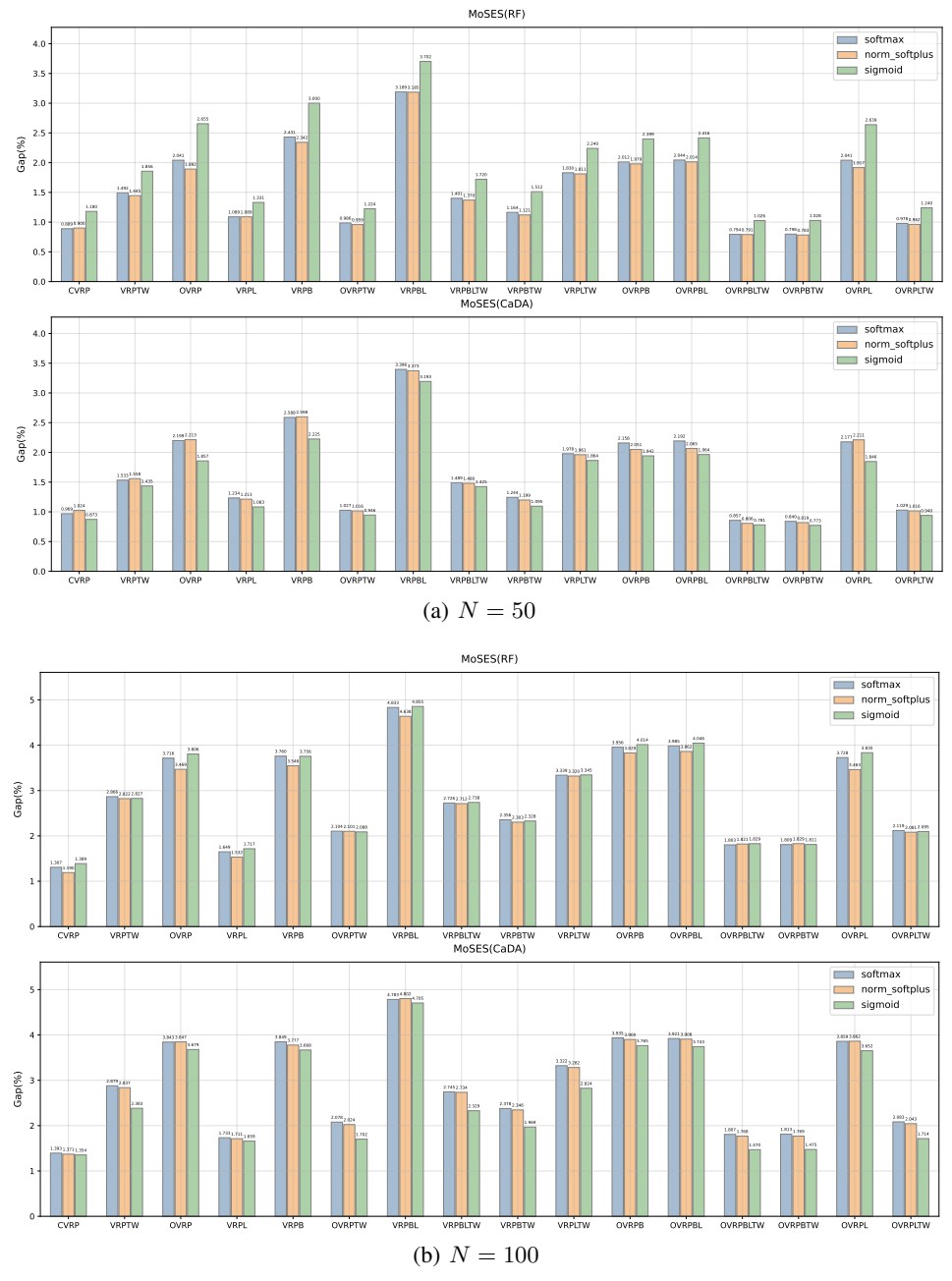

Figure 11: This figure illustrates the performance comparisons among different activation functions used in the adaptive gating mechanism of both MoSES(RF) and MoSES(CaDA) across various VRP variants under the $N = 50$ and $N = 100$ settings.

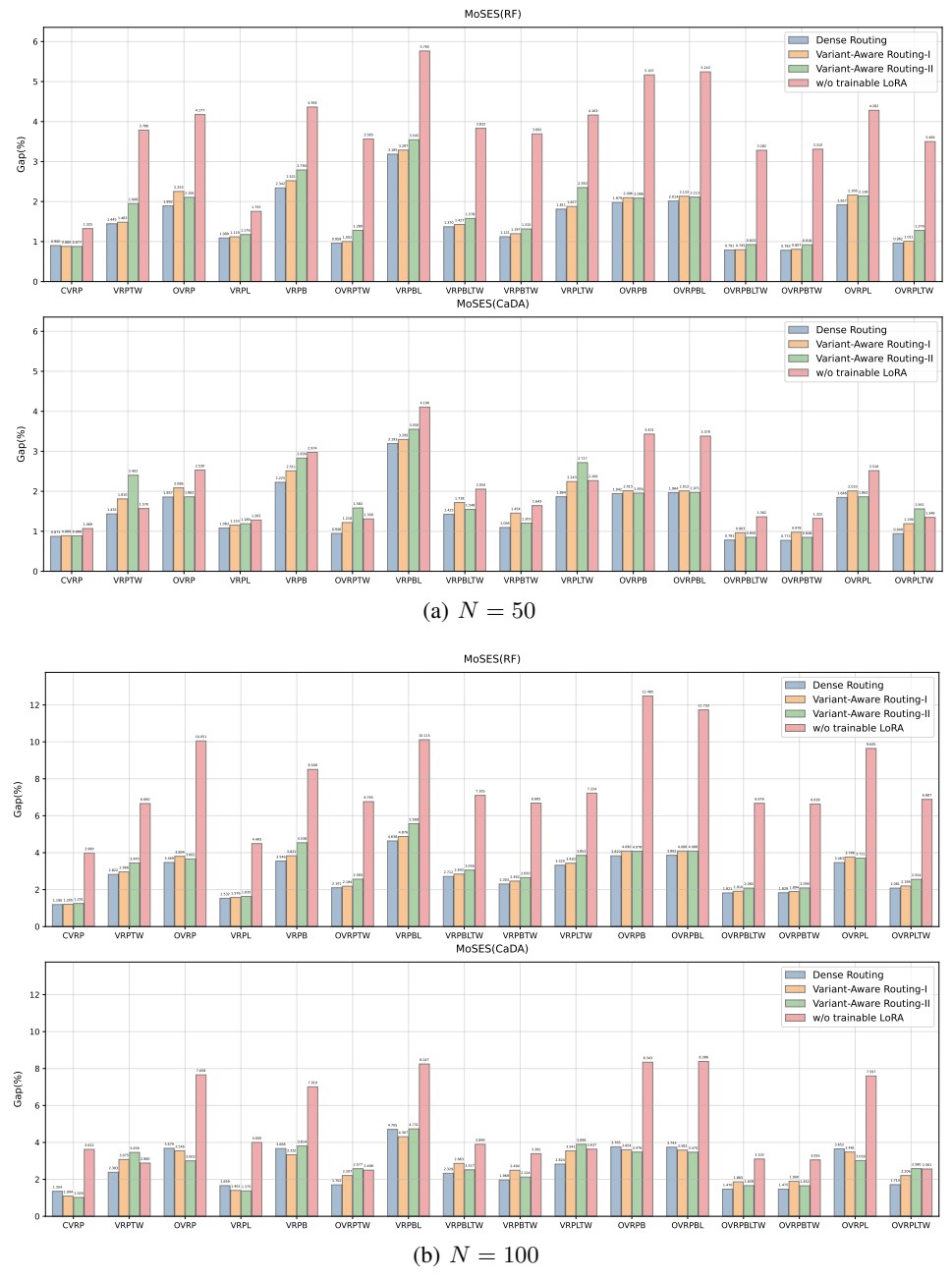

Figure 12: This figure illustrates the performance comparisons among different routing strategies used in the adaptive gating mechanism of both MoSES(RF) and MoSES(CaDA) across various VRP variants under the $N = 50$ and $N = 100$ settings.

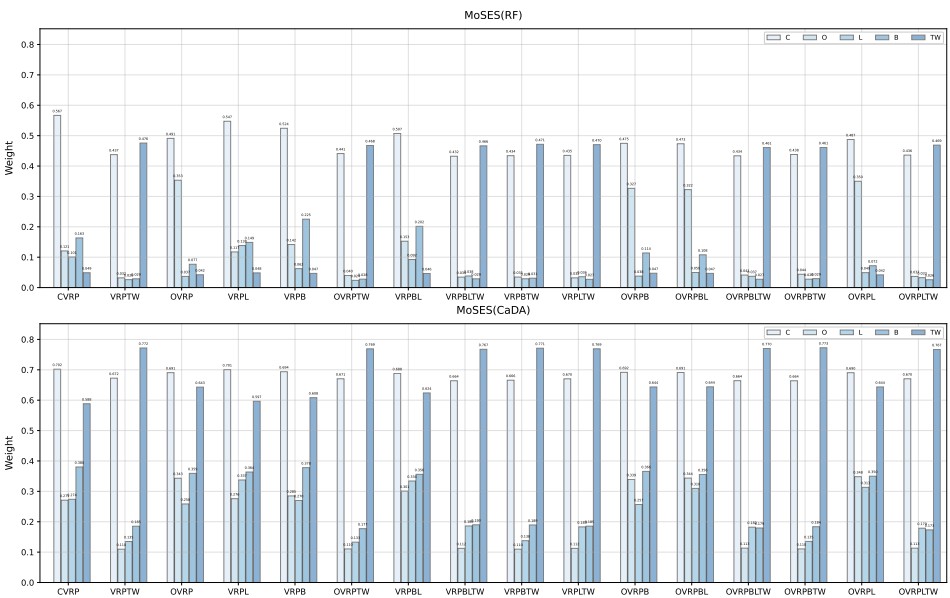

Figure 13: This figure illustrates the behavior of the adaptive gating mechanisms in both MoSES(RF) and MoSES(CaDA) under the setting $N = 100$.

## C  Proofs

**Theorem 3.** *The optimal unified policy $\pi^*$ and the $i$-th optimal basis policy $\pi^{(i)*}$ coincide in their value functions for each state $s_t$ associated with the $i$-th basis task: $V^{\pi^*,\mathcal{P}}(s_t) = V^{\pi^{(i)*},\mathcal{P}}(s_t)$. Furthermore, if both the optimal unified policy and the optimal basis policy are unique, then for each state-action pair $(s_t, a_t)$ corresponding to the $i$-th basis task, it holds that $\pi^*(a_t|s_t) = \pi^{(i)*}(a_t|s_t)$.*

*Proof.* By definition, the value function $V^{\pi^*,\mathcal{P}}$ induced by the optimal policy $\pi^*$ is greater than or equal to the value function $V^{\pi,\mathcal{P}}$ of any unified policy $\pi$ for all feasible states $s_t$ at each time step $t$. Formally, $\forall \pi, s_t \in \mathcal{S}_t, t \in \{0, \dots, T\}$, it holds that $V^{\pi^*,\mathcal{P}}(s_t) \geq V^{\pi,\mathcal{P}}(s_t)$. Likewise, the $i$-the optimal basis policy $\pi^{(i)*}$ $(i \geq 0)$ maximizes the value function $V^{\pi^{(i)},\mathcal{P}}$, evaluated on states induced by the initial state distribution defined over $\mathcal{S}_0^{(0)}$ for $i = 0$, or over $\mathcal{S}_0^{(0)} \times \mathcal{S}_0^{(i)}$ for $i \geq 1$, corresponding to the $i$-th basis task at each time step. That is, for all $\pi^{(0)}$, $s_t^{(0)} \in \mathcal{S}_t^{(0)}$, and $t \in \{0, \dots, T\}$, we have $V^{\pi^{(0)*},\mathcal{P}}(s_t^{(0)}) \geq V^{\pi^{(0)},\mathcal{P}}(s_t^{(0)})$. Similarly, for all $\pi^{(i)}$, $(s_t^{(0)}, s_t^{(i)}) \in \mathcal{S}_t^{(0)} \times \mathcal{S}_t^{(i)}$, $t \in \{0, \dots, T\}$, and $i \geq 1$, it holds that $V^{\pi^{(i)*},\mathcal{P}}((s_t^{(0)}, s_t^{(i)})) \geq V^{\pi^{(i)},\mathcal{P}}((s_t^{(0)}, s_t^{(i)}))$.

According to the definitions of the SDMDP framework and the basis task, it is evident that, at each time step, the state space associated with a basis task is a subset of the state space defined for the unified policy. Specifically, at each time step $t$, for all $s_t^{(0)} \in \mathcal{S}_t^{(0)}$, it holds that $s_t^{(0)} \in \mathcal{S}_t$. Likewise, for all $(s_t^{(0)}, s_t^{(i)}) \in \mathcal{S}_t^{(0)} \times \mathcal{S}_t^{(i)}$, and $i \geq 1$ we have $(s_t^{(0)}, s_t^{(i)}) \in \mathcal{S}_t$. As a result, at each time step $t$, for all $s_t^{(0)} \in \mathcal{S}_t^{(0)}$, it follows that $V^{\pi^*,\mathcal{P}}(s_t^{(0)}) = V^{\pi^{(0)*},\mathcal{P}}(s_t^{(0)})$. Similarly, for all $(s_t^{(0)}, s_t^{(i)}) \in \mathcal{S}_t^{(0)} \times \mathcal{S}_t^{(i)}$, and $i \geq 1$, we have $V^{\pi^*,\mathcal{P}}((s_t^{(0)}, s_t^{(i)})) = V^{\pi^{(i)*},\mathcal{P}}((s_t^{(0)}, s_t^{(i)}))$. Moreover, if the $i$-the basis task $(i > 0)$ admits a unique optimal policy $\pi^{(i)*}$, and the optimal unified policy $\pi^*$ is also unique, then for each state $s_t$ associated with the $i$-th basis task, it holds that $\pi^*(a_t|s_t) = \pi^{(i)*}(a_t|s_t)$, where $a_t \in \mathcal{A}(s_t)$. $\qquad\square$

**Assumption 3.** *In the SDMDP framework, any state $s$ is composed of $m+1$ conditionally independent basis states, denoted as $s = \{s^{(b_i)}\}_{i=0}^m$. Accordingly, we assume that any policy $\pi$ is capable of extracting the basis state embedding $z^{(b_i)} \in \mathbb{R}^d$ for each $s^{(b_i)}$, where $i = 0, \dots, m$. Under this assumption, we further posit that there exists a deterministic bijective mixture function $f_\phi : \mathcal{S} \times \prod_{i=0}^m \mathbb{R}^d \to \mathbb{R}^d$, parameterized by $\phi$, which maps the basis state embeddings $\{z^{(b_i)}\}_{i=0}^m$ to the state embedding $z \in \mathbb{R}^d$ for the given state $s$, represented as $z = f_\phi(z^{(b_0)}, \dots, z^{(b_m)}; s)$. Thus, the policy defined over the action space can be rewritten as*

$$\pi(a|s) = \sum_z \pi(a|z)\pi(z|s) = \sum_{z^{(b_0)},\dots,z^{(b_m)}} \pi(a|f_\phi(z^{(b_0)}, \dots, z^{(b_m)}; s)) \prod_{i=0}^m \pi(z^{(b_i)}|s^{(b_i)}) \qquad (9)$$

*where $\prod_{i=0}^m \pi(z^{(b_i)}|s^{(b_i)}) = \pi(z^{(b_0)}, \dots, z^{(b_m)}|s)$. The second equivalence in Equation 9 holds because $f_\phi$ is assumed to be deterministic.*

**Assumption 4.** *For any state $s$, and for any two policies $\pi$ and $\pi'$, we assume that if $\forall a \in \mathcal{A}(s), \pi(a|s) = \pi'(a|s)$, then $\forall z \in \mathbb{R}^d, \pi(z|s) = \pi'(z|s)$, and conversely.*

**Theorem 4.** *Let $J(\pi, \mathcal{P}, \mu)$ and $J(\pi_{f_\phi}, \mathcal{P}_z, \mu_z)$ denote the objective functions (expected returns) in SDMDP and LS-SDMDP, respectively. By Theorem 3 and Assumptions 3 4, it follows that the values of the objective functions are equal at their respective optimal policies, formally written as $J(\pi^*, \mathcal{P}, \mu) = J(\pi_{f_\phi}^*, \mathcal{P}_z, \mu_z)$. Moreover, the value functions of SDMDP and LS-SDMDP at their respective optimal policies satisfy the following relationship $V^{\pi^*,\mathcal{P}}(s) = \mathbb{E}_{z^{(b_0)} \sim \pi^{(b_0)*}} \cdots \mathbb{E}_{z^{(b_m)} \sim \pi^{(b_m)*}} V^{\pi_{f_\phi}^*,\mathcal{P}_z}(z^{(b_0)}, \dots, z^{(b_m)}; s)$.*

*Proof.* By definition, for any given policy $\pi$, the objective function $J(\pi, \mathcal{P}, \mu)$ within the SDMDP framework can be reformulated as shown in Equation 10. In this reformulation, Equation ① is derived by incorporating Equation 9. Subsequently, Equation ① is rearranged to yield Equation ②.

$$J(\pi, \mathcal{P}, \mu) = \mathbb{E}_{s \sim \mu} \mathbb{E}_{\tau \sim (\pi, \mathcal{P})} \left[ \sum_{t=0}^{T-1} \gamma^t r_t | s_0 = s \right]$$

$$= \sum_{s_0, a_0, \ldots, s_T} \mu(s_0) \prod_{t=0}^{T-1} \pi(a_t | s_t) \mathcal{P}(s_{t+1} | s_t, a_t) \sum_{t=0}^{T-1} \gamma^t r_t$$

$$\overset{①}{=} \sum_{s_0, a_0, \ldots, s_T} \mu(s_0) \prod_{t=0}^{T-1} \sum_{z_t^{(b_0)}, \ldots, z_t^{(b_m)}} \pi(a_t | f_\phi(z_t^{(b_0)}, \ldots, z_t^{(b_m)}; s_t))$$

$$\pi(z_t^{(b_0)}, \ldots, z_t^{(b_m)} | s_t) \mathcal{P}(s_{t+1} | s_t, a_t) \sum_{t=0}^{T-1} \gamma^t r_t$$

$$\overset{②}{=} \sum_{s_0, \ldots, s_T} \sum_{z_0^{(b_0)}, \ldots, z_0^{(b_m)}} \cdots \sum_{z_T^{(b_0)}, \ldots, z_T^{(b_m)}} \mu(s_0) \prod_{t=0}^{T-1} \pi(a_t | f_\phi(z_t^{(b_0)}, \ldots, z_t^{(b_m)}; s_t))$$

$$\pi(z_t^{(b_0)}, \ldots, z_t^{(b_m)} | s_t) \mathcal{P}(s_{t+1} | s_t, a_t) \sum_{t=0}^{T-1} \gamma^t r_t$$

$$(10)$$

$$J(\pi^*, \mathcal{P}, \mu) \overset{①}{=} \sum_{s_0, a_0, \ldots, s_T} \sum_{z_0^{(b_0)}, \ldots, z_0^{(b_m)}} \cdots \sum_{z_T^{(b_0)}, \ldots, z_T^{(b_m)}} \mu(s_0) \pi^*(z_0^{(b_0)}, \ldots, z_0^{(b_m)} | s_0)$$

$$\prod_{t=0}^{T-1} \pi^*(a_t | f_\phi(z_t^{(b_0)}, \ldots, z_t^{(b_m)}; s_t)) \pi^*(z_{t+1}^{(b_0)}, \ldots, z_{t+1}^{(b_m)} | s_{t+1}) \mathcal{P}(s_{t+1} | s_t, a_t) \sum_{t=0}^{T-1} \gamma^t r_t$$

$$\overset{②}{=} \sum_{s_0, a_0, \ldots, s_T} \sum_{z_0^{(b_0)}, \ldots, z_0^{(b_m)}} \cdots \sum_{z_T^{(b_0)}, \ldots, z_T^{(b_m)}} \mu(s_0) \prod_{i=0}^{m} \pi^*(z_0^{(b_i)} | s_0^{(b_i)})$$

$$\prod_{t=0}^{T-1} \pi^*(a_t | f_\phi(z_t^{(b_0)}, \ldots, z_t^{(b_m)}; s_t)) \prod_{i=0}^{m} \pi^*(z_{t+1}^{(b_i)} | s_{t+1}^{(b_i)}) \mathcal{P}(s_{t+1} | s_t, a_t) \sum_{t=0}^{T-1} \gamma^t r_t$$

$$\overset{③}{=} \sum_{s_0, a_0, \ldots, s_T} \sum_{z_0^{(b_0)}, \ldots, z_0^{(b_m)}} \cdots \sum_{z_T^{(b_0)}, \ldots, z_T^{(b_m)}} \mu(s_0) \prod_{i=0}^{m} \pi^{(b_i)*}(z_0^{(b_i)} | s_0^{(b_i)})$$

$$\prod_{t=0}^{T-1} \pi^*(a_t | f_\phi(z_t^{(b_0)}, \ldots, z_t^{(b_m)}; s_t)) \prod_{i=0}^{m} \pi^{(b_i)*}(z_{t+1}^{(b_i)} | s_{t+1}^{(b_i)}) \mathcal{P}(s_{t+1} | s_t, a_t) \sum_{t=0}^{T-1} \gamma^t r_t$$

$$\overset{④}{=} \sum_{s_0, a_0, \ldots, s_T} \sum_{z_0^{(b_0)}, \ldots, z_0^{(b_m)}} \cdots \sum_{z_T^{(b_0)}, \ldots, z_T^{(b_m)}} \mu_z(z_0)$$

$$\prod_{t=0}^{T-1} \pi^*(a_t | f_\phi(z_t^{(b_0)}, \ldots, z_t^{(b_m)}; s_t)) \mathcal{P}_z(z_{t+1} | s_t, a_t) \sum_{t=0}^{T-1} \gamma^t r_t$$

$$(11)$$

As shown in Equation 11, the optimal policy $\pi^*$ is substituted into the expression. Equation ① holds because the action taken at the terminal state $s_T$, denoted as $\pi(z_T^{(b_0)}, \ldots, z_T^{(b_m)} | s_T)$, does not influence the value of the objective function $J(\pi, \mathcal{P}, \mu)$. Equation ② is derived based on the conditional independence of the basis states $(s_t^{(b_0)}, \ldots, s_t^{(b_m)})$. Based on the conclusion of Theorem 3, we derive the following results: 1) In the $b_0$-th basis task, for all $s_t^{(b_0)} \in \mathcal{S}_t^{(b_0)}$ and $a_t \in \mathcal{A}(s_t^{(b_0)})$, where $0 \leq t \leq T$, it holds that $\pi^*(a_t | s_t^{(b_0)}) = \pi^{(b_0)*}(a_t | s_t^{(b_0)})$. Please note that $b_0$ is fixed to 0 within the SDMDP

framework. 2) Similarly, in the $b_i$-th basis task ($i \geq 1$), for all $(s_t^{(b_0)}, s_t^{(b_i)}) \in \mathcal{S}_t^{(b_0)} \times \mathcal{S}_t^{(b_i)}$ and $a_t \in \mathcal{A}((s_t^{(b_0)}, s_t^{(b_i)}))$, where $0 \leq t \leq T$, it holds that $\pi^*(a_t | (s_t^{(b_0)}, s_t^{(b_i)})) = \pi^{(b_i)*}(a_t | (s_t^{(b_0)}, s_t^{(b_i)}))$. Furthermore, in accordance with Assumption 4, we have $\pi^*(z_t | s_t^{(b_0)}) = \pi^{(b_0)*}(z_t | s_t^{(b_0)})$ and $\pi^*(z_t | (s_t^{(b_0)}, s_t^{(b_i)})) = \pi^{(b_i)*}(z_t | (s_t^{(b_0)}, s_t^{(b_i)}))$ for all $i \geq 1$. According to Assumption 3, for each policy, including the basis policy, there exists a mixture function $f_\phi$ that provides a bijective and deterministic mapping from the basis state embeddings to the corresponding state embedding. Thus, the state embedding $z_t$ is given by $z_t = f_\phi(z_t^{(b_0)}, z_t^{(b_i)}; (s_t^{(b_0)}, s_t^{(b_i)}))$ under both $\pi^*$ and $\pi^{(b_i)*}$. Consequently, we conclude that $\pi^*(z_t^{(b_0)} | s_t^{(b_0)}) \pi^*(z_t^{(b_i)} | s_t^{(b_i)}) = \pi^{(b_i)*}(z_t^{(b_0)} | s_t^{(b_0)}) \pi^{(b_i)*}(z_t^{(b_i)} | s_t^{(b_i)})$. Given the preceding results, we additionally assume that $\pi^*(z_t^{(b_i)} | s_t^{(b_i)}) = \pi^{(b_i)*}(z_t^{(b_i)} | s_t^{(b_i)})$, which directly supports the Equation ③. Equation ④ is obtained by incorporating the definitions of the initial state distribution and the transition probability function within the LS-SDMDP framework.

It is evident that the policy $\pi^*(a_t | f_\phi(z_t^{(b_0)}, \ldots, z_t^{(b_m)}; s_t))$ is the optimal policy for the objective function $J(\pi_{f_\phi}, \mathcal{P}_z, \mu_z)$. This is because, if this were not the case, $\pi^*$ would not qualify as the optimal policy for the objective function $J(\pi, \mathcal{P}, \mu)$. Thus, we conclude that $\pi^*(a_t | f_\phi(z_t^{(b_0)}, \ldots, z_t^{(b_m)}; s_t)) = \pi_{f_\phi}^*(a_t | z_t^{(b_0)}, \ldots, z_t^{(b_m)}; s_t)$ and $J(\pi^*, \mathcal{P}, \mu) = J(\pi_{f_\phi}^*, \mathcal{P}_z, \mu_z)$.

By the Bellman equation and the policy decomposition in Equation 9, the value function of SDMDP at the optimal policy, for each time step $0 \leq t \leq T-1$, can be expressed as follows:

$$
\begin{aligned}
V^{\pi^*, \mathcal{P}}(s_t) &= \mathbb{E}_{a_t \sim \pi^*(a_t | s_t)}[r_t + \gamma \mathbb{E}_{s_{t+1} \sim \mathcal{P}(s_{t+1} | s_t, a_t)} V^{\pi^*, \mathcal{P}}(s_{t+1})] \\
&= \mathbb{E}_{a_t \sim \pi^*(a_t | f_\phi(z_t^{(b_0)}, \ldots, z_t^{(b_m)}; s_t))} \mathbb{E}_{z_t^{(b_0)}, \ldots, z_t^{(b_m)} \sim \pi^*(z_t^{(b_0)}, \ldots, z_t^{(b_m)} | s_t)} \\
&\quad [r_t + \gamma \mathbb{E}_{s_{t+1} \sim \mathcal{P}(s_{t+1} | s_t, a_t)} V^{\pi^*, \mathcal{P}}(s_{t+1})] \\
&= \mathbb{E}_{z_t^{(b_0)} \sim \pi^*(z_t^{(b_0)} | s_t)} \cdots \mathbb{E}_{z_t^{(b_m)} \sim \pi^*(z_t^{(b_m)} | s_t)} \mathbb{E}_{a_t \sim \pi^*(a_t | f_\phi(z_t^{(b_0)}, \ldots, z_t^{(b_m)}; s_t))} \\
&\quad [r_t + \gamma \mathbb{E}_{s_{t+1} \sim \mathcal{P}(s_{t+1} | s_t, a_t)} V^{\pi^*, \mathcal{P}}(s_{t+1})]
\end{aligned}
\tag{12}
$$

Likewise, the value function of LS-SDMDP at its optimal policy can be written as:

$$
\begin{aligned}
V^{\pi_{f_\phi}^*, \mathcal{P}_z}(z_t^{(b_0)}, \ldots, z_t^{(b_m)}; s_t) &= \mathbb{E}_{a_t \sim \pi_{f_\phi}^*(a_t | z_t^{(b_0)}, \ldots, z_t^{(b_m)}; s_t)}[r_t + \gamma \mathbb{E}_{s_{t+1} \sim \mathcal{P}(s_{t+1} | s_t, a_t)} \\
&\quad \mathbb{E}_{z_{t+1}^{(b_0)} \sim \pi^{(b_0)*}} \cdots \mathbb{E}_{z_{t+1}^{(b_m)} \sim \pi^{(b_m)*}} V^{\pi_{f_\phi}^*, \mathcal{P}_z}(z_{t+1}^{(b_0)}, \ldots, z_{t+1}^{(b_m)}; s_{t+1})]
\end{aligned}
\tag{13}
$$

We use the inductive method to prove the relationship between the value functions at their respective optimal policies. At the time step $T-1$, since the value function at time step $T$ is equal to 0, the value functions defined in Equation 12 and Equation 13 can be expressed as follows:

$$
\begin{aligned}
V^{\pi^*, \mathcal{P}}(s_{T-1}) &= \mathbb{E}_{z_{T-1}^{(b_0)} \sim \pi^*(z_{T-1}^{(b_0)} | s_{T-1})} \cdots \mathbb{E}_{z_{T-1}^{(b_m)} \sim \pi^*(z_{T-1}^{(b_m)} | s_{T-1})} \\
&\quad \mathbb{E}_{a_{T-1} \sim \pi^*(a_{T-1} | f_\phi(z_{T-1}^{(b_0)}, \ldots, z_{T-1}^{(b_m)}; s_{T-1}))}[r_{T-1}] \\
V^{\pi_{f_\phi}^*, \mathcal{P}_z}(z_{T-1}^{(b_0)}, \ldots, z_{T-1}^{(b_m)}; s_{T-1}) &= \mathbb{E}_{a_{T-1} \sim \pi_{f_\phi}^*(a_{T-1} | z_{T-1}^{(b_0)}, \ldots, z_{T-1}^{(b_m)}; s_{T-1})}[r_{T-1}]
\end{aligned}
\tag{14}
$$

It is evident that the two value functions satisfy the relationship at time step $T-1$. We now assume that at any time step $t+1$, the relationship between these value functions holds. Substituting this assumption into the value function of SDMDP defined in Equation 12, yields the following:

$$
\begin{aligned}
V^{\pi^*, \mathcal{P}}(s_t) &= \mathbb{E}_{z_t^{(b_0)} \sim \pi^*(z_t^{(b_0)} | s_t)} \cdots \mathbb{E}_{z_t^{(b_m)} \sim \pi^*(z_t^{(b_m)} | s_t)} \mathbb{E}_{a_t \sim \pi^*(a_t | f_\phi(z_t^{(b_0)}, \ldots, z_t^{(b_m)}; s_t))} \\
&\quad [r_t + \gamma \mathbb{E}_{s_{t+1} \sim \mathcal{P}(s_{t+1} | s_t, a_t)} \mathbb{E}_{z_{t+1}^{(b_0)} \sim \pi^{(b_0)*}} \cdots \mathbb{E}_{z_{t+1}^{(b_m)} \sim \pi^{(b_m)*}} \\
&\quad V^{\pi_{f_\phi}^*, \mathcal{P}_z}(z_{t+1}^{(b_0)}, \ldots, z_{t+1}^{(b_m)}; s_{t+1})]
\end{aligned}
\tag{15}
$$

We thus conclude that the relationship between the value functions at their optimal policies is given by $V^{\pi^*, \mathcal{P}}(s) = \mathbb{E}_{z^{(b_0)} \sim \pi^{(b_0)*}} \cdots \mathbb{E}_{z^{(b_m)} \sim \pi^{(b_m)*}} V^{\pi_{f_\phi}^*, \mathcal{P}_z}(z^{(b_0)}, \ldots, z^{(b_m)}; s)$. $\qquad \square$

# D  Related Works

In this section, we first survey advances in task-specific neural solvers, which form the foundation for multi-task approaches. Next, we review multi-task learning techniques for VRPs. Finally, we examine recent progress in mixture-of-specialized-experts (MoSE) methods, which inspire key implementation aspects of our method.

**Task-Specific Neural VRP Solvers.** Learning-based neural solvers, individually developed for each specific VRP, fall into three main categories: constructive methods, iterative methods, and divide-and-conquer methods. *Constructive methods* craft end-to-end neural solvers to progressively infer solutions through autoregressive mechanisms. Pointer Network [67] pioneers this paradigm by effectively solving the small-scale traveling salesman problem (TSP), while Attention Model (AM) [34] emerges as the dominant architecture for subsequent task-specific neural solvers trained via reinforcement learning (RL). Some approaches consider properties inherent in TSP and capacitated VRP (CVRP), including the multiple optima [35] and the symmetry [33], to improve the solution quality. To further improve the cross-scale or cross-distribution generalization, advanced techniques, such as meta-learning [51, 60, 84], knowledge distillation [4], or ensemble learning [17, 22, 30], have been successfully transferred from other domains. Additionally, neural solvers trained via supervised learning (SL) also exhibit strong generalization capabilities [15, 46, 47]. *Iterative methods* leverage local search operators to consistently refine solutions until convergence. Specifically, L2I [45] and NeuRewriter [7] train RL policies to select among handcrafted operators for solution improvement. NLNS [24] alternates between heuristic destroy operators and learned repair policies to generate a new solution. DACT [50] advances beyond prior works by learning expressive representations for RL policies, while Neural-LKH [74] and Neural k-opt [49] specialize in k-opt algorithms by using RL policies to guide edge exchanges. However, these methods universally trade inference efficiency for solution quality with the aid of manually-designed operators. *Divide-and-Conquer methods* decompose problem instances into smaller sub-instances that are solved independently. Prior works have attempted to solve larger instances using pretrained neural solvers on heuristically sampled sub-instances [16, 32, 8]. By comparison, L2D [39] and RGB [85] learn RL policies to select subgraphs from heuristic-generated candidates. Unlike these heuristic-based methods, TAM [25], GLOP [77], UDC [82], and HLGP [56] opt to learnable RL policies to globally partition entire instances into subproblems, which are then solved by pretrained local construction policies. However, these methods critically depend on the partition policy, where even minor performance degradation may significantly impair the overall VRP solution quality.

**Multi-Task Learning for VRPs.** To cope with practical scenarios involving multiple VRP variants, the efficient transfer learning has been leveraged to obtain specialized neural solvers by considering inherent similarities among these variants. Lin et al. [41] propose a modular architecture consisting of a backbone pretrained on a canonical VRP and lightweight adapters inserted into the frozen backbone for the problem-specific fine-tuning. Likewise, GOAL [14] extend this paradigm to general COPs, along with fewer adapters. Corrêa et al. [11] adopt full-parameter fine-tuning strategy for each downstream VRP variant, yielding multiple problem-specific but parameter-inefficient neural solvers. However, these methods struggle with the combinatorial explosion of VRP variants. Notably, although the LoRA adapter is used for problem-specific adaptations in [41], its potential as part of a unified solver to handle the exponential growth of variants remains unexplored.

As an alternative, unified solvers are designed to handle multiple VRP variants simultaneously while eliminating the need for adapters or backbone duplications. Liu et al. [42] pioneer a single-solver framework that unifies variants via attribute compositions. MVMoE [83] enhances capacity with mixture-of-expert (MoE) layers that implicitly specialize for different variants. RouteFinder [3] employs modified Transformer model with mixed batch training for stable convergence. UNCO [29] resorts to the large language model (LLM) for the expressive instance and task embeddings. CaDA [38] utilizes a dual attention mechanism for superior cross-problem capabilities. Goh et al. [20] considers the more practical multi-task, multi-distribution setting by using mixture-of-depths (MoD) and context-based clustering. Liu et al. [43] leverage mixed-curvature spaces in the feature fusion stage such that the model's encoder can capture the geometric structures inherent in VRP instances. Lei et al. [37] strengthen cross-scale and cross-problem generalization capabilities of diffusion-based solvers during inference, which is orthogonal to our work. In contrast, Wang & Yu [68] decompose the model into a shared encoder, and problem-specific headers and decoders, applying multi-armed bandits for dynamic task sampling during training. Li et al. [40] adopt the same architecture with aligned

optimization directions across tasks. However, these methods fail to fully leverage the compositional structure inherent in VRP variants, each derived from a common set of basis VRP variants. Thus, the potential benefits of incorporating explicitly specialized basis solvers remain unexplored. Notably, while MoE layers are employed in [83], these learn implicit and less interpretable specialization rather than incorporating off-the-shelf basis solvers as experts.

**Mixture of Specialized Experts.** Prevailing methods focusing on mixture of specialized experts (MoSE) can be broadly categorized into two paradigms: merging entire models and module composition. Approaches based on merging entire models seek to combine independently trained models to efficiently achieve the performance comparable to model ensembling or multi-task learning. Most approaches are developed based on the shared model architecture. Wortsman et al. [70] exhibit averaging the parameters of models trained with different hyperparameter configurations improves accuracy. Matena & Raffel [52] develop a model aggregation framework guided by Fisher information. Yadav et al. [75] address interference effects in model merging to preserve critical knowledge. Tam et al. [64] formulate model fusion as a linear equation system solved through conjugate gradient optimization. Daheim et al. [12] introduce a uncertainty-based merging scheme to reduce the gradient mismatch. In contrast, several approaches enable merging across architecturally distinct models. Ainsworth et al. [1] align weights of different models to enable the model merging. Stoica et al. [63] introduce cross-task model fusion through feature-space integration. Additionally, weighted merging strategies have also been extensively explored. Ramé et al. [61] implement weighted model fusion to enhance out-of-distribution generalization, while Jin et al. [31] develop parameter-space merging optimized for multi-task performance through learned weighting schemes. Yang et al. [76] further advance this paradigm by deriving merging coefficients via unsupervised entropy minimization. However, these approaches struggle with achieving precise layer-wise and token-wise aggregation within expert models, limiting their multi-task OOD generalization.

Our implementation aligns more closely with the module composition paradigm which supports the finer-grained aggregation. The most straightforward approach involves parameter averaging of task adapters. Chronopoulou et al. [9] demonstrate test-time averaging of relevant adapters for task adaptation, while Ponti et al. [59] propose averaging adapter parameters selected by a routing function for new tasks. Beyond simple averaging, arithmetic operations have proven effective for the adapter composition. Chronopoulou et al. [10] demonstrate that basic arithmetic combinations enhance zero-shot cross-lingual transfer, while Zhang et al. [81] develop specialized arithmetic operations for lightweight adapters to boost generalization. Similarly, Ilharco et al. [28] employ arithmetic combinations of task vectors to precisely steer model behavior for novel tasks. Task similarity further facilitates the effective module composition. Lv et al. [48] develop weighted aggregation of parameter-efficient adapters based on inter-task similarity measures for novel tasks. The MoCLE [21] framework addresses task conflicts and improves generalization by activating task-customized LoRA adapters based on clustered instructions and using a trainable universal adapter. Similarly, Wu et al. [72] propose modality-agnostic task similarity measures to combine lightweight adapters for enhanced performance on multimodal downstream tasks. Adaptive gating mechanisms have emerged as the predominant approach for dynamic model composition. LoRAHub [27] employs few-shot examples to compute weighting coefficients for pre-trained LoRA modules, enabling competitive performance on novel tasks. Building on this, Ye et al. [78] develop a hierarchical routing mechanism that selects optimal Transformer layers for enhanced task generalization. The MoLE [73] framework advances this paradigm through layer-wise gating functions that learn to optimally combine LoRA experts, with these operations being implemented via dedicated Transformer blocks. AdaMoLE [44] further incorporates dynamic threshold adaptation to handle varying task complexities. Caccia et al. [5] introduce multi-head adapter routing, enabling fine-grained routing for the cross-task generalization. Module composition approaches also effectively address catastrophic forgetting. LoRAMoE [13] mitigates world knowledge degradation in frozen backbone LLMs through strategic integration of multiple LoRA modules. Similarly, AdapterFusion [58] combats catastrophic forgetting in multi-task learning scenarios by dynamically combining pretrained adapter modules. Furthermore, module composition techniques have also demonstrated significant potential for zero-shot generalization. PHATGOOSE [53] achieves this through inference-time aggregation of LoRA adapters via their pretrained gating functions. Similarly, Ostapenko et al. [55] develop a zero-shot routing mechanism that dynamically selects the most task-relevant adapters based on task similarity, eliminating the need for retraining. In addition, Zadouri et al. [79] advance efficient adaptation through fine-tuning of mixture of lightweight LoRA experts for limited computational cost scenarios. Gao et al. [18]

uncover important architectural insights: different layers benefit from varying numbers of LoRA modules, and higher network layers particularly require more experts to maintain performance.

