# OpenReview forum: "Multi-Task Vehicle Routing Solver via Mixture of Specialized Experts under State-Decomposable MDP"
_NeurIPS.cc/2025/Conference — NeurIPS 2025 poster_

### Official Review · Reviewer_jY9o · 2025-06-12

**Clarity:** 2
**Significance:** 3
**Originality:** 3
**Rating:** 4
**Confidence:** 3

**Summary:**

This paper proposes the State-Decomposable Markov Decision Process(SDMDP) to solve the multi-task vehicle routing problem(VRP). It decomposes the state of the problem into a combination of multiple basic state Spaces and combines the Mixture of Specialized Experts(MoSE) to solve the VRP problem and various VRP variants.

**Questions:**

1.Could you provide the specific expressions and answers of the SDMDP you proposed in the VRPTW and Generalized TSP problems? If there is not enough space for you, please answer the first one.
2.Could you provide comparative experiments for more larger-scale problems?
3.What are the reasons and usages for setting these two assumptions?

**Ethical Concerns:**

["NO or VERY MINOR ethics concerns only"]

**Final Justification:**

Since the concerns have been addressed, I decide to raise my score.

**Limitations:**

The increase in the time cost of MoSES is really too large. It is very important to consider how to maintain the effect while reducing the time cost.

**Paper Formatting Concerns:**

No paper formatting concerns.

**Quality:**

2

**Strengths And Weaknesses:**

Strengths:
1, The SDMDP proposed in this paper fundamentally provides a unified expression of the variants of the VRP problem, which ensures that the proposed method can solve multiple variants of VRP.
2.The effectiveness of up to 16 VRP variants was verified, demonstrating the universality of the proposed model.
3.The proposed MoSEs can be applied to a variety of models, Meanwhile, the calculation method of hyperparameters is given.
Weaknesses:
1, The full text only presents the form of SDMDP, which might make it difficult for readers to understand why your SDMDP can summarize the variants of VRP. Maybe you can add specific examples at appropriate positions and explain how to construct its MDP using your model.
2.At the experimental results level, the model with MoSES took more than twice as long as the model without MoSES to achieve an absolute Gap decrease of less than 0.4\% on average and a relative decrease ranging from 1/10 to 1/4. It seems that the extent of performance optimization is not necessarily worth using. Perhaps expanding the scale of the problem would be more persuasive.
3.You didn't provide in the main text how to prove Theorem 1 and Theorem 2. If you prove it in the appendix, providing guidance seems to be a better solution. Besides, the reasons for setting these two assumptions are also unclear. I'm not quite clear about what the functions of these two assumptions are respectively?

Overall, I think the proposal of SDMDP and LS-SDMDP is interesting and inspiring, but the effect of MoSES is not ideal, which limits the contribution of this article.

---

> ### Author Rebuttal · Authors · 2025-07-31
>
> We sincerely appreciate the reviewers’ insightful comments and constructive feedback on our work. We would like to take this opportunity to address your concerns and clarify any potential misunderstandings.
>
> **Weakness 1: example of SDMDP**
>
> We sincerely apologize that, due to page limitations, we were unable to include an example to help readers better understand how SDMDP can be used to unify VRP variants.  Under SDMDP, there are five types of basic state spaces: $\mathcal{S}^{(0)}, \ldots, \mathcal{S}^{(4)}$. $\mathcal{S}^{(0)}$ encodes node coordinates, linehaul demands, and remaining linehaul capacity, $\mathcal{S}^{(1)}$ represents a binary variable indicating whether the vehicle needs to return to the depot; $\mathcal{S}^{(2)}$ captures backhaul demands and remaining backhaul capacity; $\mathcal{S}^{(3)}$ includes the duration limit and the current traveled length of the subtour; and $\mathcal{S}^{(4)}$ contains time windows, service times, and the current time. Before each episode begins, the initial state space is constructed by selecting $\mathcal{S}^{(0)}$ and any subset of the remaining basis states. In this example, we choose $\mathcal{S}^{(0)}$, $\mathcal{S}^{(3)}$ and $\mathcal{S}^{(4)}$, which correspond to the VRP with limited duration and time windows (VRPLTW). At each time step, the policy observes the basis states $s^{(0)}$, $s^{(3)}$, and $s^{(4)}$, and a masking mechanism is applied to filter out infeasible nodes from unvisited nodes, specifically, those whose demands exceed the remaining linehaul capacity, those that would violate the duration limit, and those whose arrival time would exceed the end of their time window, from unvisited nodes. This results in a list of feasible nodes. Upon selection, the linehaul capacity, traveled length, and current time are updated accordingly, and the selected node is masked out for subsequent steps. Once a subtour is completed, the linehaul capacity is reset to its default value, the traveled length is reset to zero, and the current time is also reset to zero. This reset reflects the assumption in VRP that multiple vehicles can begin their routes concurrently.
>
> **Weakness 2: empirical improvement**
>
> We would like to clarify that the empirical improvements achieved by our method are sufficiently favorable, and it is well-recognized in the multi-task VRP literature that achieving even small improvements across all 16 tasks is highly challenging. In terms of average optimality gap, MoSES(RF) achieves 1.535\% compared to RF-TE’s 2.063\% for $N=50$, and 2.782\% compared to RF-TE’s 3.125\% for $N=100$. MoSES(CaDA) achieves 1.515\% versus CaDA’s 1.715\% for $N=50$, and 2.631\% versus CaDA’s 2.766\% for $N=100$. In the context of multi-task VRP for both $N=50$ and $N=100$, the optimality gap is already small in magnitude; therefore, the absolute decreases achieved by MoSES(RF), 0.528\% for $N=50$ and 0.343\% for $N=100$, and by MoSES(CaDA), 0.200\% for $N=50$ and 0.135\% for $N=100$, already represent noticeable improvements. Thus, to fairly and appropriately evaluate the improvement when the evaluation score is of small magnitude, relative improvement over the baseline method is a more suitable metric. MoSES(RF) achieves relative performance improvements of 25.6\% and 11.0\% over its baseline for $N=50$ and $N=100$. MoSES(CaDA) achieves relative improvements of 11.7\% and 4.9\% over its baseline for $N=50$ and $N=100$. Thus, based on this clarification, we believe the empirical performance of our method is substantially favorable compared to prior baseline methods, thereby demonstrating a noticeable empirical contribution.
>
> **Weakness 3: theorem proofs and assumptions**
>
> We sincerely apologize for the inconvenience caused by the omission of detailed theorem proofs in the main text due to page limitations. As noted in the remark following Theorem 2, the proofs are included in **the Appendix**. In the revised version, we will provide clearer guidance for each theorem to explicitly indicate that the corresponding proofs can be found in the Appendix. We also apologize for the confusion arising from the use of the two assumptions. Please refer to our responses in **Question 3**.
>
> **Question 1: SDMDP expressions**
>
> To formulate VRPTW using SDMDP, we first define the basis state spaces. Please refer to our responses in **Weakness 1** for the definitions. Before each episode begins, the initial state space is constructed by selecting $\mathcal{S}^{(0)}$ and $\mathcal{S}^{(4)}$, which correspond to the VRP with time windows (VRPTW). At each time step, the policy observes the basis states $s^{(0)}$ and $s^{(4)}$, and a masking mechanism is applied to filter out infeasible nodes from the set of unvisited nodes, specifically, those whose demands exceed the remaining linehaul capacity and those whose arrival time would exceed the end of their time window. This results in a list of feasible nodes. Upon selection, the linehaul capacity and current time are updated accordingly, and the selected node is masked out. Once a subtour is completed, the linehaul capacity is reset to its default value, and the current time is reset to zero.
>
> To formulate the Generalized TSP (GTSP) using SDMDP, we need to make adjustments to the formulation presented in our paper, as our work uses CVRP as the basis VRP variant from which other variants are derived by adding constraints. For GTSP, the basis state space $\mathcal{S}^{(0)}$ is associated with node coordinates, while each additional constraint is represented by one of the basis state spaces $\mathcal{S}^{(1)}, \ldots, \mathcal{S}^{(5)}$, resulting in six basis state spaces in total. Before each episode begins, the initial state space is constructed by selecting only $\mathcal{S}^{(0)}$. At each time step, the policy observes the basis state $\mathcal{s}^{(0)}$ and selects one unvisited node from the currently active node groups. Upon selection, the chosen node and all other nodes in the same group are masked out for subsequent steps.
>
> **Question 2: large-scale problems**
>
> We would like to emphasize that multi-task VRP is already a highly challenging problem due to the need to address multiple types of VRP variants simultaneously. As a result, current research, including the baseline methods used in our work, primarily focuses on small-scale problem instances. This is a common and necessary step in combinatorial optimization, where improving performance on small-scale problems often serves as a foundation for developing more advanced methods capable of handling large-scale instances. Therefore, the primary focus of our paper remains on solving multi-task VRPs effectively. The extension to large-scale scenarios is an important direction, but we consider it future work.
>
> To demonstrate the generalization capability of our method, we also evaluate it on larger problem instances with up to 1,000 nodes from the CVRPLIB dataset, as presented in the Appendix. In this setting, both MoSES(RF) and MoSES(CaDA) outperform their baselines. In the table below, we present a performance comparison on large-scale problem instances from CVRPLIB. For large-scale instances with 501<N≤1001, MoSES(CaDA) achieves an absolute decrease of 1.385% compared to CaDA. While MoSES(RF) shows a smaller absolute decrease of 0.744% compared to RF, this result is still acceptable when considering that RF itself achieves an absolute decrease of 0.935% compared to its baseline RF-POMO.
>
> | graph size | RF-POMO |RF-TE | CaDA | MoSES(RF) | MoSES(CaDA) |
> |----------  |---------|---------|---------|---------|---------|
> |N<251       |5.905%|5.061% |  4.772% | 4.789% | 4.724% |
> |251<=N<501  |8.339%|8.107% |  8.889% | 7.741% | 7.948% |
> |501<N<=1001 |13.188%|12.253%|  14.199%|11.509% |12.814% |
> |Average     |9.108%|8.428% |   9.230%|7.973%  |8.441%  |
>
> **Questions 3: assumptions**
>
> Both assumptions are critical for Theorem 2, which guarantees that LS-SDMDP exactly recovers SDMDP’s optimal policy when the assumptions hold. Assumption 1 ensures policy decomposition and mixture function existence, aligning with LS-SDMDP’s structure, while Assumption 2 enforces a unique state-to-embedding mapping, necessary for the theorem’s validity. Together, they provide the theoretical foundation for policy recovery.
>
> **Limitation: time overhead**
>
> In terms of the average total time overhead for 1,000 problem instances across 16 tasks, MoSES(RF) requires 5.8s compared to RF’s 2.0s for $N=50$, and 20.8s compared to RF’s 8.4s for $N=100$. MoSES(CaDA) requires 7.4s compared to CaDA’s 2.3s for $N=50$, and 24.8s compared to CaDA’s 10.5s for $N=100$. We would like to emphasize that this time overhead is the total time for 1,000 instances per task; the amortized time per instance remains at the microsecond level for both MoSES(RF) and MoSES(CaDA). In the VRP domain, it is generally acceptable to increase inference time for better solution quality within a given time budget. In practice, many industrial CO tasks allow for extended times to obtain high-quality solutions [1]. Thus, several SOTA VRP methods [2,3] commonly use search-based method or solution refinement. Compared to the time increase introduced by such techniques, the additional overhead from the network architecture changes is relatively minor. We thus believe that the increased time overhead is acceptable given the favorable empirical improvements and may not be a major concern in practical applications.
>
>
> [1] Simulation-guided Beam Search for Neural Combinatorial Optimization, NeurIPS 2022.
>
> [2] BQ-NCO: Bisimulation Quotienting for Efficient Neural Combinatorial Optimization, NeurIPS 2023.
>
> [3] Neural Combinatorial Optimization with Heavy Decoder: Toward Large Scale Generalization, NeurIPS 2023.
>
> We sincerely thank the reviewer for the insightful comments and constructive feedback, which have greatly improved conceptual clarity of our work. We will revise the manuscript accordingly to enhance clarity and address all suggestions.

---

> ### Author Response · Authors · 2025-08-07
>
> As we approach the end of the Reviewer–Author Discussion phase, we would like to confirm whether our responses have sufficiently addressed your concerns and questions. If you have any remaining questions or need further clarification, we remain actively engaged and are happy to provide additional explanations. If we have addressed your concerns, we would be grateful if you could kindly consider adjusting the rating of our paper.

---

### Official Review · Reviewer_qQzf · 2025-06-24

**Clarity:** 3
**Significance:** 3
**Originality:** 3
**Rating:** 4
**Confidence:** 3

**Summary:**

This work presents a novel approach to training a multi-task NCO model for solving vehicle routing problems using a mixture of specialized experts. It introduces a new State-Decomposable MDP formulation for VRP, where the state space is decomposed into independent basis states and extended with a learnable mixture function to enable policy reuse in the latent space. Through experiments, the authors demonstrate that they can efficiently train optimal policies using specialized experts and adaptive gating mechanisms. This approach can be used in a plug-and-play manner to enhance existing state-of-the-art models.

**Questions:**

1. Although the main idea is clear, the implementation details and the application to VRP could be explained more thoroughly. For me, only the basic state 0) for the classic VRP is clear, while the others are somewhat confusing. For example, basic state 1)-open route only introduces a binary variable - what would be the basic state for this case, and what exactly does it represent? Next, it is mentioned that 'each VRP variant can be formed by composing basic state spaces,' but isn't it mandatory that state 0) is always present?

2. LoRA originally comes from fine-tuning large language models (LLMs), where low-rank matrix factorization is important due to the heavy computational cost. Did you try using simple linear task-specific experts without projecting to a low-rank space? It seems your model isn't large enough to require low-rank factorization, and avoiding that might improve the results.

3. It is stated that the gated-LoRa method suppresses irrelevant features from the backbone. The backbone is trained on CVRP, and as mentioned in paragraphs 252–255, all the other basic VRP variants are derived from it. So, what exactly would be considered irrelevant features in the backbone?

4. Figure 3 is quite surprising, as it doesn't indicate which activation function performs best in general. Do you have any reasonable explanation for why sigmoid works well with CaDa but completely fails with RF? This is somewhat concerning, as it appears that a universal solution for the activation function does not exist and is dependent on the architecture, which reduces the generality of your method.

5. Have you considered applying the same mechanism to a unified NCO solver beyond routing problems? For such a truly universal solver, there is no clear backbone that can serve as a basis for other variants. Is it possible to apply this method to problems with no common intersection between states? This seems like a real limitation when it comes to extending this work to more general cases.

**Ethical Concerns:**

["NO or VERY MINOR ethics concerns only"]

**Final Justification:**

This is a technically solid paper, with some original ideas and potential for future work, but it is limited to multi-task VRPs. In my opinion, considering the current state-of-the-art in NCO models - many of which already focus on this area - this topic has a rather narrow research interest.

Therefore, I will maintain my initial score of weak acceptance.

**Limitations:**

The motivation for this work comes from making a unified model for solving vehicle routing problems, and the proposed solution is tailored to this class of CO problems. The approach does not seem easily extendable to other classes of CO problems or capable of making significant progress toward a foundation model for CO.

This appears to be a major limitation, as the work does not open new directions toward that goal - one that seems to be a key future milestone for the NCO research community.

**Quality:**

3

**Strengths And Weaknesses:**

Strengths:

- Unlike other similar works, this approach pretrains the model on a single task (CVRP) and then fine-tunes it on other VRP tasks with new problem attributes.
- The proposed State-Decomposable MDP framework is architecture-independent and can be applied to any multi-task NCO solver for VRPs.
- Experiments show that the proposed method clearly improves upon existing state-of-the-art multi-task vehicle routing NCO solvers.

Weaknesses:
- The proposed method is applicable when all tasks share a common intersection; otherwise, maintaining a shared backbone seems impossible.

---

> ### Author Rebuttal · Authors · 2025-07-31
>
> We sincerely thank you for your insightful comments and thoughtful questions regarding our work. We appreciate the opportunity to address your concerns and clarify any potential misunderstandings.
>
> **Weakness 1: applicability to other problems**
>
> We would like to clarify that our work specifically targets the multi-task VRP setting, which reflects a more practical demand in real-world applications. We believe that exploring the extension of our work represents a promising avenue for future research, but they are not the central objective of this work. From the broader perspective of multi-task learning, tasks solved simultaneously by a multi-task model are generally assumed to share underlying structures. This shared structure naturally implies the existence of interactions among tasks, which forms the foundation for positive knowledge transfer. To extend our work, one plausible approach would be to define a set of basis tasks, from which all target tasks can be derived. To illustrate this potential extension beyond multi-task VRPs, we can consider a simple example in the robotics domain. Suppose the goal is to learn a robot policy for washing a cup. The basis tasks might include picking up the cup, filling it with water, and using a cloth to clean it. Notably, the tasks of picking up the cup and filling it with water could also be components of another task, such as watering plants. This example demonstrates the potential of our framework to generalize to broader multi-task learning scenarios.
>
> **Question 1: VRP variant details**
>
> Due to page limitations, we have included detailed descriptions of the VRP variants in the **Appendix**. To clarify, the 0-th basis VRP variant corresponds to the classic CVRP with capacity constraints. The remaining four basis VRP variants are derived by adding one additional constraint to CVRP: open route, backhaul, duration limit, and time window, resulting in OVRP, VRPB, VRPL, and VRPTW, respectively. Specifically, we denote the basis state 0 as $s^{(0)}$, which represents the state of the classic CVRP. The basis state 1, $s^{(1)}$, is a binary variable to indicate whether a vehicle needs to return to the depot. Since OVRP extends CVRP by introducing an open-route constraint, its state consists of both $s^{(0)}$ and $s^{(1)}$. Similarly, basis state 2, $s^{(2)}$, is used in VRPB to represent backhaul demands and remaining backhaul capacity. The state of VRPB includes both $s^{(0)}$ and $s^{(2)}$. Basis state 3, $s^{(3)}$, is used in VRPL to encode the duration limit of a subtour and the current traveled length along the present subtour; thus, the state of VRPL consists of $s^{(0)}$ and $s^{(3)}$. Finally, basis state 4, $s^{(4)}$, is used in VRPTW to represent time windows, service durations, and the current time, making the state of VRPTW a combination of $s^{(0)}$ and $s^{(4)}$. Thus, in each VRP variant, the basis state $s^{(0)}$ is always present, as each variant is derived by adding at least one additional constraint, open route, backhaul, duration limit, or time window, to the classic CVRP.
>
> **Question 2: linear task-specific experts**
>
> We appreciate the reviewer's insightful suggestion to investigate linear task-specific experts as an alternative. To explore it, we designed an experiment where the basis VRP solvers for OVRP, VRPB, VRPL, and VRPTW are trained using linear adapters. These linear adapters are then integrated into the unified solver to enable reuse. We implemented this experiment on the MoSES(RF) model under the setting of $N=50$. The experimental results are presented below. We observe that the proposed linear expert contributes to performance improvement. We sincerely appreciate this insightful suggestion. Although our network avoids the heavy computational costs typical of LLMs, we use LoRA adapters to explore whether popular LLM techniques can benefit neural solvers for VRPs. Early research has begun applying large models to this area, and our use of LoRA in a smaller model may still yield valuable insights and help pave the way for future large-scale applications.
>
> | VRP Variant | LoRA expert | Linear expert |
> |-------------|----------------------|-------------------|
> | CVRP        | 0.900%  | 0.878%  |
> | VRPTW       | 1.445%  | 1.427%  |
> | OVRP        | 1.892%  | 1.808%  |
> | VRPL        | 1.089%  | 1.061%  |
> | VRPB        | 2.342%  | 2.217%  |
> | OVRPTW      | 0.959%  | 0.938%  |
> | VRPBL       | 3.185%  | 2.960%  |
> | VRPBLTW     | 1.370%  | 1.431%  |
> | VRPBTW      | 1.121%  | 1.145%  |
> | VRPLTW      | 1.811%  | 1.845%  |
> | OVRPB       | 1.979%  | 1.764%  |
> | OVRPBL      | 2.014%  | 1.775%  |
> | OVRPBLTW    | 0.791%  | 0.783%  |
> | OVRPBTW     | 0.783%  | 0.778%  |
> | OVRPL       | 1.917%  | 1.819%  |
> | OVRPLTW     | 0.962%  | 0.943%  |
> | **Average** | **1.535%** | **1.473%** |
>
> **Question 3: Gated-LoRA**
>
> Gated-LoRA is used to train adapters for the basis VRP variants. During the fine-tuning of each specialized LoRA adapter, the backbone model, pretrained on CVRP, is frozen, and only the LoRA adapter is updated. Importantly, the problem instances from these basis VRP variants are not seen by the backbone model during its training, as it was trained exclusively on CVRP instances. Each layer in the backbone model must process not only features of the CVRP state but also features of the additional constraints of the new variant. This mismatch can lead to the generation of task-misaligned features by the frozen backbone.We thus introduce a gating function into the backbone layers. Empirical results shown in **Figures 2(a) and 2(b)** demonstrate the effectiveness of the Gated-LoRA design.
>
> **Question 4: activation function**
>
> We believe **Figure 9 in the Appendix** helps address this concern, as it provides a behavioral analysis of the gating functions. In Figure 9, we plot the scores assigned by the gating function to each basis solver (LoRA expert). Before addressing the concern, we would like to emphasize that RF is a Transformer-based solver, while CaDA uses two parallel Transformer blocks in each layer. This more complex design gives CaDA stronger capacity. The four basis solvers adapted from RF tend to specialize in their respective VRP variants and may not generalize well to others. In contrast, CaDA-based basis solvers, due to CaDA’s stronger capacity, may generalize better across different basis VRP variants. Figure 9(a) shows that MoSES(RF), using L1-normalized softplus as its activation function, tends to assign high scores to relevant basis solvers while keeping scores low for irrelevant ones. Figure 9(b) shows that MoSES(CaDA), using sigmoid as its activation function, also prioritizes relevant solvers but assigns moderately higher scores to less relevant ones. This suggests that MoSES(CaDA) recognizes that even task-irrelevant solvers may still provide useful insights for solving the current VRP variant.
>
> Furthermore, we would like to clarify that the gating function serves as the implementation of the mixture function of LS-SDMDP. Therefore, it is entirely reasonable to have different implementations of the mixture function, and different activation functions simply reflect different design choices. Importantly, our goal is not to identify a universal activation function for all network. Since different network possess varying capacities, they naturally require different activation function designs to align with their inherent characteristics. It is common that different models often benefit from tailored activation functions. Moreover, in LLM, exploring different activation functions in gating function remains an active research direction. We believe that the choice of activation function should be guided by the architecture's design and capacity.
>
> **Question 5: application beyond VRP**
>
> Our current focus is on advancing toward a unified solver for multi-task VRPs. We believe that extending our framework to other CO problems is a promising and long-term direction for future research. To support such an extension, we would like to share a preliminary idea. First, CO problems could be grouped based on structural similarities, allowing tasks within each group to benefit from shared structure. Then, a representative basis task could be selected from each group, forming a set of basis tasks across groups. A shared backbone model could be trained on these selected tasks. This backbone would serve as a common foundation for each group, and adapter modules, along with their compositions, could be integrated into it. Although this idea has not yet been validated, and our current work is specifically designed for multi-task VRPs, we believe it offers a feasible path toward building a unified solver for broader CO domains.
>
> **Limitation: transferability to other problems**
>
> Firstly, we would like to emphasize that our framework is specifically designed for the multi-task VRP domain, which has a wide range of practical applications. Extending it to more general CO scenarios represents a promising and long-term research direction. Then, as illustrated in the robotics example mentioned in "Weakness 1," the observation that certain tasks can be derived from a shared set of basis tasks is common in practice. While this may not cover all real-world scenarios, it does encompass a broad spectrum of applications, further highlighting the potential of our framework to be extended beyond multi-task VRPs to more general and practical domains. Finally, based on the preliminary idea we shared in "Question 5," our work offers a feasible path toward generalizing to broader CO problems. This direction could further demonstrate the value and adaptability of our framework.
>
> We sincerely appreciate the reviewer's insightful comments and constructive feedback, which have helped us strengthen both the technical presentation and conceptual clarity of our work. We will revise our manuscript accordingly to address your suggestions and ensure greater clarity.

---

> > ### Comment · Reviewer_qQzf · 2025-08-04
> >
> > Thank you for your detailed response.
> >
> > I understand that your framework is specifically designed for multi-task VRPs, and for me, that is its main weakness. Recently, we have seen a dozen papers on this topic, all starting from the same assumptions and offering only incremental improvements over one another.
> >
> > However, your work introduces some original ideas and has the potential to open new horizons, so I will maintain my original score.

---

> > > ### Author Response · Authors · 2025-08-04
> > >
> > > We sincerely appreciate your thoughtful comments and for recognizing our work as an original contribution with the potential to be extended to new horizons. We will revise our manuscript based on your valuable questions and suggestions to improve clarity and completeness. We remain actively engaged during the discussion period, so please feel free to reach out with any further questions or requests for clarification.

---

### Official Review · Reviewer_Tp1S · 2025-07-03

**Clarity:** 3
**Significance:** 3
**Originality:** 3
**Rating:** 5
**Confidence:** 4

**Summary:**

The proposed paper tackles the topic of multi-task vehicle routing problem solvers. Specifically, the authors reformulate the problem as a State-Decomposable MDP (SDMDP), where the state space decomposes into basis states, each corresponding to a fundamental VRP variant (e.g., capacitated, open route). They then extend the work to latent space (LS-SDMDP), proving that the mixture of basis policies can recover optimal solutions under certain conditions. Specifically, in their method, the authors assume the most basic variant of the VRP as the capacitated problem aand each subsequent variant has an additional constraint. As such they train a fixed backbone trained on CVRPs, and use LoRA experts adapt the base embeddings to more constrained settings. They call this method the Mixture-of-Specialized-Experts Solver (MoSES) and use a weighted gating mechanism to concatenate the different expert outputs.

**Questions:**

1. Seeing as the method is slower, have you tried running the direct baseline (the models you adapt) with more than 8x augmentations or using input augmentation like in [1]? The difference seems small enough that for some experiments, matching the runtime might change the model standing.

2. Can experts interact with each other outside of the gated mixing? This would seem to be important for correlated tasks.

3. For different problem sizes, is there any insight on when frozen vs trainable experts? Would this be due to better expert interaction with harder tasks?

[1] Bdeir, Ahmad, Jonas K. Falkner, and Lars Schmidt-Thieme. "Attention, filling in the gaps for generalization in routing problems." ECML, 2022.

**Ethical Concerns:**

["NO or VERY MINOR ethics concerns only"]

**Limitations:**

yes

**Quality:**

1

**Strengths And Weaknesses:**

Strengths:

The authors suggest the decomposition of state-spaces for the various VRPs, and they then prove that this decomposition enables optimal policy reuse via latent space composition (LS-SDMDP). The approach is relatively novel, and they provide practical applications of the concept through the use of MoSES. The models implemented are not novel in themselves, but this also shows the flexibility of the approach as a general multi-task framework that allows for the use of plug-and-play backbones. The method also achieves convincing empirical results (although at the cost of increased runtime), the experiments are extensive on the datasets that are studied, and a good ablation study is performed for those datasets (this is a point I mention in the weaknesses below).

Weaknesses:

The proposed method requires some base assumptions to be present. VRP variant constraints need to be easily separated into additional modeled components, and the method assumes the independence of these constraints. Any correlation between them is poorly modelled since there is minimal interaction between the experts, especially when the backbone is frozen.

Additionally, the paper does not explore the performance of the model in generated out-of-distribution datasets or in generalization to different routing problem sizes. The later is particularly important since attention-based approaches are severely constrained by small problem sizes. The method is already around 4x slower on average than the direct competitor.

---

> ### Author Rebuttal · Authors · 2025-07-31
>
> We are sincerely grateful to the reviewer for the insightful comments. We hope this response helps to address your concerns and clarify any potential misunderstandings.
>
> **Weakness 1: assumptions**
>
> We would like to clarify that we do not assume the basis VRP variants, each characterized by a specific attribute or constraint, are independent of each other. Our work is based on the observation that each VRP variant arises from a composition of attributes, with each attribute corresponding to a basis VRP variant. This observation is well-supported and has been recognized in prior works [1]. Building on this, we propose the MoSES framework, which enables a unified solver to leverage the potential benefits of basis solvers, each specialized for a specific basis VRP variant. While we acknowledge that correlations exist among the basis solvers (experts), modeling these correlations is not the focus of our work.
>
> [1] Multi-task learning for routing problem with cross-problem zero-shot generalization.
>
> **Weakness 2: OOD evaluation**
>
> We apologize for the inconvenience; due to page limitations, the out-of-distribution (OOD) evaluation has been included in the **Appendix**. In the Appendix, we first evaluate our method on CVRP with varying capacities and gain several insights. We observe that MoSES(RF) with the softmax gating function achieves better performance than its baseline RF on both in-distribution (ID) and OOD evaluations (Please note that **Figure 7(a)** in Appendix shows that MoSES(RF) with softmax performs better than RF in the ID evaluation). In contrast, MoSES(RF) with softplus tends to focus more on ID performance. MoSES(CaDA) with sigmoid also outperforms CaDA under the OOD setting. Next, we evaluate our method on VRPL with different distance limits and find that MoSES(RF) demonstrates stronger generalization capability compared to both baselines and MoSES(CaDA). Finally, we assess performance on VRPTW with varying time windows, where both MoSES(RF) and MoSES(CaDA) generally outperform their respective baselines. Importantly, we also evaluate our method on the CVRPLIB dataset, which includes larger problem instances with up to 1,000 nodes. In this setting, both MoSES(RF) and MoSES(CaDA) outperform their baselines. In the table below, we present a performance comparison evaluated on large-scale problem instances from CVRPLIB. Additionally, we would like to emphasize that multi-task VRP is already a highly challenging problem due to the need to address multiple types of VRP variants simultaneously. As a result, current research, including the baseline methods used in our work, primarily focuses on small-scale problem instances. This is a common and necessary step in combinatorial optimization, where improving performance on small-scale problems often serves as a foundation for developing more advanced methods capable of handling large-scale instances. Therefore, the primary focus of our paper remains on solving multi-task VRPs effectively. The extension to large-scale scenarios is an important direction, but we consider it future work.
>
> | graph size | RF-TE | CaDA | MoSES(RF) | MoSES(CaDA) |
> |----------  |---------|---------|---------|---------|
> |N<251  |5.061% |  4.772% | 4.789% | 4.724% |
> |251<=N<501  |8.107% |  8.889% | 7.741% | 7.948% |
> |501<N<=1001 |12.253%|  14.199%|11.509% |12.814% |
> |Avg. |8.428% |  9.230%|7.973%  |8.441% |
>
> In terms of the average total time overhead for 1,000 problem instances across 16 tasks, MoSES(RF) requires 5.8s compared to RF’s 2.0s for $N=50$, and 20.8s compared to RF’s 8.4s for $N=100$. MoSES(CaDA) requires 7.4s compared to CaDA’s 2.3s for $N=50$, and 24.8s compared to CaDA’s 10.5s for $N=100$. Although our method is generally over 2x slower than the direct baseline methods, we would like to emphasize that this time overhead is the total time for 1,000 instances per task; the amortized time per instance remains at the microsecond level for both MoSES(RF) and MoSES(CaDA). Additionally, in the VRP domain, it is generally acceptable to increase inference time in order to achieve better solution quality within a given time budget. In practice, many industrial CO tasks allow for extended computation times to obtain high-quality solutions [2]. Thus, several SOTA VRP methods [3,4] commonly use search-based method or solution refinement. Compared to the time increase introduced by such techniques, the additional overhead from the network architecture changes is relatively minor. We thus believe that the increased time overhead is acceptable given the favorable empirical improvements over prior methods and may not be a major concern in practical applications.
>
> [2] Simulation-guided Beam Search for Neural Combinatorial Optimization, NeurIPS 2022.
>
> [3] BQ-NCO: Bisimulation Quotienting for Efficient Neural Combinatorial Optimization, NeurIPS 2023.
>
> [4] Neural Combinatorial Optimization with Heavy Decoder: Toward Large Scale Generalization, NeurIPS 2023.
>
> **Question 1: data augmentation**
>
> We thank the reviewer for this valuable suggestion about data augmentation for the baseline methods. We thus incorporated 32x data augmentation techniques using the method you suggested and present the updated performance of RF on N=50 in the table below. We observe that RF with 32× data augmentation does not yield performance improvement despite the increased inference time. Therefore, we believe that the additional inference time introduced by our method is justified by the performance improvement it delivers.
>
> | method | Avg. Gap | Avg. Time|
> |--------|----------|----------|
> |RF w/ 8x dihedral   | 2.063% | 2.0s |
> |RF w/ 32x symmetric | 3.261% | 6.1s |
> |MoSES(RF) w/ 8x dihedral | 1.535% | 5.8s|
>
> Additionally, we would like to emphasize that the detailed comparisons for each task, as presented in **Table 1** of our paper, are valuable for identifying potential weaknesses on specific tasks. However, in multi-task VRP, we believe it is more fair and appropriate to evaluate a method based on its overall improvement and its ability to outperform baselines across the majority of tasks. Based on this perspective, along with the considerations mentioned above, we believe the increased time overhead is acceptable given the overall favorable empirical improvements over prior methods and is not a major concern in practical applications.
>
> **Question 2: expert interaction**
>
> To thoroughly address this question, we would first like to clarify that in our method, the primary aim of the gating function is to capture the individual insights of solving basis VRP variants present in a given problem instance. While the gating function does admit interactions among experts, its main purpose is to determine which basis solvers offer the most useful insights for the specific problem instance. Additionally, our MoSES framework incorporates a trainable expert that captures a holistic understanding of VRP variants. This holistic understanding naturally includes the correlations among the basis VRP variants present in the given instance. As a result, both the gating function and the trainable expert play important roles: the gating function facilitates the generation of individual insights through expert interactions, while the trainable expert captures broader correlations and relationships among VRP variants. Therefore, we agree that it is important to consider potential interaction mechanisms beyond the gating function employed in our current approach. Such mechanisms could further enhance either the generation of individual insights from basis solvers or the holistic understanding of the given VRP variants, including their interdependencies. Exploring these directions represents a promising and long-term research opportunity. We sincerely appreciate your suggestion, which highlights a valuable path for future work and potential extensions of our framework.
>
> **Question 3: frozen and trainable experts**
>
> This is a significant question, and we believe that **Figures 2\(c) and 2(d)** in the main text of our paper help address it effectively. Before diving into the analysis, we would like to clarify the roles of the gating function and the trainable expert in our method. The gating function is primarily designed to extract individual insights from basis VRP solvers. While it allows for interactions among experts, its main goal is to identify which basis solvers provide the most relevant insights. In contrast, the trainable expert is intended to capture a holistic understanding of the given VRP variant, which naturally includes correlations among the basis VRP variants. It is important to note that we do not employ complex loss functions (e.g., mutual information) to explicitly decouple the features learned by the gating function and the trainable expert. Therefore, the roles described above reflect their primary, though not exclusive, contributions. From Figure 2\(c), we observe that for smaller problem instances ($N=50$), reducing the LoRA rank of the frozen experts (basis solvers) leads to a more significant performance drop than reducing the LoRA rank of the trainable expert. This suggests that for simpler problems, the basis solvers already contain sufficient useful insights, and the unified solver relies less on the holistic understanding provided by the trainable expert. Conversely, Figure 2(d) shows that for larger problem instances ($N=100$), reducing the LoRA rank of the trainable expert results in a greater performance drop than reducing that of the frozen experts. This indicates that for more complex problems, the unified solver requires a deeper, more holistic understanding of the VRP variant, something the trainable expert is better suited to provide.
>
> We sincerely appreciate your insightful comments and thoughtful questions. Your feedback has helped us better articulate characteristics of our approach. We will revise our manuscript accordingly to address your suggestions and ensure greater clarity.

---

> > ### Comment · Reviewer_Tp1S · 2025-08-05
> >
> > Thank you for the additional clarification. My questions have been sufficiently answered. The paper seems like it provides good insight and a good foundation for future approaches. I remain positive on the paper.  My only comment on the significance of the time difference is not that the final application might take too long, but that the additional time spent could be used for bigger or more complex models. Time normalized evaluation typically helps establish the efficiency and performance of the model but we understand that that is more difficult to do in this case.

---

> > > ### Author Response · Authors · 2025-08-05
> > >
> > > We sincerely appreciate the reviewer for the insightful questions and for considering our clarifications throughout the rebuttal. We are greatly grateful for your recognition of the good insight and the good foundation our work provides for future approaches, as well as your positive evaluation. We also appreciate your understanding of the current difficulty in improving both performance and efficiency in the multi-task VRP domain. We believe that exploring more efficient model composition methods is a promising direction for future research to further enhance efficiency. We will revise our manuscript based on your valuable questions and suggestions to improve its clarity and completeness. We remain actively engaged during the discussion period, so please feel free to reach out with any further questions or requests for clarification.

---

### Official Review · Reviewer_F5K3 · 2025-07-03

**Clarity:** 3
**Significance:** 3
**Originality:** 3
**Rating:** 4
**Confidence:** 2

**Summary:**

This paper proposes a framework to address multi-task variants of the Vehicle Routing Problem by leveraging shared structural components. The authors introduce a State-Decomposable MDP formulation that decomposes the state space across basis VRP variants and a Latent-Space SDMDP that enables policy reuse through latent embeddings and a learnable mixture function. The MoSES uses LoRA-based specialized experts and an adaptive gating mechanism to dynamically combine expert outputs. Experimental results on 16 VRP variants suggest that this approach improves solution quality and generalization compared to prior multi-task solvers. Overall, the paper is well-motivated and presents a reasonable balance between theoretical formulation and empirical demonstration.

**Questions:**

-  How will the framework perform by adding explicit task descriptors (e.g., constraint flags) into the gating mechanism to improve interpretability and performance?
- If a basis expert is poorly trained, does the mixture mechanism still maintain reasonable performance? Some robustness analysis would be helpful.

**Ethical Concerns:**

["NO or VERY MINOR ethics concerns only"]

**Quality:**

3

**Strengths And Weaknesses:**

### **Strengths**

- The idea of decomposing complex VRP tasks into shared basis variants is intuitive and offers a structured way to think about multi-task combinatorial problems.
- The latent-space policy mixture through LS-SDMDP is a natural extension that aligns with recent trends in modular learning and policy composition.
- MoSES is designed with practical considerations in mind (e.g., LoRA for parameter efficiency, compatibility with existing backbones), and experiments demonstrate its general applicability across many VRP types.
- The theoretical formulation is sound and helps justify the approach, although some assumptions (e.g., deterministic mixture function) may be strong in practice.
- The experimental section is fairly comprehensive, with comparisons to strong baselines and ablation studies.

### **Weaknesses**

- While the latent reuse framework is promising, the actual gains over baselines (e.g., RF-TE or CaDA) are incremental in some tasks, and come with added inference overhead due to dynamic expert aggregation.
- The gating mechanism does not explicitly use task-specific features, which may limit its specialization capacity or interpretability, especially when handling more diverse task compositions.
- The framework is only evaluated in the VRP domain. While the proposed MDP reformulation is general in theory, no empirical evidence is provided for transferability to other problem types.

---

> ### Author Rebuttal · Authors · 2025-07-31
>
> We sincerely appreciate the reviewers' thoughtful and constructive feedback on our work. We are grateful for the opportunity to address your concerns and clarify any potential misunderstandings.
>
> **Weakness 1： performance and time overhead**
>
> We would like to clarify that the empirical improvements achieved by our method are sufficiently favorable compared to baseline approaches considered. In terms of average optimality gap across 16 VRP tasks, MoSES(RF) achieves 1.535\% compared to RF’s 2.063\% for $N=50$, and 2.782\% compared to RF-TE’s 3.125\% for $N=100$. MoSES(CaDA) achieves 1.515\% versus CaDA’s 1.715\% for $N=50$, and 2.631\% versus CaDA’s 2.766\% for $N=100$. Furthermore, MoSES(RF) outperforms RF on all 16 tasks for both $N=50$ and $N=100$, while MoSES(CaDA) outperforms CaDA on all 16 tasks for $N=50$ and on 13 out of 16 tasks for $N=100$. The detailed comparisons for each task, as presented in Table 1 of our paper, are helpful for identifying potential weaknesses on specific tasks. However, in the multi-task VRP domain, we believe it is fairer and more appropriate to evaluate a method based on its overall improvement and its ability to outperform baselines across the majority of tasks. From this perspective, our method demonstrates favorable empirical performance compared against the baseline approaches.
>
>
> In terms of the average total time overhead for 1,000 problem instances across 16 tasks, MoSES(RF) requires 5.8s compared to RF’s 2.0s for $N=50$, and 20.8s compared to RF’s 8.4s for $N=100$. MoSES(CaDA) requires 7.4s compared to CaDA’s 2.3s for $N=50$, and 24.8s compared to CaDA’s 10.5s for $N=100$. We would like to emphasize that this time overhead is the total time for 1,000 instances per task; the amortized time per instance remains at the microsecond level for both MoSES(RF) and MoSES(CaDA). Additionally, in the VRP domain, it is generally acceptable to increase inference time in order to achieve better solution quality within a given time budget. In practice, many industrial CO tasks allow for extended computation times to obtain high-quality solutions [1]. Thus, several SOTA VRP methods [2,3] commonly use search-based method or solution refinement. Compared to the time increase introduced by such techniques, the additional overhead from the network architecture changes is relatively minor. We thus believe that the increased time overhead is acceptable given the favorable empirical improvements over prior methods and may not be a major concern in practical applications.
>
> [1] Simulation-guided Beam Search for Neural Combinatorial Optimization, NeurIPS 2022.
>
> [2] BQ-NCO: Bisimulation Quotienting for Efficient Neural Combinatorial Optimization, NeurIPS 2023.
>
> [3] Neural Combinatorial Optimization with Heavy Decoder: Toward Large Scale Generalization, NeurIPS 2023.
>
> **Weakness 2: gating mechanism**
>
> We would like to clarify that the dense routing strategy in the gating mechanism does not hinder interpretability, as evidenced by the analysis in **Figure 9 of the Appendix**. Figure 9 presents a behavioral analysis of the gating functions used in our models across 16 VRP variants. In this figure, we plot the scores assigned by the gating function to each basis solver. It is clear that for a given VRP variant, the gating mechanism tends to assign higher scores to basis solvers associated with the corresponding underlying basis VRPs. For example, in OVRPB, the gating mechanism assigns higher scores to the basis solvers corresponding to CVRP, OVRP, and VRPB. The reason the gating function does not completely zero out the remaining tasks is that many VRP variants are inherently correlated; solving instances of one task may benefit from insights learned from others. Additionally, we would like to emphasize that since our method reuses only the basis VRP solvers, the number of solvers increases linearly, which is more manageable compared to the exponential growth in neural solver. In future work, we plan to explore strategies for pruning basis solvers for specific tasks to further mitigate the potential increase in solver composition.
>
> **Weakness 3: transferability to other domains**
>
> Firstly, we would like to clarify that our work specifically focuses on the multi-task VRP setting, which reflects a more practical demand in real-world applications. Although our primary focus is on multi-task VRP, the proposed theoretical framework still maintains generality and can potentially be extended to other domains. We believe that exploring such extensions represents a promising direction for future research, but they are not the central objective of this work. Then, we would like to consider an example from robotics to illustrate this potential extension beyond multi-task VRPs. To extend our work to a more general multi-task learning setting, one plausible approach would be to define a set of basis tasks, from which all target tasks can be derived. Suppose the goal is to learn a robot policy for washing a cup. The basis tasks might include picking up the cup, filling it with water, and using a cloth to clean it. Notably, the tasks of picking up the cup and filling it with water could also be components of another task, such as watering plants. This example demonstrates how basis tasks can be reused across different target tasks, highlighting the potential of our framework to generalize to broader multi-task learning scenarios. While this may not cover all real-world scenarios, it does encompass a broad spectrum of applications, further highlighting the potential of our framework to be extended beyond multi-task VRPs to more general and practical domains.
>
> **Question 1: task descriptors**
>
> We appreciate this insightful question, which identifies an important avenue for potential performance improvement. To explore this, we augmented the gating mechanism's input to explicitly include constraint flags (i.e., a binary vector). We implemented this modification on MoSES(RF) under the $N=50$ setting. We hope you understand that we limited our implementation to this setting due to constraints in time and computational resources. In the table below, we observe that incorporating task descriptors into the gating mechanism leads to a decrease in optimality gap for some tasks, while it increases for others. Overall, the performance impact is not significant. We speculate that the gating mechanism is capable of implicitly identifying different VRP variants based solely on the problem instance. As a result, the additional task descriptors do not provide a performance gain.
>
> | VRP Variant | w/o task descriptors | w/ task descriptors |
> |-------------|---------------------|-----------------|
> | CVRP        | 0.900%  | 0.880% |
> | VRPTW       | 1.445%  | 1.452% |
> | OVRP        | 1.892%  | 1.940% |
> | VRPL        | 1.089%  | 1.058% |
> | VRPB        | 2.342%  | 2.339% |
> | OVRPTW      | 0.959%  | 0.950% |
> | VRPBL       | 3.185%  | 3.187% |
> | VRPBLTW     | 1.370%  | 1.382% |
> | VRPBTW      | 1.121%  | 1.109% |
> | VRPLTW      | 1.811%  | 1.820% |
> | OVRPB       | 1.979%  | 1.995% |
> | OVRPBL      | 2.014%  | 2.013% |
> | OVRPBLTW    | 0.791%  | 0.786% |
> | OVRPBTW     | 0.783%  | 0.789% |
> | OVRPL       | 1.917%  | 1.941% |
> | OVRPLTW     | 0.962%  | 0.952% |
> | **Average** | **1.535%** | **%1.537**|
>
> **Question 2: robustness to poorly trained basis solver**
>
> We appreciate this insightful question regarding the robustness of our method. To explore this, we conducted experiments using MoSES(RF) under the $N=50$ setting. In our method, each basis solver typically uses a LoRA rank of 32. We observed that reducing the LoRA rank to 4 significantly degrades performance, so we used basis solvers with a LoRA rank of 4 to simulate poorly trained basis solvers. Specifically, we trained four such poor basis solvers, each corresponding to one of the basis VRP variants. We then replaced one of the original high-performing solvers in MoSES(RF) with a corresponding poor solver and retrained the gating mechanism accordingly. In the four tables below, the first columns labeled "O", "L", "B", and "TW" represent the basis solvers corresponding to the respective basis VRP variants. Similarly, the first row labeled "O", "L", "B", and "TW" denotes the eight VRP variant tasks, with or without the associated constraints. From the results, we observe that the optimality gap is more significantly affected for tasks that include the corresponding constraints when the related basis solver is poorly trained. This observation is indeed reasonable and aligns with our expectations.
>
> |Method              | tasks w/ "O" | tasks w/o "O"|
> |--------------------|--------------|--------------|
> |fully trained       | 1.412% | 1.658% |
> |poorly trained "O"  | 1.524% | 1.654% |
>
> |Method              | tasks w/ "L" | tasks w/o "L"|
> |--------------------|--------------|--------------|
> |fully trained       | 1.642% | 1.428% |
> |poorly trained "L"  | 1.660% | 1.453% |
>
> |Method              | tasks w/ "B" | tasks w/o "B"|
> |--------------------|--------------|--------------|
> |fully trained       | 1.698% | 1.372% |
> |poorly trained "B"  | 1.722% | 1.393% |
>
> |Method              | tasks w/ "TW" | tasks w/o "TW"|
> |--------------------|--------------|--------------|
> |fully trained       | 1.155% | 1.915%  |
> |poorly trained "TW" | 1.378% | 1.969%  |
>
> We sincerely thank the reviewer for the insightful comments and thoughtful questions, which have helped us better understand and refine our method. We will revise our manuscript based on your feedback and include the additional experiments you suggested.

---

> > ### Comment · Reviewer_F5K3 · 2025-08-05
> >
> > Thanks for your clarification, and I will keep my original score.

---

> > > ### Author Response · Authors · 2025-08-05
> > >
> > > We sincerely appreciate the reviewer for the thoughtful comments and for considering our clarifications throughout the rebuttal. We are grateful for the opportunity to participate in this discussion. We will revise our manuscript based on your valuable questions and suggestions to improve its clarity and completeness. We remain actively engaged during the discussion period, so please feel free to reach out with any further questions or requests for clarification.

---

### Decision · Program_Chairs · 2025-09-17

**Decision:**

Accept (poster)

**Comment:**

This paper introduces the State-Decomposable MDP (SDMDP) and the Mixture-of-Specialized-Experts Solver (MoSES) to address multi-task VRPs. The formulation enables policy reuse across basis VRP variants, and the empirical evaluation over 16 tasks shows consistent improvements compared to strong baselines, supported by ablation studies and larger-scale tests. While reviewers noted concerns about runtime overhead and limited exploration beyond VRPs, the authors provided convincing clarifications that the inference cost remains acceptable and the framework offers a solid foundation for future extensions. Overall, the paper is theoretically sound, original, and demonstrates meaningful impact on multi-task combinatorial optimization.